# Provable Partially Observable Reinforcement Learning with Privileged Information

**Yang Cai**[1]    **Xiangyu Liu**[2]    **Argyris Oikonomou**[1]    **Kaiqing Zhang**[2]

[1] Yale University        [2] University of Maryland, College Park

`yang.cai@yale.edu, xyliu999@umd.edu`
`argyris.oikonomou@yale.edu, kaiqing@umd.edu`

## Abstract

Partial observability of the underlying states generally presents significant challenges for reinforcement learning (RL). In practice, certain *privileged information*, e.g., the access to states from simulators, has been exploited in training and has achieved prominent empirical successes. To better understand the benefits of privileged information, we revisit and examine several simple and practically used paradigms in this setting. Specifically, we first formalize the empirical paradigm of *expert distillation* (also known as *teacher-student* learning), demonstrating its pitfall in finding near-optimal policies. We then identify a condition of the partially observable environment, the *deterministic filter condition*, under which expert distillation achieves sample and computational complexities that are *both* polynomial. Furthermore, we investigate another successful empirical paradigm of *asymmetric actor-critic*, and focus on the more challenging setting of observable partially observable Markov decision processes. We develop a belief-weighted asymmetric actor-critic algorithm with polynomial sample and quasi-polynomial computational complexities, in which one key component is a new provable oracle for learning belief states that preserves *filter stability* under a misspecified model, which may be of independent interest. Finally, we also investigate the provable efficiency of partially observable multi-agent RL (MARL) with privileged information. We develop algorithms featuring *centralized-training-with-decentralized-execution*, a popular framework in empirical MARL, with polynomial sample and (quasi-)polynomial computational complexities in both paradigms above. Compared with a few recent related theoretical studies, our focus is on understanding practically inspired algorithmic paradigms, without computationally intractable oracles.

## 1 Introduction

In most real-world applications of reinforcement learning (RL), e.g., perception-based robot learning [46, 3], autonomous driving [73, 41], dialogue systems [88], and clinical trials [77], only *partial observations* of the environment state are available for sequential decision-making. Such partial observability presents significant challenges for efficient decision-making and learning, with known computational [66] and statistical [43, 36] barriers under the general model of partially observable Markov decision processes (POMDPs). The curse of partial observability becomes severer when *multiple* RL agents interact, where not only the environment state, but also other agents' information, are not fully-observable in decision-making [85, 80].

On the other hand, a flurry of empirical paradigms has made partially observable (multi-agent) RL promising in practice. One notable example is to exploit the *privileged information* that may be available (only) during training. The privileged information usually includes direct access to the underlying states, as well as access to other agents' observations/actions in multi-agent RL (MARL), due to the use of simulators and/or high-precision sensors for training. The latter is also known as

38th Conference on Neural Information Processing Systems (NeurIPS 2024).

the *centralized-training-with-decentralized-execution* (CTDE) framework in deep MARL, and has become prevalent in practice [53, 70, 22, 82]. These approaches can be mainly categorized into two types: i) privileged *policy* learning, where an expert/teacher policy is trained with privileged information, and then *distilled* into a student partially observable policy. This *expert distillation*, also known as *teacher-student learning*, approach has been the key to some empirical successes in robotic locomotion [45, 59] and autonomous driving [14]; ii) privileged *value* learning, where a value function is trained conditioned on privileged information, and used to improve a partially observable policy. It is typically instantiated as the *asymmetric actor-critic* algorithm [68], and serves as the backbone of some high-profiled successes in robotic manipulation [46, 3] and MARL [53, 82].

Despite the remarkable empirical successes, theoretical understandings of partially observable RL with privileged information have been rather limited, except for a few recent prominent advances in RL with *hindsight observability* [44, 30] (see Appendix B for a detailed discussion). However, most of these theoretically sound algorithms are different from those used in practice, and require computationally intractable oracles to achieve provable sample efficiency. The soundness and efficiency of the aforementioned paradigms used in practice remain elusive. In this work, we examine both paradigms of expert distillation and asymmetric actor-critic, with foresight privileged information as in these empirical works. In contrast to [44, 30], which purely focused on sample efficiency, we aim to understand the benefits of privileged information by examining these practically inspired paradigms under several POMDP models, without computationally intractable oracles. We defer a detailed literature review to Appendix B, and summarize our contribution as follows.

**Contributions.** We first formalize the empirical paradigm of *expert distillation*, and demonstrate its pitfall in distilling near-optimal policies even in observable POMDPs, a model class that was recently shown to allow provable partially observable RL without computationally intractable oracles [25]. We then identify a new condition for POMDPs, the *deterministic filter* condition, and establish sample and computational complexities that are *both* polynomial for expert distillation. The new condition is weaker and thus encompasses several known (statistically) tractable POMDP models (see Figure 1 for a summary). Further, we revisit the *asymmetric actor-critic* paradigm and analyze its efficiency under the more challenging setting of observable POMDPs above (where expert distillation fails). Identifying the inefficiency of vanilla asymmetric actor-critic, and inspired by the empirical success in *belief-state-learning*, we develop a new *belief-weighted* version of asymmetric actor-critic, with polynomial-sample and quasi-polynomial-time complexities. Key to the results is a new belief-state learning oracle that preserves *filter stability* under a misspecified model, which may be of independent interest. Finally, we also investigate the provable efficiency of partially observable multi-agent RL with privileged information, by studying algorithms under the CTDE framework, with polynomial-sample and (quasi-)polynomial-time complexities in both paradigms above.

## 2 Preliminaries

### 2.1 Partially Observable RL (with Privileged Information)

**Model.** Consider a POMDP characterized by a tuple $\mathcal{P} = (H, \mathcal{S}, \mathcal{A}, \mathcal{O}, \mathbb{T}, \mathbb{O}, \mu_1, r)$, where $H$ denotes the length of each episode, $\mathcal{S}$ is the state space with $|\mathcal{S}| = S$, $\mathcal{A}$ denotes the action space with $|\mathcal{A}| = A$. We use $\mathbb{T} = \{\mathbb{T}_h\}_{h \in [H]}$ to denote the collection of transition matrices, so that $\mathbb{T}_h(\cdot \,|\, s, a) \in \Delta(\mathcal{S})$ gives the probability of the next state if action $a$ is taken at state $s$ and step $h$. In the following discussions, for any given $a$, we treat $\mathbb{T}_h(a) \in \mathbb{R}^{|\mathcal{S}| \times |\mathcal{S}|}$ as a matrix, where each row gives the probability for reaching each next state from different current states. We use $\mu_1$ to denote the distribution of the initial state $s_1$, and $\mathcal{O}$ to denote the observation space with $|\mathcal{O}| = O$. We use $\mathbb{O} = \{\mathbb{O}_h\}_{h \in [H]}$ to denote the collection of emission matrices, so that $\mathbb{O}_h(\cdot \,|\, s) \in \Delta(\mathcal{O})$ gives the emission distribution over the observation space $\mathcal{O}$ at state $s$ and step $h$. For notational convenience, we will at times adopt the matrix convention, where $\mathbb{O}_h$ is a matrix with rows $\mathbb{O}_h(\cdot \,|\, s)$ for each $s \in \mathcal{S}$. Finally, $r = \{r_h\}_{h \in [H]}$ is a collection of reward functions, so that $r_h(s, a) \in [0, 1]$ is the reward given the state $s$ and action $a$ at step $h$. When privileged information is available, the agent can observe the underlying state $s \in \mathcal{S}$ directly *during training* (only). We thus denote the trajectory until step $h$ *with states* as $\overline{\tau}_h = (s_{1:h}, o_{1:h}, a_{1:h-1})$, the one *without states* as $\tau_h = (o_{1:h}, a_{1:h-1})$, and its space as $\mathcal{T}_h$. Finally, we use $\boldsymbol{b}_h(\tau_h)$ to denote the posterior distribution over the underlying state at step $h$ given history $\tau_h$, which is known as the *belief state* (c.f. Appendix C.1 for more details).

**Policy and value function.** We define a stochastic policy at step $h$ as:
$$\pi_h : \mathcal{O}^h \times \mathcal{A}^{h-1} \to \Delta(\mathcal{A}), \tag{2.1}$$

where the agent bases on the entire (partially observable) history for decision-making. The corresponding policy class is denoted as $\Pi_h$. We further denote $\Pi = \times_{h \in [H]} \Pi_h$. We also define $\Pi^{\text{gen}} := \{\pi_{1:H} \mid \pi_h : \mathcal{S}^h \times \mathcal{O}^h \times \mathcal{A}^{h-1} \to \Delta(\mathcal{A}) \text{ for } h \in [H]\}$ to be the most general policy space in partially observable RL with privileged state information, which can potentially depend on all historical states, observations, and actions. It can be seen that $\Pi \subseteq \Pi^{\text{gen}}$. We may also use policies that only receive a *finite memory* instead of the whole history as inputs: fix an integer $L > 0$, we define the policy space $\Pi^L$ to be the space of all possible policies $\pi = \pi_{1:H} := (\pi_h)_{h \in [H]}$ such that $\pi_h : \mathcal{Z}_h \to \Delta(\mathcal{A})$ with $\mathcal{Z}_h := \mathcal{O}^{\min\{L,h\}} \times \mathcal{A}^{\min\{L,h\}}$ for each $h \in [H]$. Finally, we define the space of state-based policies as $\Pi_{\mathcal{S}}$, i.e., for any $\pi = \pi_{1:H} \in \Pi_{\mathcal{S}}$, $\pi_h : \mathcal{S} \to \Delta(\mathcal{A})$ for all $h \in [H]$.

Given the POMDP model $\mathcal{P}$, we write $\mathbb{P}^{\mathcal{P}}_{s_{1:H+1}, a_{1:H}, o_{1:H} \sim \pi}(\mathcal{E})$ to denote the event $\mathcal{E}$ when $(s_{1:H+1}, a_{1:H}, o_{1:H})$ is drawn as a trajectory following the policy $\pi$ in the model $\mathcal{P}$. We will also use the shorthand notation $\mathbb{P}^{\pi, \mathcal{P}}(\cdot)$ if $(s_{1:H+1}, a_{1:H}, o_{1:H})$ is evident. We write $\mathbb{E}^{\mathcal{P}}_{\pi}[\cdot]$ to denote the expectation similarly. We define the value function at step $h$ as $V^{\pi, \mathcal{P}}_h(y_h) := \mathbb{E}^{\mathcal{P}}_{\pi}[\sum_{t=h}^{H} r_t(s_t, a_t) \mid y_h]$, denoting the expected accumulated rewards from step $h$, where $y_h \subseteq (s_{1:h}, o_{1:h}, a_{1:h-1})$, and we slightly abuse the notation by treating as a set the sequence of states $s_{1:h}$, the sequence of observations $o_{1:h}$, and the sequence of actions $a_{1:h-1}$ up to time $h$, which are the available information to the agent at step $h$. We say $y_h$ is *reachable* if there exists some policy $\pi \in \Pi^{\text{gen}}$ such that $\mathbb{P}^{\pi, \mathcal{P}}(y_h) > 0$. For $h = 1$, we adopt the simplified notation $v^{\mathcal{P}}(\pi) = \mathbb{E}^{\mathcal{P}}_{\pi}[\sum_{h=1}^{H} r_h(s_h, a_h)]$. Meanwhile, we also define $Q^{\pi, \mathcal{P}}_h(y_h, a_h) := \mathbb{E}^{\mathcal{P}}_{\pi}[\sum_{t=h}^{H} r_t(s_t, a_t) \mid y_h, a_h]$. We denote the occupancy measure on the state space as $d^{\pi, \mathcal{P}}_h(s_h) = \mathbb{P}^{\pi, \mathcal{P}}(s_h)$. The goal of learning in POMDPs is to find the optimal policy that *maximizes* the expected accumulated reward over the policies that take $\tau_h$ as input at each step $h \in [H]$, i.e., those $\pi \in \Pi$. Formally, we define:

**Definition 2.1** ($\epsilon$-optimal policy). Given $\epsilon > 0$, a policy $\pi^\star \in \Pi$ is $\epsilon$-optimal, if $v^{\mathcal{P}}(\pi^\star) \geq \max_{\pi \in \Pi} v^{\mathcal{P}}(\pi) - \epsilon$.

**Learning with privileged information.** Common RL algorithms for POMDPs deal with the scenario where during *both* the training and test time, the agent can only observe its historical observations and actions $\tau_h$ at step $h$, while the states are not accessible. In other words, the agent can only utilize policies from $\Pi$ to interact with the environment. In contrast, in settings with *privileged information*, e.g., training in simulators and/or using sensors with higher precision, the underlying state can be used in training. Thus, the agent is allowed to utilize policies from the class $\Pi^{\text{gen}}$ during training. Meanwhile, the objective is still to find the optimal history-dependent policy in the space of $\Pi$, since at test time, the agent cannot access the state information anymore, and it is the performance for such policies that matters eventually. For simplicity, we assume the reward function is known since under our privileged information setting, learning the reward function is much easier than learning the transition and emission, and the sample/computational complexity for the former is dominated by that for the latter. This assumption has also been made for learning in POMDPs without privileged information [36, 47, 48].

## 2.2 Partially Observable Multi-agent RL with Information Sharing

Partially observable stochastic games (POSGs) are a natural generalization of POMDPs with multiple agents of potentially independent interests. We define a POSG with $n$ agents by a tuple $\mathcal{G} = (H, \mathcal{S}, \{\mathcal{A}_i\}_{i=1}^n, \{\mathcal{O}_i\}_{i=1}^n, \mathbb{T}, \mathbb{O}, \mu_1, \{r_i\}_{i=1}^n)$, where each agent $i$ has its individual action space $\mathcal{A}_i$, observation space $\mathcal{O}_i$, and reward function $r_i = \{r_{i,h}\}_{h \in [H]}$ with $r_{i,h}(s, a) \in [0, 1]$ denoting the reward given state $s$ and joint action $a$ for agent $i$ at step $h$. An episode of POSG proceeds as follows: at each step $h$ and state $s_h$, a joint observation is drawn from $(o_{i,h})_{i \in [n]} \sim \mathbb{O}_h(\cdot \mid s_h)$, and each agent receives its own observation $o_{i,h}$, takes the corresponding action $a_{i,h}$, obtains the reward $r_{i,h}(s_h, a_h)$, where $a_h := (a_{i,h})_{i \in [n]}$, and then the system transitions to the next state as $s_{h+1} \sim \mathbb{T}_h(\cdot \mid s_h, a_h)$. Notably, each agent $i$ may not only know its local information $(o_{i,1:h}, a_{i,1:h-1})$, but also information from some other agents. Therefore, we denote the information available to each agent $i$ at step $h$ also as $\tau_{i,h} \subseteq (o_{1:h}, a_{1:h-1})$ and define the *common information* as $c_h = \cap_{i \in [n]} \tau_{i,h}$ and *private information* as $p_{i,h} = \tau_{i,h} \setminus c_h$. We denote the space for common information and private information as $\mathcal{C}_h$ and $\mathcal{P}_{i,h}$ for each agent $i$ and step $h$. The joint private information at step $h$ is denoted as $p_h = (p_{i,h})_{i \in [n]}$, where the collection of the joint private information is given by $\mathcal{P}_h = \mathcal{P}_{1,h} \times \cdots \times \mathcal{P}_{n,h}$. We refer more examples of this setting of POSG with information-sharing to Appendix C.2 (and also [62, 63, 51]). Correspondingly, the policy each agent $i$ deploys at test time takes the form of $\pi_{i,h} : \Omega_h \times \mathcal{C}_h \times \mathcal{P}_{i,h} \to \Delta(\mathcal{A}_i)$, where $\Omega_h$ is the space of random seeds. We denote the policy

| | Without PI | With PI (Ours) |
|---|---|---|
| Block MDP | With STD: Oracle-efficient [17, 18, 60] Without additional assump.: Computationally harder than SL [28] | Tabular: Poly sample + time |
| $k$-decodable POMDP | Exponential-in-$k$ sample + time [19] | FA: Poly sample + Classification (SL) oracle |
| Det. POMDP | Without WSE: Statistically hard [47] With WSE: Poly sample + time [36] | |
| POSG with det. filter | N/A | Poly sample + time |
| Observable POMDP | Quasi-poly sample + time [25] [51] | Poly sample + Quasi-poly time |
| Observable POSG | | |

Table 1: Comparison of the theoretical guarantees with and without privileged information. PI: privileged information; STD: structural assumptions on transition dynamics, e.g., deterministic transition or reachability of all states; SL: supervised learning; FA: function approximation; WSE: well-separated emission.

Figure 1: A landscape of POMDP models that partially observable RL with privileged information can/cannot address. The axes denote the "restrictiveness" of the assumptions, on the emission channels and transition dynamics, respectively.

space for agent $i$ as $\Pi_i$. If $\pi_{i,h}$ takes the state $s_h$ instead of $(c_h, p_{i,h})$ as input, we denote its policy space as $\Pi_{\mathcal{S},i}$, e.g., for each agent $i$, and policy $\pi_{1:H} \in \Pi_{\mathcal{S},i}$, we have $\pi_{i,h} : \mathcal{S} \to \Delta(\mathcal{A}_i)$ for each step $h \in [H]$. Similar to the POMDP setting, we define $\Pi^{\text{gen}}$ to be the most general policy space, i.e., $\Pi^{\text{gen}} := \{\pi_{1:H} \mid \pi_h : \mathcal{S}^h \times \mathcal{O}^h \times \mathcal{A}^{h-1} \to \Delta(\mathcal{A}) \text{ for } h \in [H]\}$. Note that this model covers several recent POSG models studied for partially observable MARL, e.g., [49, 27]. For example, at each step $h$, if there is no shared information, then $c_h = \emptyset$, and if all history information is shared, then $p_{i,h} = \emptyset$ for all $i \in [n]$. In *privileged-information*-based learning, the training algorithm may exploit not only the underlying state information, but also the observations and actions of other agents.

**Solution concepts.** The solution concepts for POSGs are usually the *equilibria*, particularly Nash equilibrium (NE) for two-player zero-sum games (i.e., when $n = 2$ and $r_{1,h} + r_{2,h} = 1$),[1] and correlated equilibrium (CE) or coarse correlated equilibrium (CCE) for general-sum games. We defer the formal definitions of these standard solution concepts to Appendix C.2.

### 2.3 Technical Assumptions for Computational Tractability

A key technical assumption is that the POMDPs/POSGs we consider satisfy an *observability* assumption, as outlined below. This observability assumption allows us to use short memory policies to approximate the optimal policy, and yields quasi-polynomial-time complexity for both planning and learning in POMDPs/POSGs [26, 25, 51]. Meanwhile, we defer an additional assumption to ensure the traceability for solving POSGs to Appendix C.3.

**Assumption 2.2** ($\gamma$-observability [20, 26, 25]). Let $\gamma > 0$. For $h \in [H]$, we say that the matrix $\mathbb{O}_h$ satisfies the $\gamma$-observability assumption if for each $h \in [H]$, for any $b, b' \in \Delta(\mathcal{S})$, $\left\| \mathbb{O}_h^\top b - \mathbb{O}_h^\top b' \right\|_1 \geq \gamma \left\| b - b' \right\|_1$. A POMDP/POSG satisfies $\gamma$-observability if all its $\mathbb{O}_h$ for $h \in [H]$ do so.

## 3 Revisiting Empirical Paradigms of RL with Privileged Information

Most empirical paradigms of RL with privileged information can be categorized into two types: i) privileged *policy* learning, where the policy in training is conditioned on the privileged information, and the trained policy is then *distilled* to a policy that does not take the privileged information as input. This is usually referred to as either *expert distillation* [14, 64, 58] or *teacher-student learning* [45, 59, 75] in the literature; ii) privileged *value* learning, where the value function is conditioned on the privileged information, and is then used to directly output a policy that takes partial observation (history) as input. One prominent example of ii) is *asymmetric-actor-critic* [68, 3]. It is worth noting that asymmetric-actor-critic is also closely related to one of the most successful paradigms for multi-agent RL, *centralized-training-with-decentralized-execution* [53, 86, 21], which is usually instantiated under the actor-critic framework, with the critic taking privileged information as input

---

[1]Note that we require $r_{1,h} + r_{2,h}$ to be 1 instead of 0 to be consistent with our assumption that $r_{i,h} \in [0,1]$ for each $i \in [0,1]$, and this requirement does not lose optimality as one can always subtract the constant-sum offset to attain a zero-sum reward structure.

in training. Here we formalize and revisit the potential pitfalls of these two paradigms, and further develop theoretical guarantees under certain additional conditions and/or algorithm variants.

## 3.1 Privileged Policy Learning: Expert Policy Distillation

The motivation behind expert policy distillation is that learning an optimal *fully observable* policy in MDPs is a much easier and better-studied problem with many known efficient algorithms. The (expected) distillation objective can be formalized as follows:

$$\widehat{\pi}^{\star} \in \arg \min_{\pi \in \Pi} \ \mathbb{E}_{\pi'}^{\mathcal{P}} \left[ \sum_{h=1}^{H} D_f \left( \pi_h^{\star}(\cdot \,|\, s_h) \,|\, \pi_h(\cdot \,|\, \tau_h) \right) \right], \tag{3.1}$$

where $\pi' \in \Pi^{\text{gen}}$ is some given behavior policy to collect exploratory trajectories, $\pi^{\star} \in \arg \max_{\pi \in \Pi_{\mathcal{S}}} v^{\mathcal{P}}(\pi)$ denotes the optimal fully observable policy, and $D_f$ denotes the general $f$-divergence to measure the discrepancy between $\pi^{\star}$ and $\pi$.

Such a formulation looks promising since it essentially circumvents the challenging issue of *exploration in partially observable environments*, by directly mimicking an expert policy that can be obtained from any off-the-shelf MDP learning algorithm. However, we point out in the following proposition that even if the POMDP satisfies Assumption 2.2, the distilled policy can still be strictly suboptimal even with infinite data, i.e., by solving the expected objective Equation (3.1) completely. We postpone the proof of Proposition 3.1 to Appendix E.

**Proposition 3.1** (Pitfall of expert policy distillation). *For any $\epsilon, \gamma \in (0, 1)$, there exists a $\gamma$-observable POMDP $\mathcal{P}^{\epsilon}$ with $H = 1$, $S = O = A = 2$ such that for any behavior policy $\pi' \in \Pi^{\text{gen}}$ and choice of $D_f$ in Equation (3.1), it holds that $v^{\mathcal{P}^{\epsilon}}(\widehat{\pi}^{\star}) \leq \max_{\pi \in \Pi} v^{\mathcal{P}^{\epsilon}}(\pi) - \frac{(1-\epsilon)(1-\gamma)}{4}$.*

The key reason Equation (3.1) fails is that in general, the underlying state can remain highly uncertain even given the full history. Thus, the distilled policy may not be able to mimic the state-based expert policy well at different states $s_h$ if the associated $\pi_h^{\star}(\cdot \,|\, s_h)$ differs significantly across $s_h$. To see how we may rule out such an issue, notice that if $\gamma = 1$ (note that according to Assumption 2.2, we have $\gamma$ is at most 1 since $\gamma \leq \|\mathbb{O}_h\|_{\infty} \leq 1$), implying that the observation can decode the underlying state, the bound in Proposition 3.1 becomes vacuous. Inspired by this, we propose the following condition that incorporates this case of $\gamma = 1$, and will be shown to suffice to make expert distillation effective.

**Definition 3.2** (Deterministic filter condition). *We say a POMDP $\mathcal{P}$ satisfies the deterministic filter condition if for each $h \geq 2$, the belief update operator under $\mathcal{P}$ satisfies that there exists an unknown function $\psi_h : \mathcal{S} \times \mathcal{A} \times \mathcal{O} \to \mathcal{S}$ such that for any reachable $s_{h-1} \in \mathcal{S}$, $o_h \in \mathcal{O}$, $a_{h-1} \in \mathcal{A}$, $U_h(b^{s_{h-1}}; a_{h-1}, o_h) = b^{\psi_h(s_{h-1}, a_{h-1}, o_h)}$, where we define for any $s \in \mathcal{S}$, $b^s \in \Delta(\mathcal{S})$ and $b^s(s) = 1$ is a one-hot vector. In addition, for $h = 1$, there exists a function $\psi_1 : \mathcal{O} \to \mathcal{S}$ such that for any reachable $o_1$, $B_1(\mu_1; o_1) = b^{\psi_1(o_1)}$, where $B_h(b; o_h) := \mathbb{P}_{s_h \sim b}^{\mathcal{P}}(\cdot \,|\, o_h) \in \Delta(\mathcal{S})$, $U_h(b; a_{h-1}, o_h) := \mathbb{P}_{s_{h-1} \sim b}^{\mathcal{P}}(\cdot \,|\, a_{h-1}, o_{h-1}) \in \Delta(\mathcal{S})$ are the belief update operators under the Bayes rule for any $b \in \Delta(\mathcal{S})$, for which we defer the formal introduction to Appendix C.1.*

Notably, this condition is weaker than and thus covers several known tractable classes of POMDPs with sample and computation efficiency guarantees including Block MDP, deterministic POMDP, $k$-decodable POMDP as well as a new setting we have identified and existing literature cannot handle. We refer the formal introduction to Appendix E and Figure 1 for an illustration.

In light of the pitfall in Proposition 3.1, we will analyze *both* the computational and statistical efficiencies of expert distillation in Section 4, under the condition in Definition 3.2.

## 3.2 Privileged Value Learning: Asymmetric Actor-Critic

Asymmetric actor-critic [68] iterates between two procedures as in standard actor-critic algorithms [42]: policy *improvement* and policy *evaluation*. As the name suggests, its key difference from the standard actor-critic is that the algorithm maintains $Q$-value functions (the critic) based on the *state/privileged information*, while the policy receives only the (partially observable) *history* as input.

**Policy evaluation.** At iteration $t - 1$, given the policy $\pi^{t-1}$, the algorithm estimates $Q$-functions in the form of $\{Q_h^{t-1}(\tau_h, s_h, a_h)\}_{h \in [H]}$, where we adopt the "unbiased" version [6] such that $Q$-

functions are conditioned on *both* the *history* and the *states*. [2] One key to achieving sample efficiency is by adding some bonus terms in policy evaluation to encourage exploration, i.e., obtaining some *optimistic Q*-function estimates, similarly as in the fully-observable MDP setting, see e.g., [11, 74], for which we defer the detailed introduction to Section 4.

**Policy improvement.** At each iteration $t$, given the critic $\{Q_h^{t-1}(\tau_h, s_h, a_h)\}_{h \in [H]}$ for $\pi^{t-1}$, the vanilla asymmetric actor-critic algorithm updates the policy according to the *sample-based* gradient estimation via $K$ trajectories $\{s_{1:H+1}^k, o_{1:H}^k, a_{1:H}^k\}_{k \in [K]}$ sampled from $\pi^{t-1}$

$$\pi^t \leftarrow \text{PROJ}_\Pi \left( \pi^{t-1} + \frac{\lambda_t}{K} \sum_{k \in [K]} \sum_{h \in [H]} \nabla_\pi \log \pi_h^{t-1}(a_h^k \mid \tau_h^k) Q_h^{t-1}(\tau_h^k, s_h^k, a_h^k) \right), \tag{3.2}$$

where $\lambda_t$ is the step-size and $\text{PROJ}_\Pi$ is the projection operator onto the space of $\Pi$, which corresponds to projecting onto the simplex of $\Delta(\mathcal{A})$ for each $h \in [H]$. Here we point out the potential drawback of the vanilla algorithm as in [68, 6], where the key insight is that for each iteration of policy evaluation and improvement, one roughly *only* performs the computation of order $\mathcal{O}(KH)$, while needing to collect $K$ new episodes of samples. Thus, the *sample complexity* will scale in the same order as the *computational complexity* when the algorithm converges after some iterations to an $\epsilon$-optimal solution, which will be super-polynomial even for $\gamma$-observable POMDPs [27]. Proof of the result is deferred to Appendix E.

**Proposition 3.3** (Inefficiency of vanilla asymmetric actor-critic). Under the tabular parameterization for both the policy and the value function, the vanilla asymmetric actor-critic algorithm (Equation (3.2)) suffers from super-polynomial sample complexity for $\gamma$-observable POMDPs under standard hardness assumptions.

To address such an issue, one may need to perform *more* computation *per iteration*, so that although the *total* computational complexity (iteration number × per-iteration computational complexity) is super-polynomial, the total iteration number can be lower such that the total sample complexity may be lower as well. This desideratum is hard to achieve if one *computes* policy update only on the *sampled* trajectories $\tau_h$ per iteration, i.e., update *asynchronously*, since this will couple the scales of computational and sample complexities similarly as Equation (3.2). In contrast, we first propose to update *all* trajectories per iteration in a *synchronous* way, with the following proximal-policy optimization-type [72] policy improvement update with the state-history-dependent $Q$-functions $\{Q_h^{t-1}(\tau_h, s_h, a_h)\}_{h \in [H]}$:

$$\pi_h^t(\cdot \mid \tau_h) \propto \pi_h^{t-1}(\cdot \mid \tau_h) \exp \left( \eta \mathbb{E}_{s_h \sim \boldsymbol{b}_h(\tau_h)} \left[ Q_h^{t-1}(\tau_h, s_h, \cdot) \right] \right), \forall h \in [H], \tau_h \in \mathcal{T}_h, \tag{3.3}$$

where we recall $\boldsymbol{b}_h(\tau_h) \in \Delta(\mathcal{S})$ denotes the belief state and $\eta > 0$ is the learning rate. This update rule also reduces to the natural policy gradient (NPG) [40] update under the softmax policy parameterization in the fully-observable case [1], when updated for each state $s_h$ separately [74]. We defer the detailed derivation of Equation (3.3) to Appendix E.

However, such an update presents two challenges: (1) It requires enumerating all possible $\tau_h$, whose number scales exponentially with the horizon, making it still computationally intractable; (2) An explicit belief function $\boldsymbol{b}_h$ is needed. Motivated by these two challenges, we propose to consider *finite-memory*-based policy and assume access to an approximate belief function $\{\boldsymbol{b}_h^{\text{apx}} : \mathcal{Z}_h \rightarrow \Delta(\mathcal{S})\}_{h \in [H]}$ (the learning for which will be made clear later). Correspondingly, the policy update is modified as:

$$\pi_h^t(\cdot \mid z_h) \propto \pi_h^{t-1}(\cdot \mid z_h) \exp \left( \eta \mathbb{E}_{s_h \sim \boldsymbol{b}_h^{\text{apx}}(z_h)} \left[ Q_h^{t-1}(z_h, s_h, \cdot) \right] \right), \forall h \in [H], z_h \in \mathcal{Z}_h.$$

Then we develop and analyze one possible approach to learning such an approximate belief efficiently (c.f. Section 5). It is worth noting that the policy optimization algorithm we aim to develop and analyze does not depend on the specific algorithm approximate belief learning. Such a decoupling enables a more modular algorithm design framework, and can potentially incorporate the rich literature on learning approximate beliefs in practice, see e.g., [24, 65, 16, 87, 83], which has mostly not been theoretically analyzed before. We will thus analyze such an oracle in Section 5.

---

[2]As pointed out in [6], the original asymmetric actor-critic [68], where the value function was only conditioned on the *state*, is a *biased* estimate of the actual history-conditioned value function that appears in the policy gradient in the infinite-horizon discounted setting. We verify that such a state-based value function is indeed also biased under our finite-horizon setting, see Remark E.1 for an example. We will thus use the unbiased value function estimate conditioned on *both* the state and the history throughout.

# 4 Provably Efficient Expert Policy Distillation

We now focus on the provable correctness and efficiency of expert policy distillation, under the deterministic filter condition in Definition 3.2. We will defer all the proofs in this section to Appendix F. Definition 3.2 motivates us to consider only succinct policies that incorporate an auxiliary parameter representing the most recent state, as well as the most recent observations and actions. We consider policies that are the composition of two functions: at step $h$ a function $g_h : \mathcal{S} \times \mathcal{A} \times \mathcal{O} \to \mathcal{S}$ that decodes the state based on the previous state, the most recent action, and the most recent observation, and a policy $\pi^E \in \Pi_{\mathcal{S}}$ that takes as input the current (decoded) underlying state and outputs a distribution over actions.

**Definition 4.1.** We define a policy class $\Pi^D$ as:

$$\Pi^D = \left\{ \pi_h^E \left( g_h(s_{h-1}, a_{h-1}, o_h) \right) : g_h : \mathcal{S} \times \mathcal{A} \times \mathcal{O} \to \mathcal{S}, \pi_h^E : \mathcal{S} \to \Delta(\mathcal{A}) \right\}_{h \in [H]},$$

where $\pi^E$ stands for an arbitrary expert policy, and $\Pi^D$ stands for the distilled policy class, and for $h = 1$, $a_0, s_0$ are some fixed dummy action and state. Intuitively, the distilled policy $\pi \in \Pi^D$ executes as follows: it firstly *decodes* the underlying states by applying $\{g_h\}_{h \in [H]}$ *recursively* along the history, and then takes actions using $\pi^E$ based on the decoded states.

Our goal is to learn the two functions independently, that is, we want to learn an approximately optimal policy $\pi^E \in \Pi_{\mathcal{S}}$ with respect to the MDP $\mathcal{M}$ derived from POMDP $\mathcal{P}$ by omitting the observations and observing the underlying state (see Definition 4.2 for a formal definition), and for each step $h \in [H]$, a decoding function $g_h(s_{h-1}, a_{h-1}, o_h)$ such that the probability of *incorrectly* decoding a state-action-observation triplet over the trajectories induced by the policy $\pi^E$ is low.

**Definition 4.2** (POMDP-induced MDP). Given a POMDP $\mathcal{P} = (H, \mathcal{S}, \mathcal{A}, \mathcal{O}, \mathbb{T}, \mathbb{O}, \mu_1, r)$, we define its associated Markov Decision Process (MDP) $\mathcal{M}$ as $\mathcal{M} = (H, \mathcal{S}, \mathcal{A}, \mathbb{T}, \mu_1, r)$ without observations.

**Definition 4.3.** Consider a POMDP $\mathcal{P}$ that satisfies Definition 3.2, and let $\psi = \{\psi_h\}_{h \in [H]}$ be the promised set of functions that always correctly decode a state-action-observation triplet into an underlying state. Consider policy $\widetilde{\pi^E} = \left\{ \pi_h^E(\psi(\cdot)) : \mathcal{S} \times \mathcal{A} \times \mathcal{O} \to \Delta(\mathcal{A}) \right\}_{h \in [H]} \in \Pi^D$. We slightly abuse the notation and simply denote by $v^{\mathcal{P}}(\pi^E) = v^{\mathcal{P}}(\widetilde{\pi^E})$.

**Lemma 4.4.** Let $\mathcal{P} = (H, \mathcal{S}, \mathcal{A}, \mathcal{O}, \mathbb{T}, \mathbb{O}, \mu_1, r)$ be a POMDP that satisfies Definition 3.2, and consider a policy $\pi^E \in \Pi_{\mathcal{S}}$. Consider a set of decoding functions $\{g_h\}_{h \in [H]}$ such that, $\mathbb{P}^{\pi^E, \mathcal{P}} \left[ \exists h \in [H] : g_h(s_{h-1}, a_{h-1}, o_h) \neq s_h \right] \leq \epsilon$. Consider the policy $\pi = \left\{ \pi_h^E(g_h(\cdot)) : \mathcal{S} \times \mathcal{A} \times \mathcal{O} \to \Delta(\mathcal{A}) \right\}_{h \in [H]}$ on the POMDP $\mathcal{P}$, then: $v^{\mathcal{P}}(\pi) \geq v^{\mathcal{P}}(\widetilde{\pi^E}) - H\epsilon$.

We can use any off-the-shelf algorithm to learn an approximate optimal policy $\pi^E$ for the associated MDP $\mathcal{M}$ (see Definition 4.2). Thus, in the rest of the section, we focus on learning the decoding function $\{g_h\}_{h \in [H]}$. To efficiently learn the decoding function, we model the access to the underlying state by keeping track of the most recent pair of the action and the observation, as well as the two most recent states. We summarize the algorithm of decoding-function learning in Algorithm 1.

**Theorem 4.5.** Consider a POMDP $\mathcal{P}$ that satisfies Definition 3.2, a policy $\pi^E \in \Pi_{\mathcal{S}}$, and let $\{g_h\}_{h \in [H]}$ be the output of Algorithm 1 with $M = \frac{AOS + \log(H/\delta)}{\epsilon^2}$. Then, with probability at least $1 - \delta$, for each step $h \in [H]$: $\mathbb{P}^{\pi^E, \mathcal{P}} \left[ \exists h \in [H] : g_h(s_{h-1}, a_{h-1}, o_h) \neq s_h \right] \leq \epsilon$, using $\text{POLY}(H, A, O, S, \frac{1}{\epsilon}, \log\left(\frac{1}{\delta}\right))$ episodes in time $\text{POLY}(H, A, O, S, \frac{1}{\epsilon}, \log\left(\frac{1}{\delta}\right))$.

The following is an immediate consequence of Lemma 4.4 and Theorem 4.5. Note that *both* the sample and computation complexities are *polynomial*, which is in stark contrast to the $k$-decodable POMDP case [19] (a special one covered by our Definition 3.2), for which the sample complexity is necessarily *exponential in $k$* when there is no privileged information [19]. In fact, thanks to privileged information, the complexities are only polynomial in horizon $H$ even when the decodable length is *unknown*. For the benefits of using privileged information in several other subclasses of problems, we refer to Table 1 for more details.

**Theorem 4.6.** Let $\mathcal{P}$ satisfy Definition 3.2 and consider any policy $\pi^E \in \Pi_{\mathcal{S}}$. Using $\text{POLY}(H, A, O, S, \frac{1}{\epsilon}, \log\left(\frac{1}{\delta}\right))$ episodes and in time $\text{POLY}(H, A, O, S, \frac{1}{\epsilon}, \log\left(\frac{1}{\delta}\right))$, we can compute a policy $\pi \in \Pi^D$ (see Definition 4.1) such that with probability at least $1 - \delta$, $v^{\mathcal{P}}(\pi) \geq v^{\mathcal{P}}(\pi^E) - \epsilon$.

**Extension to the case with general function approximation.** Due to the modularity of our algorithmic framework and its compatibility with supervised learning oracles, it can be readily generalized to the function approximation setting to handle large observation spaces. We defer the corresponding results to Appendix G.

# 5 Provable Asymmetric Actor-Critic with Approximate Belief Learning

Unlike most existing theoretical studies on provably sample-efficient partially observable RL [36, 25, 47], which directly learn an approximate *POMDP model* for planning near-optimal policies, we consider a general framework with two steps: firstly learning an approximate *belief function*, followed by adopting a *fully observable RL* subroutine on the belief state space.

## 5.1 Belief-Weighted Optimistic Asymmetric Actor-Critic

We now introduce our main algorithmic contribution to the privileged policy learning setting. Our algorithm is conceptually similar to the natural policy gradient methods [40, 1, 74] in the fully-observable setting, with additional weighting over the states $s_h$ using some learned belief states, to handle the additional *state*-dependence in the asymmetric critic. The overall algorithm is presented in Algorithm 2. The algorithm requires a belief-learning subroutine that takes the stored memory as input and outputs a belief about the underlying state (c.f. $\{b_h^{\text{apx}}\}_{h \in [H]}$). Additionally, similar to the fully observable setting, we include a subroutine to estimate the $Q$-function, which introduces additional challenges due to partial observability (see Appendix H). We establish the performance guarantee of Algorithm 2 in the following theorem. We defer the proof to Appendix H.

**Theorem 5.1** (Near-optimal policy). Fix $\epsilon, \delta \in (0,1)$. Given a POMDP $\mathcal{P}$ and an approximate belief $\{b_h^{\text{apx}} : \mathcal{Z}_h \to \Delta(\mathcal{S})\}_{h \in [H]}$, with probability at least $1 - \delta$, Algorithm 2 can learn an approximate optimal policy $\pi^\star$ of $\mathcal{P}$ in the space of $\Pi^L$ such that $v^{\mathcal{P}}(\pi^\star) \geq \max_{\pi \in \Pi^L} v^{\mathcal{P}}(\pi) - \mathcal{O}(\epsilon + H^2 \epsilon_{\text{belief}})$, with sample complexity $\text{POLY}(S, A, O, H, \frac{1}{\epsilon}, \log \frac{1}{\delta})$ and time complexity $\text{POLY}(S, A, O, H, Z, \frac{1}{\epsilon}, \log \frac{1}{\delta})$, where $\epsilon_{\text{belief}}$ is the belief-learning error defined as $\epsilon_{\text{belief}} := \max_{h \in [H]} \max_{\pi \in \Pi^L} \mathbb{E}_\pi^{\mathcal{P}} \| b_h(\tau_h) - b_h^{\text{apx}}(z_h) \|_1$ and $Z := \max_{h \in [H]} |\mathcal{Z}_h|$. Furthermore, if $\mathcal{P}$ is additionally $\gamma$-observable (c.f. Assumption 2.2), then $\pi^\star$ is also an approximate optimal policy in the space of $\Pi$ such that $v^{\mathcal{P}}(\pi^\star) \geq \max_{\pi \in \Pi} v^{\mathcal{P}}(\pi) - \mathcal{O}(\epsilon + H^2 \epsilon_{\text{belief}})$, as long as $L \geq \widetilde{\Omega}(\gamma^{-4} \log(SH/\epsilon))$.

## 5.2 Approximate Belief Learning

At a high level, our belief-learning algorithm first learns an approximate POMDP model $\widehat{\mathcal{P}}$ by explicitly exploring the state space. The main technical challenge here is that there may exist states that are reachable with very low probability, making it infeasible to collect enough samples to sufficiently explore them, thus potentially breaking the $\gamma$-observability property of the ground-truth model $\mathcal{P}$. To circumvent this issue, we ignore such hard-to-visit states and *redirect* probabilities flowing to them to certain other states. Thus, in our truncated POMDP, where each state is sufficiently explored, we can approximate the transition and emission matrices to a desired accuracy *uniformly across all the states* and preserve the $\gamma$-observability property. This ensures that the learned approximate belief function in the truncated POMDP is sufficiently close to the actual belief function of the original POMDP $\mathcal{P}$. Note that the key to achieving belief learning with *both* polynomial sample and time complexities is our explicit exploration in the state space, which relies on executing *fully observable* policies from an *MDP learning* subroutine. We remark that the belief function may also be learned even if the state space is only explored by partially observable policies, thus utilizing only hindsight observability may be sufficient for this purpose [44]. However, for such exploration to be *computationally tractable*, one requires to avoid using computationally intractable oracles for *POMDP learning*, which is in fact our final goal. We present the guarantees in the next theorem and postpone the proof to Appendix H.

**Theorem 5.2.** Consider a $\gamma$-observable POMDP $\mathcal{P}$ (c.f. Assumption 2.2) and assume that $L \geq \widetilde{\Omega}(\gamma^{-4} \log(SH/\epsilon))$ for an $\epsilon > 0$. Then, we can learn an approximate belief $\{b_h^{\text{apx}}\}_{h \in [H]}$ from Algorithm 4 using $\widetilde{\mathcal{O}}(\frac{S^2 A H^2 O + S^3 A H^2}{\epsilon^2} + \frac{S^4 A^2 H^6 O}{\epsilon \gamma^2})$ episodes in time $\text{POLY}\left(S, H, A, O, \frac{1}{\gamma}, \frac{1}{\epsilon}, \log\left(\frac{1}{\delta}\right)\right)$ such that with probability at least $1 - \delta$, for any $\pi \in \Pi^L$ and $h \in [H]$, $\mathbb{E}_\pi^{\mathcal{P}} \| b_h(\tau_h) - b_h^{\text{apx}}(z_h) \|_1 \leq \epsilon$.

Theorem 5.2 shows that an approximate belief can be learned with both polynomial samples and time, which, combined with Theorem 5.1, yields the final polynomial sample and quasi-polynomial time guarantee below. In contrast to the case without privileged information [25, 27], the sample complexity is reduced from quasi-polynomial to polynomial for $\gamma$-observable POMDPs. Note that the computational complexity remains quasi-polynomial, which is known to be unimprovable even for planning [27]. The key to such an improvement, as pointed out in Section 3.2, is the more practical update rule of actor-critic (in conjunction with our belief-weighted idea), which allows *more computation* at each iteration (instead of only performing computation at the *sampled* finite-memory). This allows the total computation to remain quasi-polynomial, while the overall sample complexity becomes polynomial. A detailed comparison can be found in Table 1.

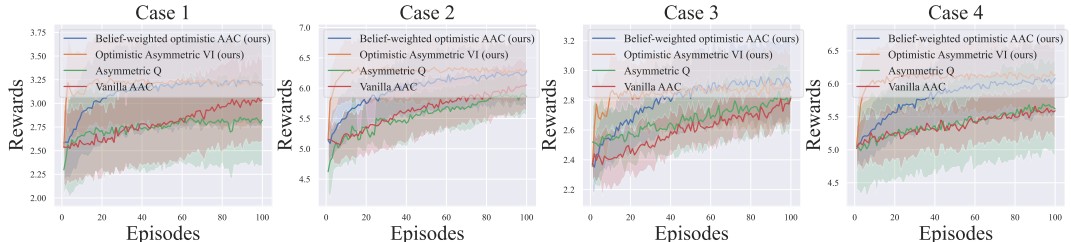

Figure 2: Results for POMDPs of different sizes, where our methods achieve the best performance with the lowest sample complexity (VI: value iteration; AAC: asymmetric actor-critic).

**Theorem 5.3.** Let $\mathcal{P}$ be a $\gamma$-observable POMDP (c.f. Assumption 2.2), and consider $L \geq \widetilde{\Omega}(\gamma^{-4} \log(SH/\epsilon))$ for an $\epsilon > 0$. With probability at least $1 - \delta$, Algorithm 2 can learn a policy $\pi \in \Pi^L$ such that $v^{\mathcal{P}}(\pi) \geq \max_{\pi' \in \Pi} v^{\mathcal{P}}(\pi') - \epsilon$, using POLY$(S, H, 1/\epsilon, 1/\gamma, \log(1/\delta), O, A)$ episodes and in time POLY$(S, H, 1/\epsilon, \log(1/\delta), O^L, A^L)$.

## 6 Numerical Validation

We now provide some numerical results for both of our principled algorithms. Here we mainly compare with two baselines, the vanilla asymmetric actor-critic [68], and asymmetric $Q$-learning [7], on two settings, POMDP under the deterministic filter condition (c.f. Definition 3.2) and general POMDPs. We report the results in Table 2 and Figure 2, where our algorithms converge faster to higher rewards. We defer the implementation details and discussions to Appendix I.

## 7 Extension to Partially Observable MARL with Privileged Information

### 7.1 Privileged Policy Learning: Equilibrium Distillation

To understand how the deterministic filter condition may be extended for POSGs, we first note the following equivalent characterization of Definition 3.2, the proof of which is deferred to Appendix J.

**Proposition 7.1.** Definition 3.2 is equivalent to the following: for each $h \in [H]$, there exists an *unknown* function $\phi_h : \mathcal{T}_h \to \mathcal{S}$ such that $\mathbb{P}^{\mathcal{P}}(s_h = \phi_h(\tau_h) \,|\, \tau_h) = 1$ for any reachable $\tau_h \in \mathcal{T}_h$.

Proposition 7.1 implies that at each step $h$, given the *entire* history information, the agent can uniquely decode the current underlying state $s_h$. Thus, we generalize this condition to POSGs by requiring each agent to uniquely decode the current state $s_h$ given the information it has collected so far.

**Definition 7.2** (Deterministic filter condition for POSGs). We say a POSG $\mathcal{G}$ satisfies the *deterministic filter condition* if for each $i \in [n]$, $h \in [H]$, there exists *an unknown* function $\phi_{i,h} : \mathcal{C}_h \times \mathcal{P}_{i,h} \to \mathcal{S}$ such that $\mathbb{P}^{\mathcal{G}}(s_h = \phi_{i,h}(c_h, p_{i,h}) \,|\, c_h, p_{i,h}) = 1$ for any reachable $(c_h, p_{i,h})$.

Here we have required that each agent can decode the underlying state through their own information *individually*. Naturally, one may wonder whether one can relax it so that only the *joint* history information of all the agents can decode the underlying state. However, we point out in the following that it does not circumvent the computational hardness of POSG, the proof of which is deferred to Appendix J. Note that the computational hardness result can not be mitigated even with privileged state information, as the hardness we state here holds even for the planning problem with model knowledge, with which one can simulate the RL problems with privileged information.

**Proposition 7.3.** Computing CCE in POSGs that satisfy that for each step $h \in [H]$, there exists a function $\phi_h : \mathcal{C}_h \times \mathcal{P}_h \to \mathcal{S}$ such that $\mathbb{P}^{\mathcal{G}}(s_h = \phi_h(c_h, p_h) \,|\, c_h, p_h) = 1$ for any reachable $(c_h, p_h)$ is still PSPACE-hard.

**Learning multi-agent individual decoding functions with unilateral exploration.** Similar to our framework for POMDPs, our framework for POSGs is also decoupled into two steps: i) learning an *expert* equilibrium policy that is fully observable, ii) learning the *decoding function*, where the first step can be instantiated by any provable off-the-shelf algorithm of learning in Markov games. The major difference from the framework for POMDPs lies in how to learn the decoding function. In Theorem J.1, we prove that the difference of the NE/CE/CCE-gap between the expert policy and the distilled student policy is bounded by the decoding errors under policies from the *unilateral deviation* of the expert policy. Hence, given the expert policy $\pi$, the key algorithmic principle is to perform *unilateral exploration* for each agent $i$ to make sure the decoding function is accurate under policies

$(\pi'_i, \pi_{-i})$ for any $\pi'_i$, keeping $\pi_{-i}$ fixed. We refer the detailed algorithm to Algorithm 5, and present below the guarantees for learning the decoding functions and the corresponding distilled policy for learning NE/CCE, while we defer the results for learning CE to Theorem J.6.

**Theorem 7.4** (Equilibria learning; Combining Theorem J.1 and Theorem J.4). Under Assumption C.8 and conditions of Definition 7.2, given a $\frac{\epsilon}{2}$-NE/CCE $\pi^E$ for the associated Markov game of $\mathcal{G}$, Algorithm 5 can learn decoding function $\{\widehat{g}_{i,h}\}_{i \in [n], h \in [H]}$ such that with probability at least $1 - \delta$, it is guaranteed that $\max_{u_i \in \Pi_i, j \in [n]} \mathbb{P}^{u_i \times \pi_{-i}, \mathcal{G}}(s_h \neq \widehat{g}_{j,h}(c_h, p_{j,h})) \leq \frac{\epsilon}{4nH^2}$, for any $i \in [n], h \in [H]$ with both sample and computational complexities POLY$(S, A, H, O, \frac{1}{\epsilon}, \log \frac{1}{\delta})$. Consequently, policy $\pi$ distilled from $\pi^E$ (c.f. Theorem J.1 for the formal distillation procedures) is an $\epsilon$-NE/CCE of $\mathcal{G}$.

## 7.2 Privileged Value Learning: Asymmetric MARL with Approximate Belief Learning

For POMDPs, we have used *finite-memory* policies for computational efficiency. We generalize to POSGs with information sharing by defining the *compression* of the common information.

**Definition 7.5** (Compressed approximate common information [57, 79, 51]). For each $h \in [H]$, given a set $\widehat{\mathcal{C}}_h$, we say Compress$_h$ is a compression function if Compress$_h \in \{f : \mathcal{C}_h \to \widehat{\mathcal{C}}_h\}$. For each $c_h \in \mathcal{C}_h$, we denote $\widehat{c}_h := \text{Compress}_h(c_h)$. We also require the compression function to satisfy the regularity condition that for each $h \in [H]$, there exists a function $\widehat{\Lambda}_{h+1}$ such that $\widehat{c}_{h+1} = \widehat{\Lambda}_{h+1}(\widehat{c}_h, \varpi_{h+1})$, for any $c_h \in \mathcal{C}_h, \varpi_{h+1} \in \Upsilon_{h+1}$, where we recall $c_{h+1} := c_h \cup \varpi_{h+1}$ and the definition of $\Upsilon_{h+1}$ in Assumption C.7.

Similar to the framework we developed for POMDPs in Section 5, we firstly develop the multi-agent RL algorithm based on some approximate belief, and then instantiate it with one provable approach for learning such an approximate belief.

**Optimistic value iteration of POSGs with approximate belief.** For POMDPs, the sufficient statistics for optimal decision-making is the posterior distribution over the state given history. However, for POSGs with information-sharing, as shown in [63, 62, 51], the sufficient statistics become the posterior distribution over the state *and the private information* given the common information, instead of only the state. Therefore, we consider the approximate belief in the form of $\widehat{P}_h : \widehat{\mathcal{C}}_h \to \Delta(\mathcal{P}_h \times \mathcal{S})$ for each $h \in [H]$, where we define the error compared with the ground-truth belief to be $\epsilon_{\text{belief}} := \max_{h \in [H]} \max_{\pi \in \Pi} \mathbb{E}_\pi^{\mathcal{G}} \sum_{s_h, p_h} |\mathbb{P}^{\mathcal{G}}(s_h, p_h \mid c_h) - \widehat{P}_h(s_h, p_h \mid \widehat{c}_h)|$, i.e., the *expected* total variation distance from the true one. Note that both $\widehat{P}_h$ and thus $\epsilon_{\text{belief}}$ have implicit dependencies on Compress$_h$, as $\widehat{c}_h := \text{Compress}_h(c_h)$. We outline our algorithm in Algorithm 7, which is conceptually similar to the algorithm for POMDPs (Algorithm 2), maintaining the asymmetric critic (i.e., value function), and performing the actor update (i.e., policy update) using the *belief-weighted* value function.

**Theorem 7.6** (Equilibria learning; Combining Theorem J.15 and Theorem J.16). Fix $\epsilon, \delta \in (0, 1)$. Under Assumption C.8, with probability at least $1 - \delta$, Algorithm 7 can learn an $(\epsilon + H^2 \epsilon_{\text{belief}})$-NE if $\mathcal{G}$ is zero-sum and $(\epsilon + H^2 \epsilon_{\text{belief}})$-CE/CCE if $\mathcal{G}$ is general-sum with sample complexity $\mathcal{O}(\frac{H^4 SAO \log(SAHO/\delta)}{\epsilon^2})$ and computational complexity POLY$(S, (AO)^{\mathcal{O}(\gamma^{-4} \log(SH/\epsilon))}, H, \frac{1}{\epsilon}, \log \frac{1}{\delta})$.

**Learning approximate belief with model truncation.** The belief learning algorithm we design for POSGs is conceptually similar to that we designed for POMDPs, where the key to achieving *both* polynomial sample and computational complexity is still to firstly learn approximate models, i.e., transitions and emissions, and then carefully *truncate* (as in Section 5.2) its transition and emission to build the approximate belief, where we defer the detailed algorithm to Algorithm 8. Next, we provide its provable guarantees, which leads to a final polynomial-sample and quasi-polynomial-time complexity result when combined with Theorem 7.6.

**Theorem 7.7.** For any $\epsilon > 0$, under Assumption 2.2, it holds that one can learn the approximate belief $\{\widehat{P}_h : \widehat{\mathcal{C}}_h \to \Delta(\mathcal{S} \times \mathcal{P}_h)\}_{h \in [H]}$ such that $\epsilon_{\text{belief}} \leq \frac{\epsilon}{H^2}$ with both polynomial sample complexity and computational complexity POLY$(S, A, O, H, \frac{1}{\gamma}, \frac{1}{\epsilon}, \log \frac{1}{\delta})$ for all the examples in Appendix C.3. As a consequence, Algorithm 7 can learn an $\epsilon$-NE if $\mathcal{G}$ is zero-sum and $\epsilon$-CE/CCE if $\mathcal{G}$ is general-sum with sample complexity $\mathcal{O}(\frac{H^4 SAO \log(SAHO/\delta)}{\epsilon^2})$ and computational complexity POLY$(S, (AO)^{\mathcal{O}(\gamma^{-4} \log(SH/\epsilon))}, H, \frac{1}{\epsilon}, \log \frac{1}{\delta})$.

## Acknowledgement

The authors would like to thank the anonymous reviewers and area chair from NeurIPS 2024 for the valuable feedback. Y.C. acknowledges the support from the NSF Awards CCF-1942583 (CAREER) and CCF-2342642. X.L. and K.Z. acknowledge the support from Army Research Laboratory (ARL) Grant W911NF-24-1-0085. K.Z. also acknowledges the support from Simons-Berkeley Research Fellowship. A.O. acknowledges financial support from a Meta PhD fellowship, a Sloan Foundation Research Fellowship and the NSF Award CCF-1942583 (CAREER). Part of this work was done while the authors were visiting the Simons Institute for the Theory of Computing.

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

# Supplementary Materials for
# "Provable Partially Observable Reinforcement Learning with Privileged Information"

## Contents

# A   Societal Impact

Our work is theoretical by nature, and aimed at better understanding reinforcement learning under partial observability with privileged information. As such, we do not anticipate any direct positive or negative societal impact from this research.

# B   Related Work

**Provable partial observable RL.**   While POMDPs are generally known to be both statistically hard [43] and computationally intractable [66], a productive line of research has identified several structured subclasses of POMDPs that can be efficiently solved. [43] introduced the class of POMDPs in the rich-observation setting, where the observation space can be large and fully reveal the underlying state, where sample-efficient RL becomes possible [35, 60]. [19] introduced $k$-step decodable POMDPs, where the last $k$ observation-action pairs can uniquely determine the state, and proposed polynomial-sample complexity algorithms (assuming $k$ is a small constant). Beyond settings where the underlying state can be *exactly* recovered, [36, 47] proposed weakly revealing POMDPs, where the observations are assumed to be informative enough. Under the weakly revealing condition (and its variant), there has been a fast-growing line of recent works developing sample-efficient RL algorithms for various settings, see e.g., [84, 15, 12, 54, 48, 90]. Notably, these algorithms are typically computationally inefficient, requiring access to an optimistic planning oracle for POMDPs. On a promising note, [26] showed that in observable POMDPs (see Assumption 2.2), one can have quasi-polynomial-time algorithms for planning the near-optimal policy, which further leads to provable RL algorithms [25, 27] with *both quasi-polynomial* samples and time.

**RL under hindsight observability.**   The closest line of research to ours are the recent theoretical studies for Hindsight Observable Markov Decision Processes (HOMDPs) [44], where the underlying state is revealed at the end of the episode; see also subsequent related works in [30, 76] with different observation feedback models. These works focused purely on *sample efficiency*, and showed that polynomial sample complexity can be achieved without (or by further relaxing) aforementioned structural assumptions of the model (e.g., observability or decodability), in both tabular and/or function approximation settings. However, the algorithms (also) require an oracle for planning or even optimistic planning in a learned approximate POMDP, which are not computationally tractable in general. Indeed, without any structural assumption, learning the optimal policy in HOMDPs is computationally no easier than the planning problem, which thus remains `PSPACE-hard`. [66]. Meanwhile, even under the additional assumption of observability on the *underlying* POMDP model, it is still not clear if these algorithms can avoid computationally intractable oracles, since the approximate POMDP that [44] needs to do planning on at every iteration during learning can be quite different from the underlying model. For example, at the beginning of exploration when not enough samples are collected, or when there exist certain states that remain less explored during the entire learning process, the potentially *misspecified emission (and transition)* may break the observability (or other structural) assumptions made for the underlying POMDP. This makes that single iteration even computationally intractable. In contrast, our focus is on better understanding practically inspired algorithmic paradigms, without computationally intractable oracles, which in practice often do have and use the privileged state information *during* each episode (instead of only at the end) [68, 45, 14].

**Most related empirical works.**   Privileged information has been widely used in empirical partially observable RL, with two main types of approaches based on privileged *policy* and privileged *value* learning, respectively. For the former, one prominent example is expert distillation [14, 64, 58], also known as *teacher-student* learning [45, 59, 75], as we analyze in Section 4. For the latter, asymmetric actor-critic [68] represents one of the well-known examples, with other studies in [7, 3]. Learning privileged value functions (to improve the policies) has also been widely used in multi-agent RL, featured in centralized-training-decentralized-execution, see e.g., [53, 22, 70, 86, 82]. Intriguingly, it was shown that if the privileged value function depends *only on* the state, the associated actor will cause bias [6, 55, 56]. This has thus necessitated the use of history/belief in asymmetric actor-critic, as in our Section 5. Notably, the empirical framework in [83] exactly matches ours, where they exploited the privileged state information in training for *belief learning*, followed by policy optimization over the learned belief states. Indeed, many empirical works explicitly separate the procedures of explicit

belief-state learning and planning [24, 65, 33, 16, 87] as we study in Section 5, oftentimes with privileged state information to supervise the belief learning procedure [61, 5].

## C  Additional Preliminaries

### C.1  Additional Preliminaries on POMDPs

**Belief and approximate belief.**  Although in a POMDP, the agent cannot see the underlying state directly, it can still form a *belief* over the underlying state by the historical observations and actions.

**Definition C.1** (Belief state update). For each $h \in [H + 1]$, the Bayes operator $B_h : \Delta(\mathcal{S}) \times \mathcal{O} \to \Delta(\mathcal{S})$ is defined for $b \in \Delta(\mathcal{S})$, and $o \in \mathcal{O}$ by:

$$B_h(b; o)(x) = \frac{\mathbb{O}_h(o \mid x)b(x)}{\sum_{z \in \mathcal{S}} \mathbb{O}_h(o \mid z)b(z)},$$

for each $x \in \mathcal{S}$. For each $h \in [H + 1]$, the belief update operator $U_h : \Delta(\mathcal{S}) \times \mathcal{A} \times \mathcal{O} \to \Delta(\mathcal{S})$, is defined by

$$U_h(b; a, o) = B_{h+1}(\mathbb{T}_h(a) \cdot b; o),$$

where $\mathbb{T}_h(a) \cdot b$ represents the matrix multiplication. We use the notation $b_h$ to denote the belief update function, which receives a sequence of actions and observations and outputs a distribution over states at the step $h$: the belief state at step $h = 1$ is defined as $\boldsymbol{b}_1(\emptyset) = \mu_1$. For any $2 \le h \le H$ and any action-observation sequence $(a_{1:h}, o_{1:h})$, we inductively define the belief state:

$$\boldsymbol{b}_{h+1}(a_{1:h}, o_{1:h}) = \mathbb{T}_h(a_h) \cdot \boldsymbol{b}_h(a_{1:h-1}, o_{1:h}),$$
$$\boldsymbol{b}_h(a_{1:h-1}, o_{1:h}) = B_h(\boldsymbol{b}_h(a_{1:h-1}, o_{1:h-1}); o_h).$$

We also define the approximate belief update using the most recent $L$-step history. For $1 \le h \le H$, we follow the notation of [26] and define

$$\boldsymbol{b}_h^{\mathrm{apx},\mathcal{P}}(\emptyset; D) = \begin{cases} \mu_1 & \text{if } h = 1 \\ D & \text{otherwise}, \end{cases}$$

where $D \in \Delta(\mathcal{S})$ is the prior for the approximate belief update. Then for any $1 \le h - L < h \le H$ and any action-observation sequence $(a_{h-L:h}, o_{h-L+1:h})$, we inductively define

$$\boldsymbol{b}_{h+1}^{\mathrm{apx},\mathcal{P}}(a_{h-L:h}, o_{h-L+1:h}; D) = \mathbb{T}_h(a_h) \cdot \boldsymbol{b}_h^{\mathrm{apx},\mathcal{P}}(a_{h-L:h-1}, o_{h-L+1:h}; D),$$
$$\boldsymbol{b}_h^{\mathrm{apx},\mathcal{P}}(a_{h-L:h-1}, o_{h-L+1:h}; D) = B_h(\boldsymbol{b}_h^{\mathrm{apx},\mathcal{P}}(a_{h-L:h-1}, o_{h-L+1:h-1}; D); o_h).$$

For the remainder of our paper, we will use an important initialization for the approximate belief, which are defined as $\boldsymbol{b}_h'(\cdot) := \boldsymbol{b}_h^{\mathrm{apx},\mathcal{P}}(\cdot; \mathrm{Unif}(\mathcal{S}))$.

### C.2  Additional Preliminaries for POSGs

**Model.**  We use a general framework of partially observable stochastic games (POSGs) as the model for partially observable MARL. Formally, we define a POSG with $n$ agents by a tuple $\mathcal{G} = (H, \mathcal{S}, \{\mathcal{A}_i\}_{i=1}^n, \{\mathcal{O}_i\}_{i=1}^n, \mathbb{T}, \mathbb{O}, \mu_1, \{r_i\}_{i=1}^n)$, where $H$ denotes the length of each episode, $\mathcal{S}$ is the state space with $|\mathcal{S}| = S$, $\mathcal{A}_i$ denotes the action space for agent $i$ with $|\mathcal{A}_i| = A_i$. We denote by $a := (a_1, \cdots, a_n)$ the joint action of all the $n$ agents, and by $\mathcal{A} = \mathcal{A}_1 \times \cdots \times \mathcal{A}_n$ the joint action space with $|\mathcal{A}| = A = \prod_{i=1}^n A_i$. We use $\mathbb{T} = \{\mathbb{T}_h\}_{h \in [H]}$ to denote the collection of transition matrices, so that $\mathbb{T}_h(\cdot \mid s, a) \in \Delta(\mathcal{S})$ gives the probability of the next state if joint action $a \in \mathcal{A}$ is taken at state $s \in \mathcal{S}$ and step $h$. In the following discussions, for any given $a$, we treat $\mathbb{T}_h(a) \in \mathbb{R}^{|\mathcal{S}| \times |\mathcal{S}|}$ as a matrix, where each row gives the probability for the next state from different current states. We use $\mu_1$ to denote the distribution of the initial state $s_1$, and $\mathcal{O}_i$ to denote the observation space for agent $i$ with $|\mathcal{O}_i| = O_i$. We denote by $o := (o_1, \ldots, o_n)$ the joint observation of all $n$ agents, and by $\mathcal{O} := \mathcal{O}_1 \times \cdots \times \mathcal{O}_n$ with $|\mathcal{O}| = O = \prod_{i=1}^n O_i$. We use $\mathbb{O} = \{\mathbb{O}_h\}_{h \in [H]}$ to denote the collection of the joint emission matrices, so that $\mathbb{O}_h(\cdot \mid s) \in \Delta(\mathcal{O})$ gives the emission distribution over the joint observation space $\mathcal{O}$ at state $s$ and step $h$. For notational convenience, we will at times adopt the matrix convention, where $\mathbb{O}_h$ is a matrix with rows $\mathbb{O}_h(\cdot \mid s_h)$. We also denote

$\mathbb{O}_{i,h}(\cdot \mid s) \in \Delta(\mathcal{O}_i)$ as the marginalized emission for agent $i$ agent. Finally, $r_i = \{r_{i,h}\}_{h \in [H]}$ is a collection of reward functions, so that $r_{i,h}(s_h, a_h)$ is the reward of agent $i$ agent given the state $s_h$ and joint action $a_h$ at step $h$.

Similar to a POMDP, in a POSG, the states are not observable to the agents, and each agent can only access its own individual observations. The game proceeds as follows. At the beginning of each episode, the environment samples $s_1$ from $\mu_1$. At each step $h$, each agent $i$ observes its own observation $o_{i,h}$, where $o_h := (o_{1,h}, \ldots, o_{n,h})$ is sampled jointly from $\mathbb{O}_h(\cdot \mid s_h)$. Then each agent $i$ takes the action $a_{i,h}$ and receives the reward $r_{i,h}(s_h, a_h)$. After that the environment transitions to the next state $s_{h+1} \sim \mathbb{T}_h(\cdot \mid s_h, a_h)$. The current episode terminates once $s_{H+1}$ is reached.

**Information sharing, common and private information.** Each agent $i$ in the POSG maintains its own information, $\tau_{i,h}$, a collection of historical observations and actions at step $h$, namely, $\tau_{i,h} \subseteq \{o_1, a_1, o_2, \cdots, a_{h-1}, o_h\}$, and the collection of such history at step $h$ is denoted by $\mathcal{T}_{i,h}$.

In many practical examples, agents may share part of the history with each other, which may introduce some *information structures* of the game that may lead to both sample and computation efficiencies. The information sharing splits the history into *common/shared* and *private* information for each agent. The *common information* at step $h$ is a subset of the joint history $\tau_h = (\tau_{i,h})_{i \in [n]}$: $c_h \subseteq \{o_1, a_1, o_2, \cdots, a_{h-1}, o_h\}$, which is available to *all the agents* in the system, and the collection of the common information is denoted as $\mathcal{C}_h$ and we define $C_h = |\mathcal{C}_h|$. Given the common information $c_h$, each agent also has her private information $p_{i,h} = \tau_{i,h} \setminus c_h$, where the collection of the private information for agent $i$ is denoted as $\mathcal{P}_{i,h}$ and its cardinality as $P_{i,h}$. The cardinality of the joint private information is $P_h = \prod_{i=1}^n P_{i,h}$. We allow $c_h$ or $p_{i,h}$ to take the special value $\emptyset$ when there is *no* common or private information. In particular, when $\mathcal{C}_h = \{\emptyset\}$, the problem reduces to the general POSG without any favorable information structure; when $\mathcal{P}_{i,h} = \{\emptyset\}$, every agent holds the same history, and it reduces to a POMDP when the agents share a common reward function, where the goal is usually to find the team-optimal solution.

**Policies and value functions.** We define a stochastic policy for agent $i$ at step $h$ as:

$$\pi_{i,h} : \Omega_h \times \mathcal{P}_{i,h} \times \mathcal{C}_h \to \Delta(\mathcal{A}_i), \tag{C.1}$$

where $\Omega_h$ is the random seed space, which is shared among agents and $\omega_{i,h} \in \Omega_h$ is the random seed for agent $i$. The corresponding policy class is denoted as $\Pi_{i,h}$. Hereafter, unless otherwise noted, when referring to *policies*, we mean the policies given in the form of (C.1), which maps the available information of agent $i$, i.e., the private information together with the common information, to the distribution over her actions.

We define $\pi_i$ as a sequence of policies for agent $i$ at all steps $h \in [H]$, i.e., $\pi_i = (\pi_{i,1}, \cdots, \pi_{i,H})$. We further denote $\Pi_i = \times_{h \in [H]} \Pi_{i,h}$ as the policy space for agent $i$ and $\Pi = \prod_{i \in [n]} \Pi_i$ as the joint policy space. As a special case, we define the space of *deterministic* policy as $\widetilde{\Pi}_i$, where $\widetilde{\pi}_i \in \widetilde{\Pi}_i$ maps the private information and common information to a *deterministic* action for agent $i$ and the joint space as $\widetilde{\Pi} = \prod_{i \in [n]} \widetilde{\Pi}_i$.

A *product* policy is denoted as $\pi = \pi_1 \times \pi_2 \cdots \times \pi_n \in \Pi$ if the distributions of drawing each seed $\omega_{i,h}$ for different agents are independent, and a (potentially correlated) joint policy is denoted as $\pi = \pi_1 \odot \pi_2 \cdots \odot \pi_n \in \Pi$.

We are now ready to define the *value function* conditioned on the common information under our model of POSG with information sharing:

**Definition C.2** (Value function with information sharing). For each agent $i \in [n]$ and step $h \in [H]$, given common information $c_h$ and joint policy $\pi = (\pi_i)_{i=1}^n \in \Pi$, the *value function conditioned on the common information* of agent $i$ is defined as: $V_{i,h}^{\pi,\mathcal{G}}(c_h) := \mathbb{E}_\pi^{\mathcal{G}} \left[ \sum_{h'=h}^H r_{i,h'}(s_{h'}, a_{h'}) \,\middle|\, c_h \right]$, where the expectation is taken over the randomness from the model $\mathcal{G}$, policy $\pi$, and the random seeds. For any $c_{H+1} \in \mathcal{C}_{H+1} : V_{i,H+1}^{\pi,\mathcal{G}}(c_{H+1}) := 0$. From now on, we will refer to it as *value function* for short.

Another key concept in our analysis is the belief about the state *and* the private information conditioned on the common information among agents. Formally, at step $h$, given policies from 1 to $h-1$,

we consider the common-information-based conditional belief $\mathbb{P}_h^{\pi_{1:h-1},\mathcal{G}}(s_h, p_h \mid c_h)$. This belief not only infers the current underlying state $s_h$, but also all agents' private information $p_h$. With the common-information-based conditional belief, the value function given in Definition C.2 has the following recursive structure:

$$V_{i,h}^{\pi,\mathcal{G}}(c_h) = \mathbb{E}_\pi^{\mathcal{G}}[r_{i,h}(s_h, a_h) + V_{i,h+1}^{\pi,\mathcal{G}}(c_{h+1}) \mid c_h], \tag{C.2}$$

where the expectation is taken over the randomness of $(s_h, p_h, a_h, o_{h+1})$. With this relationship, we can define the *prescription-value* function correspondingly, a generalization of the *action-value* function in (fully observable) stochastic games and MDPs to POSGs with information sharing, as follows.

**Definition C.3** (Prescription-value function with information sharing). At step $h$, given the common information $c_h$, joint policies $\pi = (\pi_i)_{i=1}^n \in \Pi$, and prescriptions $(\gamma_{i,h})_{i=1}^n \in \Gamma_h$, the *prescription-value function conditioned on the common information and joint prescription* of agent $i$ is defined as:

$$Q_{i,h}^{\pi,\mathcal{G}}(c_h, (\gamma_{j,h})_{j\in[n]}) := \mathbb{E}_\pi^{\mathcal{G}}\big[r_{i,h}(s_h, a_h) + V_{i,h+1}^{\pi,\mathcal{G}}(c_{h+1}) \,\big|\, c_h, (\gamma_{j,h})_{j\in[n]}\big],$$

where prescription $\gamma_{i,h} \in \Delta(\mathcal{A}_i)^{P_{i,h}}$ replaces the partial function $\pi_{i,h}(\cdot \mid \omega_{i,h}, c_h, \cdot)$ in the value function. From now on, we will refer to it as *prescription-value function* for short. With such a prescription-value function, agents can take actions purely based on their local private information [63, 62, 51].

This prescription-value function indicates the expected return for agent $i$ when all the agents firstly adopt the prescriptions $(\gamma_{j,h})_{j\in[n]}$ and then follow the policy $\pi$.

**Equilibrium notions.** With the definition of value functions, we can accordingly define the solution concepts. Here we define the notions of $\epsilon$-NE, $\epsilon$-CCE, $\epsilon$-CE, and $\epsilon$-team optimum under the information-sharing framework as follows. For a joint policy $(\pi_i)_{i=1}^n \in \Pi$ we denote the expected reward of agent $i$ by $v_i^{\mathcal{G}}(\pi) = \mathbb{E}_\pi^{\mathcal{P}}[\sum_{h=1}^H r_{i,h}(s_h, a_h)]$.

**Definition C.4** ($\epsilon$-approximate Nash equilibrium with information sharing). For any $\epsilon \geq 0$, a product policy $\pi^\star \in \Pi$ is an $\epsilon$-approximate Nash equilibrium of the POSG $\mathcal{G}$ with information sharing if:

$$\text{NE-gap}(\pi^\star) := \max_i \left( \max_{\pi_i' \in \Pi_i} v_i^{\mathcal{G}}(\pi_i' \times \pi_{-i}^\star) - v_i^{\mathcal{G}}(\pi^\star) \right) \leq \epsilon.$$

**Definition C.5** ($\epsilon$-approximate coarse correlated equilibrium with information sharing). For any $\epsilon \geq 0$, a joint policy $\pi^\star \in \Pi$ is an $\epsilon$-approximate coarse correlated equilibrium of the POSG $\mathcal{G}$ with information sharing if:

$$\text{CCE-gap}(\pi^\star) := \max_i \left( \max_{\pi_i' \in \Pi_i} v_i^{\mathcal{G}}(\pi_i' \times \pi_{-i}^\star) - v_i^{\mathcal{G}}(\pi^\star) \right) \leq \epsilon.$$

**Definition C.6** ($\epsilon$-approximate correlated equilibrium with information sharing). For any $\epsilon \geq 0$, a joint policy $\pi^\star \in \Pi$ is an $\epsilon$-approximate correlated equilibrium of the POSG $\mathcal{G}$ with information sharing if:

$$\text{CE-gap}(\pi^\star) := \max_i \left( \max_{\phi_i} v_i^{\mathcal{G}}((m_i \diamond \pi_i^\star) \odot \pi_{-i}^\star) - v_i^{\mathcal{G}}(\pi^\star) \right) \leq \epsilon,$$

where $m_i$ is called *strategy modification* for agent $i$, and $m_i = \{m_{i,h,c_h,p_{i,h}}\}_{h,c_h,p_{i,h}}$, with each $m_{i,h,c_h,p_{i,h}} : \mathcal{A}_i \to \mathcal{A}_i$ being a mapping from the action set to itself. The space of $m_i$ is denoted as $\mathcal{M}_i$. The composition $m_i \diamond \pi_i$ is defined as follows: at step $h$, when agent $i$ is given $c_h$ and $p_{i,h}$, the joint action chosen to be $(a_{1,h}, \cdots, a_{i,h}, \cdots, a_{n,h})$ will be modified to $(a_{1,h}, \cdots, m_{i,h,c_h,p_{i,h}}(a_{i,h}), \cdots, a_{n,h})$. Note that this definition follows from that in [78, 50, 38, 49, 51] when there exists certain common information, and is a natural generalization of the definition in the normal-form game case [71]. We denote by $\mathcal{M}_i^{\text{gen}}$ the space of all possible strategy modifications $m_i$ if it conditions on any history information instead of only $(c_h, p_{i,h})$. Similarly, we use $\mathcal{M}_{\mathcal{S},i}$ to denote the space of all possible strategy modifications $m_i$ if it only conditions on the current state e.g., a modification $m_{i,h,s} : \mathcal{A}_i \to \mathcal{A}_i$.

### C.2.1 Evolution of the Common and Private Information

**Assumption C.7** (Evolution of common and private information)**.** We assume that common information and private information evolve over time as follows:

- Common information $c_h$ is non-decreasing over time, that is, $c_h \subseteq c_{h+1}$ for all $h$. Let $\varpi_{h+1} = c_{h+1} \setminus c_h$. Thus, $c_{h+1} = \{c_h, \varpi_{h+1}\}$. Further, we have

$$\varpi_{h+1} = \chi_{h+1}(p_h, a_h, o_{h+1}), \tag{C.3}$$

  where $\chi_{h+1}$ is a fixed transformation. We use $\Upsilon_{h+1}$ to denote the collection of $\varpi_{h+1}$ at step $h$.

- Private information evolves according to:

$$p_{i,h+1} = \xi_{i,h+1}(p_{i,h}, a_{i,h}, o_{i,h+1}), \tag{C.4}$$

  where $\xi_{i,h+1}$ is a fixed transformation.

Equation (C.3) states that the increment in the common information depends on the "new" information $(a_h, o_{h+1})$ generated between steps $h$ and $h+1$ and part of the old information $p_h$. The incremental common information can be obtained by certain sharing and communication protocols among the agents. Equation (C.4) implies that the evolution of private information only depends on the newly generated private information $a_{i,h}$ and $o_{i,h+1}$. These evolution rules are standard in the literature [62, 63], specifying the source of common information and private information. Based on such evolution rules, we define $\{f_h\}_{h\in[H]}$ and $\{g_h\}_{h\in[H]}$, where $f_h : \mathcal{A}^h \times \mathcal{O}^h \to \mathcal{C}_h$ and $g_h : \mathcal{A}^h \times \mathcal{O}^h \to \mathcal{P}_h$ for $h \in [H]$, as the mappings that map the joint history to common information and joint private information, respectively.

### C.3 Strategy Independence of Belief and Examples

To solve a POSG without computationally intractable oracles, certain information-sharing is needed even under the observability assumption [51]. We thus make the following assumption as in [51].

**Assumption C.8** (Strategy independence of beliefs [62, 31, 51])**.** Consider any step $h \in [H]$, any policy $\pi \in \Pi$, and any realization of common information $c_h$ that has a non-zero probability under the trajectories generated by $\pi_{1:h-1}$. Consider any other policies $\pi'_{1:h-1}$, which also give a non-zero probability to $c_h$. Then, we assume that: for any such $c_h \in \mathcal{C}_h$, and any $p_h \in \mathcal{P}_h, s_h \in \mathcal{S}$,
$$\mathbb{P}_h^{\pi_{1:h-1}, \mathcal{G}}(s_h, p_h \,|\, c_h) = \mathbb{P}_h^{\pi'_{1:h-1}, \mathcal{G}}(s_h, p_h \,|\, c_h).$$

We provide examples satisfying this assumption in Appendix C.2, which include the fully-sharing structure as in [27, 69] as a special case. Finally, we also assume that common information and private information evolve over time properly in Assumption C.7, as standard in [62, 63, 51], which covers the models considered in [27, 69, 49].

Here we take the examples from [52, 51] to illustrate the generality of the information-sharing framework.

**Example C.9** (One-step delayed sharing)**.** At any step $h \in [H]$, the common and private information are given as $c_h = \{o_{2:h-1}, a_{1:h-1}\}$ and $p_{i,h} = \{o_{i,h}\}$, respectively. In other words, the players share all the action-observation history until the previous step $h-1$, with only the new observation being the private information. This model has been shown useful for power control [2].

**Example C.10** (State controlled by one controller with asymmetric delay sharing)**.** We assume there are 2 players for convenience. It extends naturally to $n$-player settings. Consider the case where the state dynamics are controlled by player 1, i.e., $\mathbb{T}_h(\cdot \,|\, s_h, a_{1,h}, a_{2,h}) = \mathbb{T}_h(\cdot \,|\, s_h, a_{1,h}, a'_{2,h})$ for all $(s_h, a_{1,h}, a_{2,h}, a'_{2,h}, h)$. There are two kinds of delay-sharing structures we could consider: **Case A:** the information structure is given as $c_h = \{o_{1,2:h}, o_{2,2:h-d}, a_{1,1:h-1}\}$, $p_{1,h} = \emptyset$, $p_{2,h} = \{o_{2,h-d+1:h}\}$, i.e., player 1's observations are available to player 2 instantly, while player 2's observations are available to player 1 with a delay of $d \geq 1$ time steps. **Case B:** similar to **Case A** but player 1's observation is available to player 2 with a delay of 1 step. The information structure is given as $c_h = \{o_{1,2:h-1}, o_{2,2:h-d}, a_{1,1:h-1}\}$, $p_{1,h} = \{o_{1,h}\}$, $p_{2,h} = \{o_{2,h-d+1:h}\}$, where $d \geq 1$. This kind of asymmetric sharing is common in network routing [67], where packages arrive at different hosts with different delays, leading to asymmetric delay sharing among hosts.

**Example C.11** (Symmetric information game). Consider the case when all observations and actions are available for all the agents, and there is no private information. Essentially, we have $c_h = \{o_{2:h}, a_{1:h-1}\}$ and $p_{i,h} = \emptyset$. We will also denote this structure as *fully sharing* hereafter.

**Example C.12** (Information sharing with one-directional-one-step delay). Similar to the previous cases, we also assume there are 2 players for ease of exposition, and the case can be generalized to multi-player cases straightforwardly. Similar to the one-step delay case, we consider the situation where all observations of player 1 are available to player 2, while the observations of player 2 are available to player 1 with one-step delay. All the past actions are available to both players. That is, in this case, $c_h = \{o_{1,2:h}, o_{2,2:h-1}, a_{1:h-1}\}$, and player 1 has no private information, i.e., $p_{1,h} = \emptyset$, and player 2 has private information $p_{2,h} = \{o_{2,h}\}$.

**Example C.13** (Uncontrolled state process). Consider the case where the state transition does not depend on the actions, that is, $\mathbb{T}_h(\cdot \mid s_h, a_h) = \mathbb{T}_h(\cdot \mid s_h, a'_h)$ for any $s_h, a_h, a'_h, h$. Note that the agents are still coupled through the joint reward. An example of this case is the information structure where controllers share their observations with a delay of $d \geq 1$ time steps. In this case, the common information is $c_h = \{o_{2:h-d}\}$ and the private information is $p_{i,h} = \{o_{i,h-d+1:h}\}$. Such information structures can be used to model repeated games with incomplete information [4].

# D  Collection of Algorithms

---

**Algorithm 1** Learning Decoding Function with Privileged Information

---

**Require:**
- POMDP $\mathcal{P}$,
- Expert policy $\pi^E \in \Pi_{\mathcal{S}}$,
- Number of sampled episodes per step $M$.

**Ensure:** A decoding function for each step $\{g_h\}_{h \in [H]}$ (see Theorem 4.6)

  For the $h = 1$ step: Collect $M$ state-observation pairs from the first-step $\widehat{D}_1 = \left\{ \left( s_1^{(i)}, o_1^{(i)} \right) \right\}_{i \in [M]}$

  on POMDP $\mathcal{P}$ and define the decoding function $g_1$ for the first step as:

$$g_1(o_1) = \left\{ s_1 : (s_1, o_1) \in \widehat{D}_1 \right\}$$

**for** each step $h \in [2, H]$ **do**

  Collect $M$ episodes $\left\{ \left( s_{1:H+1}^{(i)}, o_{1:H}^{(i)}, a_{1:H}^{(i)} \right) \right\}_{i \in [M]}$ on POMDP $\mathcal{P}$ using policy $\pi^E$ and let:

$$\widehat{D}_h := \left\{ \left( s_{h-1}^{(i)}, a_{h-1}^{(i)}, o_h^{(i)}, s_h^{(i)} \right) \right\}_{i \in [M]}$$

  Define the decoding function $g_h$ for step $h$ as:

$$g_h(s_{h-1}, a_{h-1}, o_h) = \left\{ s_h : (s_{h-1}, a_{h-1}, o_h, s_h) \in \widehat{D}_h \right\}$$

**end for**
**return** $\{g_h\}_{h \in [H]}$

---

---

**Algorithm 2** Belief-Weighted Optimistic Asymmetric Actor-Critic with Privileged Information

---

**Require:**
- POMDP $\mathcal{P}$,
- Subroutine $T_Q$ that given policy $\pi \in \Pi^L$, outputs $\{\widetilde{Q}\}_{h \in [H]}$ that approximates $\{Q_h^{\pi, \mathcal{P}}\}_{h \in [H]}$,
- Subroutine $T_{\boldsymbol{b}}$ that outputs $\{\boldsymbol{b}_h^{\mathrm{apx}}\}_{h \in [H]}$ that approximate $\{\boldsymbol{b}_h\}_{h \in [H]}$,
- Initial finite-memory policy $\pi^0 = \{\pi_h^0\}_{h \in [H]} \in \Pi^L$ for the POMDP $\mathcal{P}$, step-size $\eta$, and number of iterations $T$.

**Ensure:** A near-optimal policy

$\{\boldsymbol{b}_h^{\mathrm{apx}}\}_{h \in [H]} \leftarrow T_{\boldsymbol{b}}(\mathcal{P})$
**for** Iterations $t = 1 \ldots, T$ **do**
$\quad \{\widetilde{Q}_h^{t-1}\}_{h \in [H]} \leftarrow T_Q(\mathcal{P}, \pi^{t-1})$
$\quad$ Update the policy for each $a_h \in \mathcal{A}$, $z_h \in \mathcal{Z}_h$ as

$$\pi_h^t(a_h \mid z_h) \propto \pi_h^{t-1}(a_h \mid z_h) \cdot \exp\left(\eta \mathbb{E}_{s_h \sim \boldsymbol{b}_h^{\mathrm{apx}}(z_h)}\left[\widetilde{Q}_h^{t-1}(z_h, s_h, a_h)\right]\right)$$

$\quad$ Denote $\pi^t = \pi_{1:H}^t$
**end for**
**return** A policy uniform at random sampled from set $\{\pi^t\}_{t \in [T]}$

---

**Algorithm 3** Optimistic $Q$-function Estimation with Privileged Information

---

**Require:**
- POMDP $\mathcal{P}$, policy $\pi_{1:H} \in \Pi^L$ such that $\pi_h : \mathcal{S} \to \Delta(\mathcal{A})$,
- Number of episodes $M$ per step.

**Ensure:** Approximate $Q$-functions $\{\widetilde{Q}_h\}_{h \in [H]}$ (see Lemma H.3)
**Initialize:** $\forall z_{H+1} \in \mathcal{Z}_{H+1}, s_{H+1} \in \mathcal{S}, a_{H+1} \in \mathcal{A}$

$$\widetilde{Q}_{H+1}(z_{H+1}, s_{H+1}, a_{H+1}) \leftarrow 0, \qquad \pi_{H+1}(a_{H+1} \mid z_{H+1}) \leftarrow \frac{1}{A}$$

**for** step $h = H, \ldots, 1$ **do**
$\quad$ Collect $M$ trajectories using policy $\pi$ and let $D_h = \{\overline{\tau}^{(i)}\}_{i \in [M]}$ be the collected trajectories.
$\quad$ Compute empirical counts and define empirical distributions:

$$\widehat{\mathbb{T}}_h(s_{h+1} \mid s_h, a_h) = \frac{|\{\overline{\tau} \in D_h : (s_h', a_h', s_{h+1}') = (s_h, a_h, s_{h+1})\}|}{|\{\overline{\tau} \in D_h : (s_h', a_h') = (s_h, a_h)\}|}$$

$$\widehat{\mathbb{O}}_h(o_h \mid s_h) = \frac{|\{\overline{\tau} \in D_h : (s_h', o_h') = (s_h, o_h)\}|}{|\{\overline{\tau} \in D_h : s_h' = s_h\}|}$$

$\quad$ **for** each memory-state pair $(z_h, s_h) \in \mathcal{Z}_h \times \mathcal{S}$ **do**

$$\begin{aligned}
\widetilde{Q}(z_h, s_h, a_h) = \min\Bigg( &H - h + 1, \mathbb{E}_{\substack{s_{h+1} \sim \widehat{\mathbb{T}}_h(\cdot \mid s_h, a_h), \\ o_{h+1} \sim \widehat{\mathbb{O}}_{h+1}(\cdot \mid s_{h+1})}} [\widetilde{V}_{h+1}(z_{h+1}, s_{h+1})] \\
&+ r(s_h, a_h) + H \cdot \min\left(2, C \cdot \sqrt{\frac{S \log(1/\delta_1)}{\max(N_h(s_h, a_h), 1)}}\right) \\
&+ \mathbb{E}_{s_{h+1} \sim \widehat{\mathbb{T}}_h(\cdot \mid s_h, a_h)} H \cdot \min\left(2, C \cdot \sqrt{\frac{O \log(1/\delta_1)}{\max(N_{h+1}(s_{h+1}), 1)}}\right)\Bigg),
\end{aligned}$$

$\qquad$ where $\widetilde{V}_{h+1}(z_{h+1}, s_{h+1}) = \mathbb{E}_{a_{h+1} \sim \pi_{h+1}(\cdot \mid z_{h+1})}[\widetilde{Q}_{h+1}(z_{h+1}, s_{h+1}, a_{h+1})]$
$\quad$ **end for**
**end for**
**return** $\{\widetilde{Q}_h\}_{h \in [H]}$

---

---

**Algorithm 4** Approximate Belief Learning with Privileged Information via Model Truncation

---

**Require:**
- POMDP $\mathcal{P} = (H, \mathcal{S}, \mathcal{A}, \mathcal{O}, \{\mathbb{T}_h\}_{h\in[H]}, \{\mathbb{O}_h\}_{h\in[H]}, \mu_1, \{r_h\}_{h\in[H]})$,
- An MDP learning oracle MDP_Learning that efficiently learns an approximate optimal policy of an MDP,
- Number of trajectories $N$,
- The threshold $\epsilon$.

**Ensure:** Approximate belief $\{b_h^{\text{apx}}\}_{h\in[H]}$ (See Theorem H.5)

  **for** $h \in [H], s_h \in \mathcal{S}$ **do**

    $\widehat{r}_{h'}(s_h', a_h') \leftarrow \mathbb{1}[h' = h, s_h' = s_h]$ for any $(h', s_h', a_h') \in [H] \times \mathcal{S} \times \mathcal{A}$

    $\mathcal{M} \leftarrow (H, \mathcal{S}, \mathcal{A}, \{\mathbb{T}_h\}_{h\in[H]}, \mu_1, \{\widehat{r}_h\}_{h\in[H]})$ to be the MDP associated with $\mathcal{P}$

    $\Psi(h, s_h) \leftarrow \text{MDP\_Learning}(\mathcal{M})$

    Collect $N$ trajectories by executing policy $\Psi(h, s_h)$ for the first $h - 1$ steps then take action $a_h$ for each $a_h \in \mathcal{A}$ deterministically and denote the dataset $\{(s_h^i, o_h^i, a_h^i, s_{h+1}^i)\}_{i\in[NA]}$

    **for** $(o_h, a_h, s_{h+1}) \in \mathcal{O} \times \mathcal{A} \times \mathcal{S}$ **do**

      $N_h(s_h) \leftarrow \sum_{i\in[NA]} \mathbb{1}[s_h^i = s_h]$

      $N_h(s_h, a_h) \leftarrow \sum_{i\in[NA]} \mathbb{1}[s_h^i = s_h, a_h^i = a_h]$

      $N_h(s_h, a_h, s_{h+1}) \leftarrow \sum_{i\in[NA]} \mathbb{1}[s_h^i = s_h, a_h^i = a_h, s_{h+1}^i = s_{h+1}]$

      $N_h(s_h, o_h) \leftarrow \sum_{i\in[NA]} \mathbb{1}[s_h^i = s_h, o_h^i = o_h]$

      $\widehat{\mathbb{T}}_h(s_{h+1} \mid s_h, a_h) \leftarrow \frac{N_h(s_h, a_h, s_{h+1})}{N_h(s_h, a_h)}$

      $\widehat{\mathbb{O}}_h(o_h \mid s_h) \leftarrow \frac{N_h(s_h, o_h)}{N_h(s_h)}$

    **end for**

  **end for**

  **for** $h \in [H]$ **do**

    $\mathcal{S}_h^{\text{low}} \leftarrow \left\{ s_h \in \mathcal{S} \mid \frac{N_h(s_h)}{NA} \leq \epsilon \right\}$

    $\mathcal{S}_h^{\text{high}} \leftarrow \left\{ s_h \in \mathcal{S} \mid \frac{N_h(s_h)}{NA} > \epsilon \right\}$

  **end for**

  **for** $(h, s_h, o_h, a_h, s_{h+1}) \in [H] \times \mathcal{S}_h^{\text{high}} \times \mathcal{O} \times \mathcal{A} \times \mathcal{S}_h^{\text{high}}$ **do**

    $\widehat{\mathbb{T}}_h^{\text{trunc}}(s_{h+1} \mid s_h, a_h) \leftarrow \widehat{\mathbb{T}}_h(s_{h+1} \mid s_h, a_h) + \frac{\sum_{s_{h+1}' \in \mathcal{S}_{h+1}^{\text{low}}} \widehat{\mathbb{T}}_h(s_{h+1}' \mid s_h, a_h)}{|\mathcal{S}_{h+1}^{\text{high}}|}$

    $\widehat{\mathbb{O}}_h^{\text{trunc}}(o_h \mid s_h) \leftarrow \widehat{\mathbb{O}}_h(o_h \mid s_h)$

    $\widehat{\mu}_1^{\text{trunc}}(s_1) := \widehat{\mu}_1(s_1) + \frac{\sum_{s_1' \in \mathcal{S}_1^{\text{low}}} \widehat{\mu}_1(s_1')}{|\mathcal{S}_1^{\text{high}}|}, \forall s_1 \in \mathcal{S}_1^{\text{high}}$

  **end for**

  Let

$$\widehat{\mathcal{P}}^{\text{sub}} := (H, \{\mathcal{S}_h^{\text{high}}\}_{h\in[H]}, \mathcal{A}, \mathcal{O}, \{\widehat{\mathbb{T}}_h^{\text{trunc}}\}_{h\in[H]}, \{\widehat{\mathbb{O}}_h^{\text{trunc}}\}_{h\in[H]}, \widehat{\mu}_1^{\text{trunc}}, \{r_h\}_{h\in[H]})$$

  Define $\{\widehat{b}_h^{',\text{sub}} : \mathcal{Z}_h \to \Delta(\mathcal{S}_h^{\text{high}})\}_{h\in[H]}$ to be the approximate belief w.r.t. $\widehat{\mathcal{P}}^{\text{sub}}$

  Define $\{b_h^{\text{apx}} : \mathcal{Z}_h \to \Delta(\mathcal{S})\}_{h\in[H]}$ such that $b_h^{\text{apx}}(z_h)(s_h) = \widehat{b}_h^{',\text{sub}}(z_h)(s_h)$ for $s_h \in \mathcal{S}_h^{\text{high}}$ and 0 otherwise.

  **return** $\{b_h^{\text{apx}}\}_{h\in[H]}$

---

**Algorithm 5** Learning Multi-Agent (Individual) Decoding Functions with Privileged Information (NE/CCE Version)

---

**Require:**
- POSG $\mathcal{G} = (H, \mathcal{S}, \mathcal{A}, \mathcal{O}, \{\mathbb{T}_h\}_{h \in [H]}, \{\mathbb{O}_h\}_{h \in [H]}, \mu_1, \{r_i\}_{i \in [n]})$,
- $\pi \in \Pi_{\mathcal{S}}$, controller set $\{\mathcal{I}_h \subseteq [n]\}_{h \in [H]}$,
- Procedure MDP_Learning$(\cdot, \cdot)$ that takes as input an MDP and a reward function and returns an approximate optimal policy,
- Number of trajectories $N$.

**Ensure:** A decoding function for each agent and step $\{\widehat{g}_{j,h}\}_{j \in [n], h \in [H]}$ (see Theorem J.4)

    **for** $h \in [H]$, $s_h \in \mathcal{S}$ **do**

        **for** $i \in [n]$ **do**

           $\widehat{r}_{i,h'}(s'_h, a'_h) \leftarrow \mathbb{1}[h' = h, s'_h = s_h]$ for any $(h', s'_h, a'_h) \in [H] \times \mathcal{S} \times \mathcal{A}$

           Define the Markov game $\mathcal{M}$ associated with $\mathcal{G}$ as $\mathcal{M} = (H, S, A, \{\mathbb{T}_h\}_{h \in [H]}, \mu_1)$, where we omit the specification for the reward functions and one can specify them arbitrarily

           Define $\mathcal{M}(\pi_{-i})$ to be the MDP marginalized by $\pi_{-i}$

           $\Psi_i(h, s_h) \leftarrow$ MDP_Learning$(\mathcal{M}(\pi_{-i}), \widehat{r}_i)$

        **end for**

        For each $i \in [n]$, $a_h \in \mathcal{A}$, collect $N$ trajectories by executing policy $\Psi_i(h, s_h) \times \pi_{-i}$ for the first $h - 1$ steps then take action $a_h$ deterministically and denote the dataset $\{(s_h^{k,i}, o_h^{k,i}, a_h^{k,i}, s_{h+1}^{k,i})\}_{k \in [NA]}$

        **for** $(o_h, a_h, s_{h+1}) \in \mathcal{O} \times \mathcal{A} \times \mathcal{S}$ **do**

           $N_h(s_h) \leftarrow \sum_{k \in [N], i \in [n]} \mathbb{1}[s_h^{k,i} = s_h]$

           $N_h(s_h, a_{\mathcal{T}_h, h}, s_{h+1}) \leftarrow \sum_{k \in [N], i \in [n]} \mathbb{1}[s_h^{k,i} = s_h, a_{\mathcal{T}_h, h}^{k,i} = a_{\mathcal{T}_h, h}, s_{h+1}^{k,i} = s_{h+1}]$

           $N_h(s_h, o_h) \leftarrow \sum_{k \in [N], i \in [n]} \mathbb{1}[s_h^{k,i} = s_h, o_h^{k,i} = o_h]$

           $\widehat{\mathbb{T}}_h(s_{h+1} \mid s_h, a_{\mathcal{T}_h, h}) \leftarrow \frac{N_h(s_h, a_{\mathcal{T}_h, h}, s_{h+1})}{N_h(s_h, a_{\mathcal{T}_h, h})}$

           $\widehat{\mathbb{O}}_h(o_h \mid s_h) \leftarrow \frac{N_h(s_h, o_h)}{N_h(s_h)}$

        **end for**

    **end for**

    Define $\widehat{\mathcal{G}} := (H, \mathcal{S}, \mathcal{A}, \mathcal{O}, \{\widehat{\mathbb{T}}_h\}_{h \in [H]}, \{\widehat{\mathbb{O}}_h\}_{h \in [H]}, \mu_1, \{r_i\}_{i \in [n]})$

    Define $\widehat{g}_{j,h}(s_h \mid c_h, p_{j,h}) := \mathbb{P}^{\widehat{\mathcal{G}}}(s_h \mid c_h, p_{j,h})$ for each $j \in [n]$, $h \in [H]$, $c_h \in \mathcal{C}_h$, $p_{j,h} \in \mathcal{P}_{j,h}$

    **return** $\{\widehat{g}_{j,h}\}_{j \in [n], h \in [H]}$

---

---

**Algorithm 6** Learning Multi-Agent (Individual) Decoding Functions with Privileged Information (CE Version)

---

**Require:**
- POSG $\mathcal{G} = (H, \mathcal{S}, \mathcal{A}, \mathcal{O}, \{\mathbb{T}_h\}_{h \in [H]}, \{\mathbb{O}_h\}_{h \in [H]}, \mu_1, \{r_i\}_{i \in [n]})$,
- $\pi \in \Pi_{\mathcal{S}}$, controller set $\{\mathcal{I}_h \subseteq [n]\}$,
- Procedure MDP_Learning$(\cdot, \cdot)$ that takes as input an MDP and a reward function and returns an approximate optimal policy,
- Number of trajectories $N$.

**Ensure:** A decoding function for each agent and step $\{\widehat{g}_{j,h}\}_{j \in [n], h \in [H]}$ (see Theorem J.6)

  **for** $h \in [H]$, $s_h \in \mathcal{S}$ **do**

    **for** $i \in [n]$ **do**

      $\widehat{r}_{i,h'}(s'_h, a'_h) \leftarrow \mathbb{1}[h' = h, s'_h = s_h]$ for any $(h', s'_h, a'_h) \in [H] \times \mathcal{S} \times \mathcal{A}$.

      Define $\mathcal{M}_i^{\text{extended}}(\pi)$ to be the *extended* MDP, which is defined in Definition J.5.

      $\Psi_i(h, s_h) \leftarrow \text{MDP\_Learning}(\mathcal{M}_i^{\text{extended}}(\pi), \widehat{r}_i)$

    **end for**

    For each $i \in [n]$, $a_h \in \mathcal{A}$, collect $N$ trajectories by executing policy $\Psi_i(h, s_h) \times \pi_{-i}$ for the first $h - 1$ steps then take action $a_h$ deterministically and denote the dataset $\{(s_h^{k,i}, o_h^{k,i}, a_h^{k,i}, s_{h+1}^{k,i})\}_{k \in [NA]}$

    **for** $(o_h, a_h, s_{h+1}) \in \mathcal{O} \times \mathcal{A} \times \mathcal{S}$ **do**

      $N_h(s_h) \leftarrow \sum_{k \in [N], i \in [n]} \mathbb{1}[s_h^{k,i} = s_h]$

      $N_h(s_h, a_{\mathcal{T}_h, h}, s_{h+1}) \leftarrow \sum_{k \in [N], i \in [n]} \mathbb{1}[s_h^{k,i} = s_h, a_{\mathcal{T}_h, h}^{k,i} = a_{\mathcal{T}_h, h}, s_{h+1}^{k,i} = s_{h+1}]$

      $N_h(s_h, o_h) \leftarrow \sum_{k \in [N], i \in [n]} \mathbb{1}[s_h^{k,i} = s_h, o_h^{k,i} = o_h]$

      $\widehat{\mathbb{T}}_h(s_{h+1} \mid s_h, a_{\mathcal{T}_h, h}) \leftarrow \frac{N_h(s_h, a_{\mathcal{T}_h, h}, s_{h+1})}{N_h(s_h, a_{\mathcal{T}_h, h})}$

      $\widehat{\mathbb{O}}_h(o_h \mid s_h) \leftarrow \frac{N_h(s_h, o_h)}{N_h(s_h)}$

    **end for**

  **end for**

  Define $\widehat{\mathcal{G}} := (H, \mathcal{S}, \mathcal{A}, \mathcal{O}, \{\widehat{\mathbb{T}}_h\}_{h \in [H]}, \{\widehat{\mathbb{O}}_h\}_{h \in [H]}, \mu_1, \{r_i\}_{i \in [n]})$

  Define $\widehat{g}_{j,h}(s_h \mid c_h, p_{j,h}) := \mathbb{P}^{\widehat{\mathcal{G}}}(s_h \mid c_h, p_{j,h})$ for each $j \in [n]$, $h \in [H]$, $c_h \in \mathcal{C}_h$, $p_{j,h} \in \mathcal{P}_{j,h}$

  **return** $\{\widehat{g}_{j,h}\}_{j \in [n], h \in [H]}$

---

**Algorithm 7** Optimistic Common-Information-Based Value Iteration with Privileged Information

**Require:**
- POSG $\mathcal{G} = (H, \mathcal{S}, \mathcal{A}, \mathcal{O}, \{\mathbb{T}_h\}_{h \in [H]}, \{\mathbb{O}_h\}_{h \in [H]}, \mu_1, \{r_i\}_{i \in [n]})$,
- An approximate belief $\{\widehat{P}_h : \widehat{\mathcal{C}}_h \to \Delta(\mathcal{S} \times \mathcal{P}_h)\}_{h \in [H]}$,
- Number of iterations $K$.

**Ensure:** An approximate equilibrium policy

**Initialize:**

$$N_h^0(s_h, a_h) \leftarrow 0, \quad N_h^0(s_h, a_h, o_h) \leftarrow 0, \quad \widehat{\mathbb{J}}^0(o_{h+1} \mid s_h, a_h) \leftarrow \frac{1}{O}$$

**for** $k \in [K]$ **do**
    **for** $h \leftarrow H, H-1, \cdots, 1$ **do**
        **for** $\widehat{c}_h \in \widehat{\mathcal{C}}_h$ **do**

$Q_{i,h}^{\mathrm{high},k}(\widehat{c}_h, p_h, s_h, a_h)$
$\leftarrow \min\left\{ r_{i,h}(s_h, a_h) + b_h^{k-1}(s_h, a_h) + \mathbb{E}_{o_{h+1} \sim \widehat{\mathbb{J}}_h^{k-1}(\cdot \mid s_h, a_h)}\left[ V_{i,h+1}^{\mathrm{high},k}(\widehat{c}_{h+1}) \right], H - h + 1 \right\}$ for $i \in [n]$

$\quad Q_{i,h}^{\mathrm{low},k}(\widehat{c}_h, p_h, s_h, a_h)$
$\quad \leftarrow \max\left\{ r_{i,h}(s_h, a_h) - b_h^{k-1}(s_h, a_h) + \mathbb{E}_{o_{h+1} \sim \widehat{\mathbb{J}}_h^{k-1}(\cdot \mid s_h, a_h)}\left[ V_{i,h+1}^{\mathrm{low},k}(\widehat{c}_{h+1}) \right], 0 \right\}$ for $i \in [n]$

        Define

$Q_{i,h}^{\mathrm{high},k}(\widehat{c}_h, \gamma_h) := \mathbb{E}_{s_h, p_h \sim \widehat{P}_h(\cdot, \cdot \mid \widehat{c}_h)} \mathbb{E}_{\{a_{j,h} \sim \gamma_{j,h}(\cdot \mid p_{j,h})\}_{j \in [n]}}\left[ Q_{i,h}^{\mathrm{high},k}(\widehat{c}_h, p_h, s_h, a_h) \right]$ for $i \in [n]$

        Define

$Q_{i,h}^{\mathrm{low},k}(\widehat{c}_h, \gamma_h) := \mathbb{E}_{s_h, p_h \sim \widehat{P}_h(\cdot, \cdot \mid \widehat{c}_h)} \mathbb{E}_{\{a_{j,h} \sim \gamma_{j,h}(\cdot \mid p_{j,h})\}_{j \in [n]}}\left[ Q_{i,h}^{\mathrm{low},k}(\widehat{c}_h, p_h, s_h, a_h) \right]$ for $i \in [n]$

        $\{\pi_{j,h}^k(\cdot \mid \cdot, \widehat{c}_h, \cdot)\}_{j \in [n]} \leftarrow$ Bayesian-CE/CCE$(\{Q_{j,h}^{\mathrm{high},k}(\widehat{c}_h, \cdot)\}_{j \in [n]})$ (c.f. Appendix J.1)
        $V_{i,h}^{\mathrm{high},k}(\widehat{c}_h) \leftarrow \mathbb{E}_{\omega_h}\left[ Q_{i,h}^{\mathrm{high},k}(\widehat{c}_h, \{\pi_{j,h}^k(\cdot \mid \omega_{j,h}, \widehat{c}_h, \cdot)\}_{j \in [n]}) \right]$ for $i \in [n]$
        $V_{i,h}^{\mathrm{low},k}(\widehat{c}_h) \leftarrow \mathbb{E}_{\omega_h}\left[ Q_{i,h}^{\mathrm{low},k}(\widehat{c}_h, \{\pi_{j,h}^k(\cdot \mid \omega_{j,h}, \widehat{c}_h, \cdot)\}_{j \in [n]}) \right]$ for $i \in [n]$
        **end for**
    **end for**
    Execute $\pi^k$ and get trajectory $(s_{1:H}^k, a_{1:H}^k, o_{1:H+1}^k)$
    **for** $h \in [H], s_h \in \mathcal{S}, a_h \in \mathcal{A}, o_{h+1} \in \mathcal{O}$ **do**
        $N_h^k(s_h, a_h) \leftarrow \sum_{l \in [k]} \mathbb{1}[s_h^l = s_h, a_h^l = a_h]$
        $N_h^k(s_h, a_h, o_{h+1}) \leftarrow \sum_{l \in [k]} \mathbb{1}[s_h^l = s_h, a_h^l = a_h, o_{h+1}^l = o_{h+1}]$
        $\widehat{\mathbb{J}}_h^k(o_{h+1} \mid s_h, a_h) \leftarrow \frac{N_h^k(s_h, a_h, o_{h+1})}{N_h^k(s_h, a_h)}$
    **end for**
**end for**
$k^\star \leftarrow \arg\min_{k \in [K], i \in [n]} V_{i,1}^{\mathrm{high},k}(c_1^k) - V_{i,1}^{\mathrm{low},k}(c_1^k)$
**return** $\pi^{k^\star}$ for general-sum games or the marginalized policy of $\pi^{k^\star}$ for zero-sum games

**Algorithm 8** Approximate Belief Learning for MARL with Privileged Information

**Require:**
- POSG $\mathcal{G} = (H, \mathcal{S}, \mathcal{A}, \mathcal{O}, \{\mathbb{T}_h\}_{h \in [H]}, \{\mathbb{O}_h\}_{h \in [H]}, \mu_1, \{r_{i,h}\}_{i \in [n], h \in [H]})$,
- Controller set $\{\mathcal{I}_h \subseteq [n]\}_{h \in [H]}$,
- Procedure MDP_Learning$(\cdot, \cdot)$,
- Number of trajectories $N$,
- Accuracy $\epsilon$.

**Ensure:** An approximate belief

**for** $h \in [H], s_h \in \mathcal{S}$ **do**

$\widehat{r}_{h'}(s'_h, a'_h) \leftarrow \mathbb{1}[h' = h, s'_h = s_h]$ for any $(h', s'_h, a'_h) \in [H] \times \mathcal{S} \times \mathcal{A}$.

$\mathcal{M} \leftarrow (H, \mathcal{S}, \mathcal{A}, \{\mathbb{T}_h\}_{h \in [H]}, \mu_1)$ to be the MDP by ignoring the observation and emission of $\mathcal{G}$, where we omit the specification for the reward functions and one can specify them arbitrarily

$\Psi(h, s_h) \leftarrow$ MDP_Learning$(\mathcal{M}, \widehat{r})$

Collect $N$ trajectories by executing policy $\Psi(h, s_h)$ for the first $h - 1$ steps then take action $a_h$ for each $a_h \in \mathcal{A}$ deterministically and denote the dataset $\{(s^i_h, o^i_h, a^i_h, s^i_{h+1})\}_{i \in [NA]}$

**for** $(o_h, a_h, s_{h+1}) \in \mathcal{O} \times \mathcal{A} \times \mathcal{S}$ **do**

$N_h(s_h) \leftarrow \sum_{i \in [NA]} \mathbb{1}[s^i_h = s_h]$

$N_h(s_h, a_{\mathcal{T}_h, h}) \leftarrow \sum_{i \in [NA]} \mathbb{1}[s^i_h = s_h, a^i_{\mathcal{T}_h, h} = a_{\mathcal{T}_h, h}]$

$N_h(s_h, a_{\mathcal{T}_h, h}, s_{h+1}) \leftarrow \sum_{i \in [NA]} \mathbb{1}[s^i_h = s_h, a^i_{\mathcal{T}_h, h} = a_{\mathcal{T}_h, h}, s^i_{h+1} = s_{h+1}]$

$N_h(s_h, o_h) \leftarrow \sum_{i \in [NA]} \mathbb{1}[s^i_h = s_h, o^i_h = o_h]$

$\widehat{\mathbb{T}}_h(s_{h+1} \mid s_h, a_{\mathcal{T}_h, h}) \leftarrow \frac{N_h(s_h, a_h, s_{h+1})}{N_h(s_h, a_{\mathcal{T}_h, h})}$

$\widehat{\mathbb{O}}_h(o_h \mid s_h) \leftarrow \frac{N_h(s_h, o_h)}{N_h(s_h)}$

**end for**

**end for**

**for** $h \in [H]$ **do**

$\mathcal{S}^{\text{low}}_h \leftarrow \left\{ s_h \in \mathcal{S} \mid \frac{N_h(s_h)}{NA} \le \epsilon \right\}$

$\mathcal{S}^{\text{high}}_h \leftarrow \left\{ s_h \in \mathcal{S} \mid \frac{N_h(s_h)}{NA} > \epsilon \right\}$

**end for**

**for** $(h, s_h, o_h, a_h, s_{h+1}) \in [H] \times \mathcal{S}^{\text{high}}_h \times \mathcal{O} \times \mathcal{A} \times \mathcal{S}^{\text{high}}_h$ **do**

$\widehat{\mathbb{T}}^{\text{trunc}}_h(s_{h+1} \mid s_h, a_h) \leftarrow \widehat{\mathbb{T}}_h(s_{h+1} \mid s_h, a_h) + \frac{\sum_{s'_{h+1} \in \mathcal{S}^{\text{low}}_{h+1}} \widehat{\mathbb{T}}_h(s'_{h+1} \mid s_h, a_h)}{|\mathcal{S}^{\text{high}}_{h+1}|}$

$\widehat{\mathbb{O}}^{\text{trunc}}_h(o_h \mid s_h) \leftarrow \widehat{\mathbb{O}}_h(o_h \mid s_h)$

$\widehat{\mu}^{\text{trunc}}_1(s_1) := \widehat{\mu}_1(s_1) + \frac{\sum_{s'_1 \in \mathcal{S}^{\text{low}}_1} \widehat{\mu}_1(s'_1)}{|\mathcal{S}^{\text{high}}_1|}, \forall s_1 \in \mathcal{S}^{\text{high}}_1$

**end for**

Let

$$\widehat{\mathcal{G}}^{\text{sub}} := (H, \{\mathcal{S}^{\text{high}}_h\}_{h \in [H]}, \mathcal{A}, \mathcal{O}, \{\widehat{\mathbb{T}}^{\text{trunc}}_h\}_{h \in [H]}, \{\widehat{\mathbb{O}}^{\text{trunc}}_h\}_{h \in [H]}, \widehat{\mu}^{\text{trunc}}_1, \{r_{i,h}\}_{i \in [n], h \in [H]})$$

Define $\{\widetilde{P}_h : \widehat{\mathcal{C}}_h \to \Delta(\mathcal{S}^{\text{high}}_h \times \mathcal{P}_h)\}_{h \in [H]}$ to be the approximate belief w.r.t. $\widehat{\mathcal{G}}^{\text{sub}}$

Define $\{\widehat{P}_h : \widehat{\mathcal{C}}_h \to \Delta(\mathcal{S} \times \mathcal{P}_h)\}_{h \in [H]}$ such that $\widehat{P}_h(s_h, p_h \mid \widehat{c}_h) = \widetilde{P}_h(s_h, p_h \mid \widehat{c}_h)$ for $s_h \in \mathcal{S}^{\text{high}}_h$ and 0 otherwise

**return** $\{\widehat{P}_h\}_{h \in [H]}$

# E    Missing Details in Section 3

**Proof of Proposition 3.1:** We recall that $\boldsymbol{b}_h(\cdot)$ is the belief of the agent about the underlying state, see Appendix C for a detailed introduction. Note that Equation (3.1) can be written as

$$\widehat{\pi}^\star \in \arg\min_{\pi \in \Pi} \sum_{h=1}^{H} \mathbb{E}^{\mathcal{P}}_{\tau_h \sim \pi'} \mathbb{E}_{s_h \sim \boldsymbol{b}_h(\tau_h)} \left[ D_f(\pi_h^\star(\cdot \,|\, s_h) \,|\, \pi_h(\cdot \,|\, \tau_h)) \right].$$

Therefore, for any $h \in [H]$ and $\tau_h$ such that $\mathbb{P}^{\pi', \mathcal{P}}(\tau_h) > 0$, we can optimize $\pi$ separately for each $h \in [H]$ and $\tau_h$ as:

$$\widehat{\pi}_h^\star(\cdot \,|\, \tau_h) \in \underset{q \in \Delta(\mathcal{A})}{\arg\min} \ \mathbb{E}_{s_h \sim \boldsymbol{b}_h(\tau_h)} \left[ D_f(\pi_h^\star(\cdot \,|\, s_h) \,|\, q) \right].$$

Now we are ready to construct the counter-example of $\gamma$-observable POMDP $\mathcal{P}^\epsilon$ for some $\epsilon \in (0,1)$ with $H = 1$, $\mathcal{S} = \{s^1, s^2\}$, $\mathcal{A} = \{a^1, a^2\}$, and $\mathcal{O} = \{o^1, o^2\}$. We let $\mu_1 = (\frac{1-\gamma}{2-\gamma}, \frac{1}{2-\gamma})$, $\mathbb{O}_1(o^1 \,|\, s^1) = 1$, and $\mathbb{O}_1(o^1 \,|\, s^2) = 1 - \gamma$, $\mathbb{O}_1(o^2 \,|\, s^2) = \gamma$. Therefore, it is direct to see that $\mathbb{O}_1$ is exactly $\gamma$-observable. Most importantly, we choose $r_1(s^1, a^1) = 1$, $r_1(s^1, a^2) = 0$, and $r_1(s^2, a^1) = 0$, $r_1(s^2, a^2) = \epsilon$.

Therefore, given such a reward function, the fully observable expert policy is given by $\pi_1^\star(a^1 \,|\, s^1) = 1$ and $\pi_1^\star(a^2 \,|\, s^2) = 1$, i.e., choosing $a^1$ at state $s^1$ and $a^2$ at state $s^2$ deterministically. Meanwhile, by our construction, one can compute that the belief given observation $o^1$ ensures $\boldsymbol{b}_1(o^1) = \mathrm{Unif}(\mathcal{S})$. Hence, the corresponding "distilled" partially observable policy under observation $o^1$ is given by

$$\begin{aligned}
\widehat{\pi}_1^\star(\cdot \,|\, o^1) &= \underset{q \in \Delta(\mathcal{A})}{\arg\min} \ \mathbb{E}_{s_1 \sim \boldsymbol{b}_1(o^1)} \left[ D_f(\pi_1^\star(\cdot \,|\, s_1) \,|\, q) \right] \\
&= \underset{q \in \Delta(\mathcal{A})}{\arg\min} \ \frac{D_f(\pi_1^\star(\cdot \,|\, s^1) \,|\, q) + D_f(\pi_1^\star(\cdot \,|\, s^2) \,|\, q)}{2} \\
&= \underset{q \in \Delta(\mathcal{A})}{\arg\min} \ \frac{f(1/q(a^1))q(a^1) + f(0)q(a^2) + f(0)q(a^1) + f(1/q(a^2))q(a^2)}{2} \\
&= \underset{q \in \Delta(\mathcal{A})}{\arg\min} \ \frac{f(0) + f(1/q(a^1))q(a^1) + f(1/q(a^2))q(a^2)}{2},
\end{aligned}$$

where the last step is due to $q \in \Delta(\mathcal{A})$. Now consider the function $g(x) = x f(1/x)$ for $x > 0$. It is direct to compute that $g'(x) = f(1/x) - \frac{f'(1/x)}{x}$, and $g''(x) = \frac{f''(1/x)}{x^3} \geq 0$ due to the convexity of the function $f$. Thus, we conclude that $g$ is also convex. By Jensen's inequality, we have

$$\frac{f(1/q(a^1))q(a^1) + f(1/q(a^2))q(a^2)}{2} \geq f(2/(q(a^1) + q(a^2)))(q(a^1) + q(a^2))/2 = f(2)/2,$$

where the equality holds when $q(a^1) = q(a^2) = \frac{1}{2}$. This indicates that $\widehat{\pi}_1^\star(\cdot \,|\, o_1) = \mathrm{Unif}(\mathcal{A})$. On the other hand, combining the fact that $\boldsymbol{b}_1(o^1) = \mathrm{Unif}(\mathcal{S})$ with $\epsilon < 1$, it is direct to see that the optimal partially observable policy $\widetilde{\pi} \in \arg\max_{\pi \in \Pi} v^{\mathcal{P}}(\pi)$ satisfies $\widetilde{\pi}_1(a^1 \,|\, o^1) = 1$. Now we are ready to evaluate the optimality gap between $\widetilde{\pi}$ and $\widehat{\pi}^\star$ as follows

$$\begin{aligned}
v^{\mathcal{P}^\epsilon}(\widetilde{\pi}) - v^{\mathcal{P}^\epsilon}(\widehat{\pi}^\star) &= \mathbb{P}^{\mathcal{P}^\epsilon}(o^1)(V_1^{\widetilde{\pi}, \mathcal{P}^\epsilon}(o^1) - V_1^{\widehat{\pi}^\star, \mathcal{P}^\epsilon}(o^1)) + \mathbb{P}^{\mathcal{P}^\epsilon}(o^2)(V_1^{\widetilde{\pi}, \mathcal{P}^\epsilon}(o^2) - V_1^{\widehat{\pi}^\star, \mathcal{P}^\epsilon}(o^2)) \\
&\geq \mathbb{P}^{\mathcal{P}^\epsilon}(o^1)(V_1^{\widetilde{\pi}, \mathcal{P}^\epsilon}(o^1) - V_1^{\widehat{\pi}^\star, \mathcal{P}^\epsilon}(o^1)),
\end{aligned}$$

where the last step is due to the fact that $\widetilde{\pi}$ is the optimal policy, leading to the fact that $V_1^{\widetilde{\pi}, \mathcal{P}^\epsilon}(o^2) - V_1^{\widehat{\pi}^\star, \mathcal{P}^\epsilon}(o^2) \geq 0$. Now it is not hard to compute that

$$\mathbb{P}^{\mathcal{P}^\epsilon}(o^1) \geq 1 - \gamma.$$

Meanwhile, we can evaluate that

$$V_1^{\widetilde{\pi}, \mathcal{P}^\epsilon}(o^1) = \frac{1}{2}, \quad V_1^{\widehat{\pi}^\star, \mathcal{P}^\epsilon}(o^1) = \frac{1 + \epsilon}{4}$$

and correspondingly $V_1^{\widetilde{\pi}, \mathcal{P}^\epsilon}(o^1) - V_1^{\widehat{\pi}^\star, \mathcal{P}^\epsilon}(o^1) = \frac{1 - \epsilon}{4}$, implying that $v^{\mathcal{P}^\epsilon}(\widetilde{\pi}) - v^{\mathcal{P}^\epsilon}(\widehat{\pi}^\star) \geq \frac{(1 - \gamma)(1 - \epsilon)}{4}$. This concludes our proof. ∎

Note that another counterexample in a similar spirit was also constructed in [34], demonstrating that the expert policy for a poorly chosen agent-environment boundary can be useless in imitation learning, although the $\gamma$-observability property is not satisfied for the construction therein.

**Remark E.1.** The counter-example $\mathcal{P}^\epsilon$ constructed above can be also used to demonstrate the *bias* of the state-only-based value function as an estimate of the history-based value function that appeared in the policy gradient formula for POMDPs, in the finite-horizon setting, i.e. $\mathbb{E}_{s_h \sim \boldsymbol{b}_h(\tau_h)}[V_h^{\pi,\mathcal{P}^\epsilon}(s_h)] \neq V_h^{\pi,\mathcal{P}^\epsilon}(\tau_h)$ (mirroring Theorem 4.2 of [6]). Specifically, in the counter-example above, we consider the policy $\pi$ such that $\pi_1(a_1 \mid o_1) = 1$ and $\pi_1(a_2 \mid o_2) = 1$. The state-only-based value function can be evaluated as

$$V_1^{\pi,\mathcal{P}^\epsilon}(s_1) = \frac{1-\gamma}{2-\gamma}, \qquad V_1^{\pi,\mathcal{P}^\epsilon}(s_2) = \gamma\epsilon,$$

which implies that $\mathbb{E}_{s_1 \sim \boldsymbol{b}_1(o_1)}[V_1^{\pi,\mathcal{P}^\epsilon}(s_1)] = \frac{V_1^{\pi,\mathcal{P}^\epsilon}(s_1) + V_1^{\pi,\mathcal{P}^\epsilon}(s_2)}{2} = \frac{\frac{1-\gamma}{2-\gamma} + \gamma\epsilon}{2}$. On the other hand, it holds that $V_1^{\pi,\mathcal{P}^\epsilon}(o_1) = \frac{1+\epsilon}{2}$, which is not equal to $\mathbb{E}_{s_1 \sim \boldsymbol{b}_1(o_1)}[V_1^{\pi,\mathcal{P}^\epsilon}(s_1)]$, showing the bias of such a state-only value function.

**Example E.2** (Deterministic POMDP [36, 81]). We say a POMDP $\mathcal{P}$ is of deterministic transition if entries of matrices $\{\mathbb{T}_h\}_{h \in [H]}$ and the vector $\mu_1$ are either 0 or 1. Note that we do not make any assumptions on the emission matrices.

**Example E.3** (Block MDP [43, 18]). We say a POMDP $\mathcal{P}$ is a block MDP if for any $h \in [H]$, $s_h, s_h' \in \mathcal{S}$, it holds that $\mathrm{supp}(\mathbb{O}_h(\cdot \mid s_h)) \cap \mathrm{supp}(\mathbb{O}_h(\cdot \mid s_h')) = \emptyset$ when $s_h \neq s_h'$.

**Example E.4** ($k$-step decodable POMDP [19]). We say a POMDP $\mathcal{P}$ is a $k$-step decodable POMDP if there exists an unknown decoder $\phi^\star = \{\phi_h^\star : \mathcal{Z}_h \to \mathcal{S}\}_{h \in [H]}$ such that for any $h \in [H]$ and reachable trajectory $\tau_h$, $\mathbb{P}^{\mathcal{P}}(s_h = \phi_h^\star(z_h) \mid \tau_h) = 1$, where $\mathcal{Z}_h = (\mathcal{O} \times \mathcal{A})^{\max\{h-1,k-1\}} \times \mathcal{O}$, $z_h = ((o,a)_{k(h):h-1}, o_h)$, and $k(h) = \max\{h - k + 1, 1\}$.

Finally, to understand how our condition can extend beyond known examples in the literature, we show that one can indeed allow the decoding length of Example E.4 to be unknown and arbitrary (instead of being a small known constant as in [19] to have provably efficient algorithms).

**Example E.5** (POMDP with arbitrary, unknown decodable length). This example is similar to Example E.4, but the decoding length $m$ is unknown and not necessarily a small constant.

In light of the pitfall in Proposition 3.1, we will analyze *both* the computational and statistical efficiencies of expert distillation in Section 4, under the condition in Definition 3.2.

**Proof of Example E.2 & Example E.3 & Example E.4 & Example E.5:** To see why those examples follow our Definition 3.2, it is indeed an immediate result of Proposition 7.1. ∎

**Proof of Proposition 3.3:**

Here we evaluate the computational complexity and sample complexity of each iteration $t$ as follows.

**Sample complexity:** The algorithm executes the policy $\pi^{t-1}$ and collect $K$ episodes, denoted as $\{s_{1:H+1}^k, o_{1:H}^k, a_{1:H}^k\}_{k \in [K]}$. Thus, the sample complexity of each iteration is $\Theta(K)$.

**Computational complexity for policy evaluation:** The policy evaluation of the vanilla asymmetric actor-critic is done by minimizing the Bellman error. In the finite-horizon setting with tabular parameterization, it is equivalent to performing the following update for each $h \in [H]$ in a backward way and each $k \in [K]$:

$$Q_h^t(\tau_h^k, s_h^k, a_h^k) \leftarrow (1-\alpha)Q_h^{t-1}(\tau_h^k, s_h^k, a_h^k)$$
$$+ \alpha \left( r_h(s_h^k, a_h^k) + \frac{1}{|\mathcal{J}(\tau_h^k, s_h^k, a_h^k)|} \sum_{j \in \mathcal{J}(\tau_h^k, s_h^k, a_h^k)} Q_{h+1}^t(\tau_{h+1}^j, s_{h+1}^j, a_{h+1}^j) \right),$$

for some stepsize $\alpha \in (0,1)$, where $\mathcal{J}(\tau_h^k, s_h^k, a_h^k) := \{j \in [K] \mid (\tau_h^j, s_h^j, a_h^j) = (\tau_h^k, s_h^k, a_h^k)\}$. Therefore, the computational complexity for this procedure is of $\mathrm{POLY}(H, K)$.

**Computational complexity for policy improvement:** For tabular parameterization, computing $\nabla \log \pi_h^{t-1}(a_h^k \mid \tau_h^k)$ takes $\mathcal{O}(1)$ computation. Hence the policy update in Equation (3.2) performs $\textsc{poly}(H, K)$ computation.

Meanwhile, under the exponential time hypothesis, there is no polynomial time algorithm for even planning an $\epsilon$-approximate optimal policy in $\gamma$-observable POMDPs [26]. This implies that the vanilla asymmetric actor-critic needs to take super-polynomial time to find an approximately optimal policy. This implies the corresponding sample complexity has to be super-polynomial.

Finally, we remark that even if we let the policy and the $Q$-function not depend on the entire history $\tau_h$ but only the finite-memory $z_h$, the proof still holds. The key is that whenever one only *computes* at the *sampled* history/finite-memories, i.e., updates the policy in an *asynchronous* way (in contrast to the *synchronous* one where the policies at *all* histories/finite-memories are updated), the sample and computational complexities will be coupled with the same order per iteration, which implies a super-polynomial sample complexity due to the super-polynomial computational complexity. This completes the proof. ∎

**Derivation for the closed-form update Equation (3.3).** Note that the proximal policy optimization [72] update has the policy improvement as follows

$$\pi^t \leftarrow \arg\max_\pi \left\{ L^{t-1}(\pi) - \eta^{-1} \mathbb{E}_{\pi^{t-1}}^{\mathcal{P}} \left[ \sum_{h \in [H]} \mathrm{KL}(\pi_h(\cdot \mid \tau_h) \mid \pi_h^{t-1}(\cdot \mid \tau_h)) \right] \right\}, \tag{E.1}$$

where $\eta$ is some learning rate and $L^{t-1}(\pi)$ is a first-order approximation of the expected accumulated rewards at $\pi^{t-1}$:

$$L^{t-1}(\pi) := v^{\mathcal{P}}(\pi^{t-1}) + \mathbb{E}_{\pi^{t-1}}^{\mathcal{P}} \left[ \sum_{h \in [H]} \langle Q_h^{t-1}(\tau_h, s_h, \cdot), \pi_h(\cdot \mid \tau_h) - \pi_h^{t-1}(\cdot \mid \tau_h) \rangle \right].$$

By plugging $L^{t-1}(\pi)$ into Equation (E.1), with simple algebric manipulations, we prove that:

$$\pi_h^t(\cdot \mid \tau_h) \propto \pi_h^{t-1}(\cdot \mid \tau_h) \exp\left( \eta \mathbb{E}_{s_h \sim \boldsymbol{b}_h(\tau_h)} \left[ Q_h^{t-1}(\tau_h, s_h, \cdot) \right] \right).$$

# F    Missing Details in Section 4

**Proof of Lemma 4.4:** The proof follows by the assumption that the total cumulative reward is at most $H$,

$$v^{\mathcal{P}}(\pi) \geq \mathbb{E}_\pi^{\mathcal{P}} \left[ \left( \sum_{h \in [H]} r_h \right) \mathbb{1}[\forall h :\in [H] : g_h(s_{h-1}, a_{h-1}, o_h) = s_h] \right]$$

$$= \mathbb{E}_{\pi^E}^{\mathcal{P}} \left[ \left( \sum_{h \in [H]} r_h \right) \mathbb{1}[\forall h :\in [H] : g_h(s_{h-1}, a_{h-1}, o_h) = s_h] \right]$$

$$\quad + \mathbb{E}_{\pi^E}^{\mathcal{P}} \left[ \left( \sum_{h \in [H]} r_h \right) \mathbb{1}[\exists h :\in [H] : g_h(s_{h-1}, a_{h-1}, o_h) \neq s_h] \right]$$

$$\quad - \mathbb{E}_{\pi^E}^{\mathcal{P}} \left[ \left( \sum_{h \in [H]} r_h \right) \mathbb{1}[\exists h :\in [H] : g_h(s_{h-1}, a_{h-1}, o_h) \neq s_h] \right]$$

$$= \mathbb{E}_{\pi^E}^{\mathcal{P}} \left[ \left( \sum_{h \in [H]} r_h \right) \right] - \mathbb{E}_{\pi^E}^{\mathcal{P}} \left[ \left( \sum_{h \in [H]} r_h \right) \mathbb{1}[\exists h :\in [H] : g_h(s_{h-1}, a_{h-1}, o_h) \neq s_h] \right]$$

$$\geq v^{\mathcal{P}}(\pi^E) - H \mathbb{P}^{\pi^E, \mathcal{P}} \left[ \exists h :\in [H] : g_h(s_{h-1}, a_{h-1}, o_h) \neq s_h \right]$$

$$\geq v^{\mathcal{P}}(\pi^E) - H\epsilon,$$

which completes the proof. ∎

**Proof of Theorem 4.5:** For each step $h \in [H]$ we define $D_h$ to be the distribution over the underlying state $s_{h-1}$ at step $h-1$, taken action $a_{h-1} \in \mathcal{A}$ based on $\pi^E$, the underlying state transitioned to $s_h \in \mathcal{S}$, and the observation $o_h \sim \mathbb{O}_h(\cdot \,|\, s_h)$. We remind that we include at step zero, a dummy state-observation pair $(s_0, o_0)$, for notational convenience. Formally, $D_h$ is defined as

$$D_h(s_{h-1}, a_{h-1}, o_h, s_h) := \mathbb{P}^{\pi^E, \mathcal{P}}\left[s_{h-1}, a_{h-1}, o_h, s_h\right].$$

We first use union bound to decompose the probability that we incorrectly decode,

$$
\begin{aligned}
\mathbb{P}^{\pi^E, \mathcal{P}}\left[\exists h \in [H] : g_h\left(s_{h-1}, a_{h-1}, o_h\right) \neq s_h\right] &\leq \sum_{h \in [H]} \mathbb{P}^{\pi^E, \mathcal{P}}\left[g_h\left(s_{h-1}, a_{h-1}, o_h\right) \neq s_h\right] \\
&= \sum_{h \in [H]} \mathbb{P}_{(s_{h-1}, a_{h-1}, o_h, s_h) \sim D_h}\left[g_h\left(s_{h-1}, a_{h-1}, o_h\right) \neq s_h\right].
\end{aligned}
$$
$$\text{(F.1)}$$

For each $h \in [H]$, we can use $M$ episodes to collect $M$ samples from the distribution $D_h$. In addition, since state $s_{H+1}$ is dummy, we need not to collect episodes for $D_{H+1}$. Denote the set of collected samples by $\widehat{D}_h^M$. We define the decoding $g_h$ for step $h \in [H]$ as follows:

$$g_h(s_{h-1}, a_{h-1}, o_h) = \left\{s_h \mid (s_{h-1}, a_{h-1}, o_h, s_h) \in \widehat{D}_h^M\right\}.$$

Observe that by Definition 3.2, $\{s_h \mid (s_{h-1}, a_{h-1}, o_h, s_h) \in \widehat{D}_h^M\}$ is either the empty set or contains only a single elements, in which case, it is true that $g_h(s_{h-1}, a_{h-1}, o_h) = \psi_h(s_{h-1}, a_{h-1}, o_h)$ ($\psi$ is the real decoding function, see Definition 3.2). Moreover, we slightly abuse the notation and let $\widetilde{D}_h^M$ denote the empirical distribution induced by the samples in $\widehat{D}_h^M$. Thus, with probability at least $1 - \frac{\delta}{H}$ and setting $M = \Theta\left(\frac{A \cdot O \cdot S + \log(H/\delta)}{\epsilon^2}\right)$ for each step $h \in [H]$, we obtain the following by the result in [13]:

$$
\begin{aligned}
\mathbb{P}^{\pi^E, \mathcal{P}}\left[g_h\left(s_{h-1}, a_{h-1}, o_h\right) \neq s_h\right] &= \mathbb{P}^{\pi^E, \mathcal{P}}\left[g_h\left(s_{h-1}, a_{h-1}, o_h\right) = \emptyset\right] \\
&= \mathbb{P}_{(s_{h-1}, a_{h-1}, o_h, s_h) \sim D_h}\left[(s_{h-1}, a_{h-1}, o_h, s_h) \notin \widehat{D}_h^M\right] \\
&= \sum_{u \in \mathrm{supp}(D_h)} \mathbb{P}_{u' \sim D_h}[u = u'] \mathbb{1}[u \notin \mathrm{supp}(\widetilde{D}_h^M)] \\
&\leq d_{TV}(D_h, \widetilde{D}_h^M) \\
&\leq \epsilon.
\end{aligned}
$$

Thus, by union bound, with probability at least $1 - \delta$, we have that for each step $h \in [H]$,

$$\mathbb{P}_{(s_{h-1}, a_{h-1}, o_h, s_h) \sim D_h}\left[g_h\left(s_{h-1}, a_{h-1}, o_h\right) \neq s_h\right] \leq \epsilon,$$

which in combination with Equation (F.1) concludes the proof. Finally, we note that we used a total of $\Theta\left(H \cdot \frac{A \cdot O \cdot S + \log(H/\delta)}{\epsilon^2}\right)$ episodes from the POMDP, and the computational time was $\mathrm{POLY}(H, A, O, S, \frac{1}{\epsilon}, \log\left(\frac{1}{\delta}\right))$. ∎

# G   Provably Efficient Expert Policy Distillation with Function Approximation

We now turn our attention to the rich-observation setting under our deterministic filter condition. Definition 3.2 motivates us to consider only succinct policies that incorporate an auxiliary parameter representing the most recent state, as well as the most recent observations and actions. To handle the large observation space, we further assume that for each step $h \in [H]$, the agent selects a decoding function $g_h$ from a family of *multi-class classifiers* $\mathcal{F}_h \subset \{\mathcal{S} \times \mathcal{A} \times \mathcal{O} \to \mathcal{S}\}$. For the function class $\mathcal{F}_h$, we make the standard realizability assumption. We formally summarize our assumptions in Assumption G.1.

**Assumption G.1.** We consider a POMDP that satisfies Definition 3.2. In addition, to derive learning algorithms that do not dependent on $O$, for each step $h \in [H]$, we assume that we have access to a class of functions $\mathcal{F}_h : \mathcal{S} \times \mathcal{A} \times \mathcal{O} \to \mathcal{S}$ such that the perfect decoding function $\psi_h \in \mathcal{F}_h$.

We aim for our final bounds to depend on a complexity measure of the function class $\mathcal{F} = \{\mathcal{F}_h\}_{h \in [H]}$ rather than the cardinality of the observation space $\mathbb{O}$. We utilize the Daniely and Shalev-Shwartz-Dimension (DS Dimension) (Theorem G.2), which characterizes the PAC learnability for multi-class classification [8]. Defining the DS dimension is beyond the scope of our paper; we direct interested readers to [8] for further details. For intuition, readers can think of the DS Dimension as a certificate of PAC learnability without loss of intuition.

**Theorem G.2** (Theorem 1 in [8]). Consider a family of multi-class classifiers $\mathcal{F}$ that map features in space $x \in \mathcal{X}$ to labels in space $y \in \mathcal{Y}$. Moreover, assume there is a joint probability distribution $D$ over features in $\mathcal{X}$ and labels in $\mathcal{Y}$, and that there exists $g^* \in \mathcal{F}$ such that for each $(x, y) \in \mathrm{supp}(D)$, $g^*(x) = y$. Given $n$ samples from $D$, there exists an algorithm that with probability at least $1 - \delta$ outputs $\widetilde{g} \in \mathcal{F}$ such that

$$\mathbb{P}_{(x,y) \sim D}[\widetilde{g}(x) \neq y] \leq \widetilde{\mathcal{O}}\left(\frac{d_{DS}^{3/2}(\mathcal{F}) + \log\left(\frac{1}{\delta}\right)}{n}\right),$$

where $d_{DS}(\mathcal{F})$ denotes the Daniely and Shalev-Shwartz-Dimension of the function class $\mathcal{F}$.

We are now ready to present the main theorem of this section.

**Theorem G.3.** Consider a POMDP $\mathcal{P}$ that satisfies Definition 3.2, a policy $\pi^E \in \Pi_{\mathcal{S}}$, and let $\{\mathcal{F}_h \subseteq \{\mathcal{S} \times \mathcal{A} \times \mathcal{O} \to \mathcal{S}\}\}_{h \in [H]}$ be the decoding function class, and $\psi_h \in \mathcal{F}_h$ for each $h \in [H]$, i.e., $\{\mathcal{F}_h\}_{h \in [H]}$ is realizable. Then given access to the classification oracle of [9], there exists an algorithm learning the decoding function $\{g_h\}_{h \in [H]}$ such that with probability at least $1 - \delta$, for each step $h \in [H]$:

$$\mathbb{P}^{\pi^E, \mathcal{P}}\left[\exists h \in [H] : g_h(s_{h-1}, a_{h-1}, o_h) \neq s_h\right] \leq \epsilon,$$

using $\mathcal{O}\left(\frac{H^2\left(\max_{h \in [H]} d_{DS}^{3/2}(\mathcal{F}_h) + \log\left(\frac{1}{\delta}\right)\right)}{\epsilon}\right)$ episodes, where $d_{DS}(\mathcal{F}_h)$ is the Daniely and Shalev-Shwartz-Dimension of $\mathcal{F}_h$ [8].

Combining Theorem G.3 and Lemma 4.4, we obtain the final polynomial sample complexity in this function approximation setting, using classification (supervised learning) oracles (c.f. Table 1).

**Proof of Theorem G.3:**

For each step $h \in [H]$, we define $D_h$ to be the distribution over the underlying state $s_{h-1}$ at step $h-1$, taken action $a_{h-1} \in \mathcal{A}$ from $\pi^E$, the underlying state transitioned to $s_h \in \mathcal{S}$, and the hallucinated observation $o_h \sim \mathbb{O}_h(\cdot \mid s_h)$ (we remind readers that for step 0, we use dummy state $s_0$ and action $a_0$). Formally, the probability that the sequence $(s_{h-1}, a_{h-1}, o_h, s_h)$ is sampled from $D_h$ equals to

$$D_h(s_{h-1}, a_{h-1}, o_h, s_h) := \mathbb{P}^{\pi^E, \mathcal{P}}[s_{h-1}, a_{h-1}, o_h, s_h].$$

We first use union bound to decompose the misclassification error,

$$\mathbb{P}^{\pi^E, \mathcal{P}}\left[\exists h \in [H] : \widetilde{g}_h(s_{h-1}, a_{h-1}, o_h) \neq s_h\right] \leq \sum_{h \in [H]} \mathbb{P}^{\pi^E, \mathcal{P}}\left[\widetilde{g}_h(s_{h-1}, a_{h-1}, o_h) \neq s_h\right]$$

$$= \sum_{h \in [H]} \mathbb{P}_{(s_{h-1}, a_{h-1}, o_h, s_h) \sim D_h}\left[\widetilde{g}_h(s_{h-1}, a_{h-1}, o_h) \neq s_h\right].$$

(G.1)

For each $h \in [H]$, we can use $\widetilde{\mathcal{O}}\left(\frac{H}{\epsilon} \cdot \left(\max_{h \in [H]} d_{DS}^{3/2}(\mathcal{F}_h) + \log\left(\frac{1}{\delta}\right)\right)\right)$ episodes to collect $\widetilde{\mathcal{O}}\left(\frac{H}{\epsilon} \cdot \left(\max_{h \in [H]} d_{DS}^{3/2}(\mathcal{F}_h) + \log\left(\frac{1}{\delta}\right)\right)\right)$ samples from distribution $D_h$. Hence, by Theorem G.2, with probability at least $1 - \frac{\delta}{H}$, we have that

$$\mathbb{P}_{(s_{h-1}, a_{h-1}, o_h, s_h) \sim D_h}\left[\widetilde{g}_h(s_{h-1}, a_{h-1}, o_h) \neq s_h\right] \leq \frac{\epsilon}{H}.$$

Thus, by union bound, with probability at least $1 - \delta$, using a total of $\widetilde{\mathcal{O}}\left(\frac{H^2}{\epsilon} \cdot \left(\max_{h \in [H]} d_{DS}^{3/2}(\mathcal{F}_h) + \log\left(\frac{1}{\delta}\right)\right)\right)$ episodes we have that,

$$\sum_{h \in [H]} \mathbb{P}_{(s_{h-1}, a_{h-1}, o_h, s_h) \sim D_h}\left[\widetilde{g}_h(s_{h-1}, a_{h-1}, o_h) \neq s_h\right] \leq \epsilon,$$

which in combination with Equation (G.1) concludes the proof. ∎

## H   Missing Details in Section 5

**Proof of Theorem 5.1:** Let $\pi^* \in \operatorname{argmax}_{\pi \in \Pi^L} \mathbb{E}_{s_1 \sim \mu_1}[V_1^\pi(s_1)]$, where we define $V_1^\pi(s_1) := \mathbb{E}_{o_1 \sim \mathbb{O}_1(\cdot \mid s_1), a_1 \sim \pi_1(\cdot \mid o_1)}[Q_h^\pi(z_1 = (o_1), s_1, a_1)]$. We first note the following equation

$$\frac{1}{T} \sum_{t \in [T]} \mathbb{E}_{s_1 \sim \mu_1}[V_1^{\pi^t}(s_1)]$$

$$= \mathbb{E}_{s_1 \sim \mu_1}[V_1^{\pi^*}(s_1)] + \frac{1}{T} \sum_{t \in [T]} \mathbb{E}_{s_1 \sim \mu_1}\left(\widetilde{V}_1^{\pi^t}(s_1) - V_1^{\pi^*}(s_1)\right) + \frac{1}{T} \sum_{t \in [T]} \mathbb{E}_{s_1 \sim \mu_1}\left(V_1^{\pi^t}(s_1) - \widetilde{V}_1^{\pi^t}(s_1)\right).$$

$$(\text{H}.1)$$

Next, we make use of the performance difference lemma [39, 1, 74] on the extended space $\prod_{h \in [H]}(\mathcal{Z}_h \times \mathcal{S})$.

**Definition H.1.** Consider the class of policies $\Pi^{PL}$ such that at step $h \in [H]$, the policies in $\Pi^{PL}$ take an action based on finite memory up to this step and the use of the underlying state, e.g., for each policy $\pi_{1:H} \in \Pi^{PL}$, $\pi_h : \mathcal{Z}_h \times \mathcal{S} \to \Delta(\mathcal{A})$.

**Observation 1.** Note that $\Pi^L \subseteq \Pi^{PL}$.

**Lemma H.2** (Performance difference Lemma [39, 1, 74]; see e.g., Lemma 1 in [74]). For any pair of policies $\pi = \{\pi_h\}_{h \in [H]}, \pi' = \{\pi'_h\}_{h \in [H]} \in \Pi^{PL}$, and approximation of the $Q$-function of policy $\pi$, we have that for each state $s_1 \in \operatorname{supp}(\mu_1)$:

$$\widetilde{V}_1^\pi(z_1, s_1) - V_1^{\pi'}(z_1, s_1)$$

$$= \sum_{h \in [H]} \mathbb{E}_{\overline{\tau}_h \sim \pi' \mid z_1}\left[\left\langle \widetilde{Q}_h^\pi(z_h, s_h, \cdot), \pi_h(\cdot \mid z_h, s_h) - \pi'_h(\cdot \mid z_h, s_h)\right\rangle\right]$$

$$+ \sum_{h \in [H]} \mathbb{E}_{\overline{\tau}_h \sim \pi' \mid z_1}\left[\widetilde{Q}_h^\pi(z_h, s_h, a_h) - \mathbb{E}_{\substack{s_{h+1} \sim \mathbb{T}_h(\cdot \mid s_h, a_h),\\ o_{h+1} \sim \mathbb{O}_{h+1}(\cdot \mid s_{h+1})}}\left[r_h(s_h, a_h) + \widetilde{V}_{h+1}^\pi(z_{h+1}, s_{h+1})\right]\right],$$

where $\widetilde{V}_h^\pi(z_h, s_h) = \mathbb{E}_{a_h \sim \pi_h(\cdot \mid z_h)}[\widetilde{Q}_h^\pi(z_h, s_h, a_h)]$.

Setting $\pi = \pi^t \in \Pi^L \subseteq \Pi^{LP}$, and $\pi' = \pi^* \in \Pi^L \subseteq \Pi^{LP}$, where we remind that $\pi^* \in \operatorname{argmax}_{\pi \in \Pi^L} V_1^\pi(s_1)$, and for each $z_h \in \mathcal{Z}_h$ and $h \in [H]$, we abuse the notation by letting $\pi_h^t(\cdot \mid z_h, s_h) = \pi_h^t(\cdot \mid z_h)$ and $\pi_h^*(\cdot \mid z_h, s_h) = \pi_h^*(\cdot \mid z_h)$ for all $s_h \in \mathcal{S}$. The above formulation is thus simplified to

$$\mathbb{E}_{s_1 \sim \mu_1}[\widetilde{V}_1^{\pi^t}(s_1) - V_1^{\pi^*}(s_1)]$$

$$= \sum_{h \in [H]} \mathbb{E}_{\overline{\tau}_h \sim \pi^*}\left[\left\langle \widetilde{Q}_h^{\pi^t}(z_h, s_h, \cdot), \pi_h^t(\cdot \mid z_h, s_h) - \pi_h^*(\cdot \mid z_h, s_h)\right\rangle\right]$$

$$+ \sum_{h \in [H]} \mathbb{E}_{\overline{\tau}_h \sim \pi^*}\left[\widetilde{Q}_h^\pi(z_h, s_h, a_h) - \mathbb{E}_{\substack{s_{h+1} \sim \mathbb{T}_h(\cdot \mid s_h, a_h),\\ o_{h+1} \sim \mathbb{O}_{h+1}(\cdot \mid s_{h+1})}}\left[r_h(s_h, a_h) + \widetilde{V}_{h+1}^\pi(z_{h+1}, s_{h+1})\right]\right]$$

$$\geq \sum_{h \in [H]} \mathbb{E}_{\overline{\tau}_h \sim \pi^*}\left[\left\langle \widetilde{Q}_h^{\pi^t}(z_h, s_h, \cdot), \pi_h^t(\cdot \mid z_h, s_h) - \pi_h^*(\cdot \mid z_h, s_h)\right\rangle\right],$$

where in the inequality above we used Lemma H.3. Since our policy does not depend on the realized underlying state $s_h$,

$$\mathbb{E}_{s_1 \sim \mu_1}[\widetilde{V}_1^{\pi^t}(s_1) - V_1^{\pi^*}(s_1)]$$

$$\geq \sum_{h \in [H]} \mathbb{E}_{\overline{\tau}_h \sim \pi^*}\left[\left\langle \widetilde{Q}_h^{\pi^t}(z_h, s_h, \cdot), \pi_h^t(\cdot \mid z_h, s_h) - \pi_h^*(\cdot \mid z_h, s_h)\right\rangle\right]$$

$$= \sum_{h \in [H]} \mathbb{E}_{\tau_h \sim \pi^*}\left[\left\langle \mathbb{E}_{s_h \sim \boldsymbol{b}(\tau_h)}\left[\widetilde{Q}_h^{\pi^t}(z_h, s_h, \cdot)\right], \pi_h^t(\cdot \mid z_h) - \pi_h^*(\cdot \mid z_h)\right\rangle\right]$$

$$= \sum_{h \in [H]} \mathbb{E}_{\tau_h \sim \pi^*}\left[\left\langle \mathbb{E}_{s_h \sim \boldsymbol{b}^{\mathrm{apx}}(z_h)}\left[\widetilde{Q}_h^{\pi^t}(z_h, s_h, \cdot)\right], \pi_h^t(\cdot \mid z_h) - \pi_h^*(\cdot \mid z_h)\right\rangle\right]$$

$$+ \sum_{h \in [H]} \mathbb{E}_{\tau_h \sim \pi^*}\left[\left\langle \mathbb{E}_{s_h \sim \boldsymbol{b}(\tau_h)}\left[\widetilde{Q}_h^{\pi^t}(z_h, s_h, \cdot)\right] - \mathbb{E}_{s_h \sim \boldsymbol{b}^{\mathrm{apx}}(z_h)}\left[\widetilde{Q}_h^{\pi^t}(z_h, s_h, \cdot)\right], \pi_h^t(\cdot \mid z_h) - \pi_h^*(\cdot \mid z_h)\right\rangle\right]$$

$$\geq \sum_{h \in [H]} \mathbb{E}_{\tau_h \sim \pi^*}\left[\left\langle \mathbb{E}_{s_h \sim \boldsymbol{b}^{\mathrm{apx}}(z_h)}\left[\widetilde{Q}_h^{\pi^t}(z_h, s_h, \cdot)\right], \pi_h^t(\cdot \mid z_h) - \pi_h^*(\cdot \mid z_h)\right\rangle\right]$$

$$- \sum_{h \in [H]} \mathbb{E}_{\tau_h \sim \pi^*}\left[\left\|\mathbb{E}_{s_h \sim \boldsymbol{b}(\tau_h)}\left[\widetilde{Q}_h^{\pi^t}(z_h, s_h, \cdot)\right] - \mathbb{E}_{s_h \sim \boldsymbol{b}^{\mathrm{apx}}(z_h)}\left[\widetilde{Q}_h^{\pi^t}(z_h, s_h, \cdot)\right]\right\|_1\right]$$

$$\geq \sum_{h \in [H]} \mathbb{E}_{\tau_h \sim \pi^*}\left[\left\langle \mathbb{E}_{s_h \sim \boldsymbol{b}^{\mathrm{apx}}(z_h)}\left[\widetilde{Q}_h^{\pi^t}(z_h, s_h, \cdot)\right], \pi_h^t(\cdot \mid z_h) - \pi_h^*(\cdot \mid z_h)\right\rangle\right] - 2 \cdot H \cdot \sum_{h \in [H]} \mathbb{E}_{\tau_h \sim \pi^*}\left[d_{TV}(\boldsymbol{b}_h(\tau_h), \boldsymbol{b}_h^{\mathrm{apx}}(z_h))\right].$$

The last inequality follows by Lemma H.3. By averaging we get,

$$\frac{1}{T} \sum_{t \in [T]} \mathbb{E}_{s_1 \sim \mu_1}[\widetilde{V}_1^{\pi^t}(s_1)]$$

$$\geq \mathbb{E}_{s_1 \sim \mu_1}[V_1^{\pi^*}(s_1)] + \frac{1}{T} \sum_{h \in [H]} \mathbb{E}_{\tau_h \sim \pi^*}\left[\sum_{t \in [T]} \left\langle \mathbb{E}_{s_h \sim \boldsymbol{b}^{\mathrm{apx}}(z_h)}\left[\widetilde{Q}_h^{\pi^t}(z_h, s_h, \cdot)\right], \pi_h^t(\cdot \mid z_h) - \pi_h^*(\cdot \mid z_h)\right\rangle\right]$$

$$- 2 \cdot H \cdot \sum_{h \in [H]} \mathbb{E}_{\tau_h \sim \pi^*}\left[d_{TV}(\boldsymbol{b}_h(\tau_h), \boldsymbol{b}_h^{\mathrm{apx}}(z_h))\right]$$

$$\geq \mathbb{E}_{s_1 \sim \mu_1}[V_1^{\pi^*}(s_1)] + \frac{H}{T} \max_{h \in [H]} \mathbb{E}_{\tau_h \sim \pi^*}\left[\sum_{t \in [T]} \left\langle \mathbb{E}_{s_h \sim \boldsymbol{b}^{\mathrm{apx}}(z_h)}\left[\widetilde{Q}_h^{\pi^t}(z_h, s_h, \cdot)\right], \pi_h^t(\cdot \mid z_h) - \pi_h^*(\cdot \mid z_h)\right\rangle\right]$$

$$- 2 \cdot H^2 \cdot \max_{h \in [H]} \mathbb{E}_{\tau_h \sim \pi^*}\left[d_{TV}(\boldsymbol{b}_h(\tau_h), \boldsymbol{b}_h^{\mathrm{apx}}(z_h))\right]$$

$$\geq \mathbb{E}_{s_1 \sim \mu_1}[V_1^{\pi^*}(s_1)] - \frac{2H\sqrt{H \log(|\mathcal{A}|)}}{\sqrt{T}} - 2 \cdot H^2 \cdot \max_{h \in [H]} \mathbb{E}_{\tau_h \sim \pi^*}\left[d_{TV}(\boldsymbol{b}_h(\tau_h), \boldsymbol{b}_h^{\mathrm{apx}}(z_h))\right],$$

where the last inequality follows since for fixed $h \in [H]$ and $z_h \in \mathcal{Z}_h$, the agent updates her policy on memory $z_h$ according to MWU on feedback $\left\{\mathbb{E}_{s_h \sim \boldsymbol{b}^{\mathrm{apx}}(z_h)}\left[\widetilde{Q}_h^{\pi^t}(z_h, s_h, a)\right]\right\}_{a \in \mathcal{A}}$, and thus, the accumulate regret is bounded by (Section 4.3 in [10]):

$$\sum_{t \in [T]} \left\langle \mathbb{E}_{s_h \sim \boldsymbol{b}^{\mathrm{apx}}(z_h)}\left[\widetilde{Q}_h^{\pi^t}(z_h, s_h, \cdot)\right], \pi_h^*(\cdot \mid z_h) - \pi_h^t(\cdot \mid z_h)\right\rangle$$

$$\leq \frac{\log(|\mathcal{A}|)}{\eta} + \eta \cdot T \cdot \left\|Q_h^{\pi^t}(z_h, s_h, \cdot)\right\|_{+\infty} \leq \frac{\log(|\mathcal{A}|)}{\eta} + \eta \cdot T \cdot H = 2\sqrt{T \cdot H \log(|\mathcal{A}|)}.$$

The proof follows by combining Equation (H.1) and the inequality above. Finally, to achieve the near optimality in the class of $\Pi^L$, we bound the optimistic estimate using Equation (H.2) in Lemma H.3, and its global near-optimality for a large enough $L$ under $\gamma$-observability is a direct consequence of Theorem 4.1 in [26]. ∎

**Lemma H.3** (Optimistic $Q$-function - adapted from [48]). Given a policy $\pi \in \Pi^L$, and a parameter $M \in \mathbb{N}$, let $\{\widetilde{Q}_h^\pi : \mathcal{Z}_h \times \mathcal{S} \times \mathcal{A} \to [0, H]\}_{h \in [H]}$ be the output of Algorithm 3. Then, with probability at least $1 - \delta$: $\forall z_h \in \mathcal{Z}_h, s_h \in \mathcal{S}, a_h \in \mathcal{A}$

$$H - h + 1 \geq \widetilde{Q}_h^\pi(z_h, s_h, a_h) \geq \mathbb{E}_{\substack{s_{h+1} \sim \mathbb{T}_h(\cdot \,|\, s_h, a_h), \\ o_{h+1} \sim \mathbb{O}_{h+1}(\cdot \,|\, s_{h+1})}} \left[ r_h(s_h, a_h) + \widetilde{V}_{h+1}^\pi(z_{h+1}, s_{h+1}) \right],$$

$$\mathbb{E}_{s_1 \sim \mu_1}[\widetilde{V}_1^\pi(s_1)] - \mathbb{E}_{s_1 \sim \mu_1}[V_1^\pi(s_1)] \leq O\left( H^2 \cdot \sqrt{\frac{\max(O, S) \cdot S \cdot A}{M} \cdot \log\left(\frac{S \cdot A}{\delta}\right) \log\left(\frac{M \cdot S \cdot A \cdot H}{\delta}\right)} \right), \tag{H.2}$$

where $V_1^\pi(s_1) = \mathbb{E}_{o_1 \sim \mathbb{O}_1(\cdot \,|\, s_1), a_1 \sim \pi_1(\cdot \,|\, o_1)}[Q_h^\pi(z_1 = (o_1), s_1, a_1)]$, $\widetilde{V}_1^\pi(s_1) = \mathbb{E}_{o_1 \sim \mathbb{O}_1(\cdot \,|\, s_1), a_1 \sim \pi_1(\cdot \,|\, o_1)}[\widetilde{Q}_h^\pi(z_1 = (o_1), s_1, a_1)]$ and $\widetilde{V}_h^\pi(z_h, s_h) = \mathbb{E}_{a_h \sim \pi_h(\cdot \,|\, z_h)}[\widetilde{Q}_h^\pi(z_h, s_h, a_h)]$. Moreover, Algorithm 3 needs a total of $H \cdot M$ episodes from POMDP $\mathcal{P}$ and runs in time $\text{POLY}(H, M, S, A^L, O^L)$.

*Proof.* For each step $h \in [H]$, collect $M$ trajectories with states using policy $\pi$ on POMDP $\mathcal{P}$ and let $D_h = \{\overline{\tau}^{(i)}\}_{i \in [M]}$ be those collected trajectories. Define the empirical transition, observation and reward distribution as follows:

$$N_h(s_h, a_h, s_{h+1}) = \big| \{\overline{\tau} = (s_1', o_1', a_1', r_1' \ldots, s_h', o_h', a_h', r_h') \in D_h$$
$$: (s_h, a_h, s_{h+1}) = (s_h', a_h', s_{h+1}')\} \big|,$$

$$N_h(s_h, a_h) = \sum_{s_{h+1} \in \mathcal{S}} N_h(s_h, a_h, s_{h+1}),$$

$$N_h(s_h) = \sum_{a_h \in \mathcal{A}} N_h(s_h, a_h),$$

$$N_h(s_h, o_h) = \big| \{\overline{\tau} = (s_1', o_1', a_1', r_1' \ldots, s_h', o_h', a_h', r_h') \in D_h : (s_h, o_h) = (s_h', o_h')\} \big|,$$

$$\widehat{\mathbb{T}}_h(s_{h+1} \mid s_h, a_h) = \frac{N_h(s_h, a_h, s_{h+1})}{N_h(s_h, a_h)},$$

$$\widehat{\mathbb{O}}_h(o_h \mid s_h) = \frac{N_h(s_h, o_h)}{N_h(s_h)}.$$

Set $\delta_1 = \frac{\delta}{2 \cdot S \cdot (A+1)}$. By [13], there exists a constant $C > 0$ such that for each step $h \in [H]$, state $s \in \mathcal{S}$ and action $a \in \mathcal{A}$ with probability at least $1 - \delta_1$:

$$\|\mathbb{T}_h(\cdot \mid s_h, a_h) - \widehat{\mathbb{T}}_h(\cdot \mid s_h, a_h)\|_1 \leq \min\left( 2, C \cdot \sqrt{\frac{S \log(1/\delta_1)}{\max(N_h(s_h, a_h), 1)}} \right),$$

$$\|\mathbb{O}_h(\cdot \mid s_h) - \widehat{\mathbb{O}}_h(\cdot \mid s_h)\|_1 \leq \min\left( 2, C \cdot \sqrt{\frac{O \log(1/\delta_1)}{\max(N_h(s_h), 1)}} \right).$$

For the rest of the proof, we condition on this event. By union bound, this happens with probability at least $1 - \frac{\delta}{2}$. We define the optimistic $Q$-function recursively as follows for a memory-state pair $(z_h, s_h) \in \mathcal{Z}_h \times \mathcal{S}$:

$$\widetilde{Q}_{H+1}^\pi(z_{H+1}, s_{H+1}, \cdot) = 0, \qquad \forall z_{H+1} \in \mathcal{Z}_{H+1}, s_{H+1} \in \mathcal{S}$$

$$\widetilde{Q}_h^\pi(z_h, s_h, a_h) = \min\left( H - h + 1, \mathbb{E}_{\substack{s_{h+1} \sim \widehat{\mathbb{T}}_h(\cdot \,|\, s_h, a_h), \\ o_{h+1} \sim \widehat{\mathbb{O}}_{h+1}(\cdot \,|\, s_{h+1})}} [\widetilde{V}_{h+1}^\pi(z_{h+1}, s_{h+1})] + r(s_h, a_h) \right.$$

$$+ H \cdot \min\left( 2, C \cdot \sqrt{\frac{S \log(1/\delta_1)}{\max(N_h(s_h, a_h), 1)}} \right)$$

$$\left. + \mathbb{E}_{s_{h+1} \sim \widehat{\mathbb{T}}_h(\cdot \,|\, s_h, a_h)} H \cdot \min\left( 2, C \cdot \sqrt{\frac{O \log(1/\delta_1)}{\max(N_{h+1}(s_{h+1}), 1)}} \right) \right),$$

where $\widetilde{V}_{h+1}^{\pi}(z_{h+1}, s_{h+1}) = \mathbb{E}_{a_{h+1} \sim \pi_{h+1}(\cdot \mid z_{h+1})}[\widetilde{Q}_{h+1}^{\pi}(z_{h+1}, s_{h+1}, a_{h+1})]$. Hence the time complexity of our algorithm is $\text{POLY}(H, M, S, A^L, O^L)$. To prove the first condition, we fix step $h \in [H], z_h \in \mathcal{Z}_h, a_h \in \mathcal{A}$ and state $s_h \in \mathcal{S}$ and consider the case where $\widetilde{Q}_h^{\pi}(z_h, s_h, a_h) = H - h + 1$. In this case, since by assumption on the POMDP $\mathcal{P}$, $r_h(s_h, a_h) \leq 1$, and by definition of $\widetilde{Q}_{h+1}^{\pi}(\cdot, \cdot, \cdot) \leq H - h$, we have:

$$\widetilde{Q}_h^{\pi}(z_h, s_h, a_h) = 1 + H - h \geq \mathbb{E}_{\substack{s_{h+1} \sim \mathbb{T}_h(\cdot \mid s_h, a_h), \\ o_{h+1} \sim \mathbb{O}_{h+1}(\cdot \mid s_{h+1})}}[r_h(s_h, a_h) + \widetilde{V}_{h+1}^{\pi}(z_{h+1}, s_{h+1})].$$

If $\widetilde{Q}_h^{\pi}(z_h, s_h, a_h) \neq H - h + 1$, observe that:

$$\widetilde{Q}_h^{\pi}(z_h, s_h, a_h) = \mathbb{E}_{\substack{s_{h+1} \sim \widehat{\mathbb{T}}_h(\cdot \mid s_h, a_h), \\ o_{h+1} \sim \widehat{\mathbb{O}}_{h+1}(\cdot \mid s_{h+1})}} [\widetilde{V}_{h+1}^{\pi}(z_{h+1}, s_{h+1})] + r(s_h, a_h) + H \cdot \min\left(2, C \cdot \sqrt{\frac{S \log(1/\delta_1)}{\max(N_h(s_h, a_h), 1)}}\right)$$

$$+ \mathbb{E}_{s_{h+1} \sim \widehat{\mathbb{T}}_h(\cdot \mid s_h, a_h)} H \cdot \min\left(2, C \cdot \sqrt{\frac{O \log(1/\delta_1)}{\max(N_{h+1}(s_{h+1}), 1)}}\right)$$

$$\geq \mathbb{E}_{\substack{s_{h+1} \sim \mathbb{T}_h(\cdot \mid s_h, a_h), \\ o_{h+1} \sim \mathbb{O}_{h+1}(\cdot \mid s_{h+1})}} [r_h(s_h, a_h) + \widetilde{V}_{h+1}^{\pi}(z_{h+1}, s_{h+1})],$$

and hence, $\{\widetilde{Q}_h^{\pi}\}_{h \in [H]}$ satisfies the first condition. Moreover, it holds that:

$$\widetilde{Q}_h^{\pi}(z_h, s_h, a_h) \leq \mathbb{E}_{\substack{s_{h+1} \sim \mathbb{T}_h(\cdot \mid s_h, a_h), \\ o_{h+1} \sim \mathbb{O}_{h+1}(\cdot \mid s_{h+1})}} [r_h(s_h, a_h) + \widetilde{V}_{h+1}^{\pi}(z_{h+1}, s_{h+1})] + 2H \cdot \min\left(2, C \cdot \sqrt{\frac{S \log(1/\delta_1)}{\max(N_h(s_h, a_h), 1)}}\right)$$

$$+ 2 \cdot \mathbb{E}_{s_{h+1} \sim \widehat{\mathbb{T}}_h(\cdot \mid s_h, a_h)} H \cdot \min\left(2, C \cdot \sqrt{\frac{O \log(1/\delta_1)}{\max(N_{h+1}(s_{h+1}), 1)}}\right)$$

$$\leq \mathbb{E}_{\substack{s_{h+1} \sim \mathbb{T}_h(\cdot \mid s_h, a_h), \\ o_{h+1} \sim \mathbb{O}_{h+1}(\cdot \mid s_{h+1})}} [r_h(s_h, a_h) + \widetilde{V}_{h+1}^{\pi}(z_{h+1}, s_{h+1})] + 6H \cdot \min\left(2, C \cdot \sqrt{\frac{S \log(1/\delta_1)}{\max(N_h(s_h, a_h), 1)}}\right)$$

$$+ 2 \cdot \mathbb{E}_{s_{h+1} \sim \mathbb{T}_h(\cdot \mid s_h, a_h)} H \cdot \min\left(2, C \cdot \sqrt{\frac{O \log(1/\delta_1)}{\max(N_{h+1}(s_{h+1}), 1)}}\right).$$

Thus, it holds that:
$$\widetilde{V}_h^{\pi}(z_h, s_h) - V_h^{\pi}(z_h, s_h)$$

$$\leq \mathbb{E}_{\substack{a_h \sim \pi_h(\cdot \mid z_h), \\ s_{h+1} \sim \mathbb{T}_h(\cdot \mid s_h, a_h), \\ o_{h+1} \sim \mathbb{O}_{h+1}(\cdot \mid s_{h+1})}} [\widetilde{V}_{h+1}^{\pi}(z_{h+1}, s_{h+1}) - V_{h+1}^{\pi}(z_{h+1}, s_{h+1})] + 6 \cdot C \cdot H \cdot \mathbb{E}_{a_h \sim \pi_h(\cdot \mid z_h)} \sqrt{\frac{S \log(1/\delta_1)}{\max(N_h(s_h, a_h), 1)}}$$

$$+ 2 \cdot C \cdot H \cdot \mathbb{E}_{\substack{a_h \sim \pi_h(\cdot \mid z_h), \\ s_{h+1} \sim \mathbb{T}_h(\cdot \mid s_h, a_h)}} \sqrt{\frac{O \log(1/\delta_1)}{\max(N_{h+1}(s_{h+1}), 1)}}.$$

Thus, we conclude that

$$\mathbb{E}_{s_1 \sim \mu_1}[\widetilde{V}_1^{\pi}(s_1)] - \mathbb{E}_{s_1 \sim \mu_1}[V_1^{\pi}(s_1)] \leq \mathbb{E}_{\tau = (s_1, a_1, \ldots, s_{H+1}) \sim \pi} \left[ \sum_{h \in [H]} 8 \cdot H \cdot C \cdot \sqrt{\frac{\max(S, O) \log(1/\delta_1)}{\max(N_h(s_h, a_h), 1)}} \right]$$

$$= 8 \cdot H \sqrt{\max(O, S) \cdot \log(1/\delta_1)} \cdot C \cdot \sum_{h \in [H]} \mathbb{E}_{\tau = (s_1, a_1, \ldots, s_{H+1}) \sim \pi} \left[ \sqrt{\frac{1}{\max(N_h(s_h, a_h), 1)}} \right].$$

To finish the proof, we make use of the following lemma.

**Lemma H.4** (Lemma 6 in [48]). *For each step $h \in [H]$, and state-action pair $(s_h, a_h) \in \mathcal{S} \times \mathcal{A}$, with probability at least $1 - \delta_2$:*

$$\sqrt{\frac{1}{\max(N_h(s_h, a_h), 1)}} = O\left(\sqrt{\frac{S \cdot A \log(M/\delta_2)}{M}}\right).$$

By setting $\delta_2 = \frac{\delta}{2 \cdot S \cdot A \cdot H}$, and taking union bound we have that with probability at least $1 - \delta$:

$$\mathbb{E}_{s_1 \sim \mu_1}[\widetilde{V}_1^\pi(s_1)] - \mathbb{E}_{s_1 \sim \mu_1}[V_1^\pi(s_1)]$$

$$= 8\sqrt{\max(O, S) \cdot \log(1/\delta_1)} \cdot C \cdot \sum_{h \in [H]} \mathbb{E}_{\tau = (s_1, a_1, \ldots, s_{H+1}) \sim \pi} \left[ \sqrt{\frac{1}{\max(N_h(s_h, a_h), 1)}} \right]$$

$$\leq O\left( H^2 \cdot \sqrt{\frac{\max(O, S) \cdot S \cdot A}{M} \cdot \log\left(\frac{S \cdot A}{\delta}\right) \log\left(\frac{M \cdot S \cdot A \cdot H}{\delta}\right)} \right).$$

$\square$

**Proof of Theorem 5.2:** The proof of Theorem 5.2 follows by combining Theorem H.5 and Theorem H.6 below. Theorem H.5 proves that we can approximately learn a POMDP model $\mathcal{P}$ computationally and sample efficiently, thanks to the privileged information.

**Theorem H.5.** Fix any $\epsilon, \delta \in (0, 1)$. Algorithm 4 can learn the approximate POMDP model with transition $\widehat{\mathbb{T}}_{1:H}$ and emission $\widehat{\mathbb{O}}_{1:H}$ such that with probability at least $1 - \delta$, for any policy $\pi \in \Pi^{\mathrm{gen}}$ and $h \in [H]$

$$\mathbb{E}_\pi^{\mathcal{P}} \left[ \|\mathbb{T}_h(\cdot \mid s_h, a_h) - \widehat{\mathbb{T}}_h(\cdot \mid s_h, a_h)\|_1 + \|\mathbb{O}_h(\cdot \mid s_h) - \widehat{\mathbb{O}}_h(\cdot \mid s_h)\|_1 \right] \leq O(\epsilon),$$

using $\mathrm{POLY}(S, A, H, O, \frac{1}{\epsilon}, \log(\frac{1}{\delta}))$ episodes in time $\mathrm{POLY}(S, A, H, O, \frac{1}{\epsilon}, \log(\frac{1}{\delta}))$.

*Proof.* Note that by Lemma H.11, it suffices to consider only $\pi \in \Pi_\mathcal{S}$ as the optimal value for policies in $\Pi^{\mathrm{gen}}$ can be achieved by those in $\Pi_\mathcal{S}$ (by considering $r_h(s_h, a_h) := \|\mathbb{T}_h(\cdot \mid s_h, a_h) - \widehat{\mathbb{T}}_h(\cdot \mid s_h, a_h)\|_1 + \|\mathbb{O}_h(\cdot \mid s_h) - \widehat{\mathbb{O}}_h(\cdot \mid s_h)\|_1$). For each $h \in [H]$ and $s_h \in \mathcal{S}$, we define

$$p_h(s_h) = \max_{\pi \in \Pi_\mathcal{S}} d_h^\pi(s_h).$$

Fix $\epsilon_1 > 0$, we define $\mathcal{U}(h, \epsilon_1) = \{s_h \in \mathcal{S} \mid p_h(s_h) \geq \epsilon_1\}$. By the guarantee of the EULER algorithm from [89, 37], one can learn a policy $\Psi(h, s_h)$ with sample complexity $\widetilde{\mathcal{O}}(\frac{S^2 A H^4}{\epsilon_1})$ such that $d_h^{\Psi(h, s_h)}(s_h) \geq \frac{p_h(s_h)}{2}$ for each $s_h \in \mathcal{U}(h, \epsilon_1)$ with probability $1 - \delta_1$. Now we assume this event holds for any $h \in [H]$ and $s_h \in \mathcal{U}(h, \epsilon_1)$. For each $s_h \in \mathcal{S}$ and $a_h \in \mathcal{A}$, we have executed in Algorithm 4 the policy $\Psi(h, s_h)$ followed by an action $a_h \in \mathcal{A}$ for $N$ episodes, and denote the number of episodes that $s_h$ and $a_h$ are visited as $N_h(s_h, a_h)$. Then with probability $1 - e^{N\epsilon_1/8}$, $N_h(s_h, a_h) \geq \frac{N p_h(s_h)}{2}$ by Chernoff bound. Now conditioned on this event, we are ready to evaluate the following: for any $\pi \in \Pi_\mathcal{S}$

$$\frac{1}{2} \cdot \mathbb{E}_\pi^{\mathcal{P}} \|\mathbb{T}_h(\cdot \mid s_h, a_h) - \widehat{\mathbb{T}}_h(\cdot \mid s_h, a_h)\|_1 = \frac{1}{2} \sum_{s_h, a_h} d_h^\pi(s_h) \pi_h(a_h \mid s_h) \|\mathbb{T}_h(\cdot \mid s_h, a_h) - \widehat{\mathbb{T}}_h(\cdot \mid s_h, a_h)\|_1$$

$$\leq S\epsilon_1 + \frac{1}{2} \sum_{s_h \in \mathcal{U}(h, \epsilon_1), a_h} d_h^\pi(s_h) \pi_h(a_h \mid s_h) \sqrt{\frac{2S \log(1/\delta_2)}{N p_h(s_h)}}$$

$$\leq S\epsilon_1 + \sum_{s_h \in \mathcal{U}(h, \epsilon_1), a_h} d_h^{\Psi(h, s_h)}(s_h) \pi_h(a_h \mid s_h) \sqrt{\frac{2S \log(1/\delta_2)}{N p_h(s_h)}}$$

$$\leq S\epsilon_1 + \sum_{s_h} \sqrt{d_h^{\Psi(h, s_h)}(s_h)} \sqrt{\frac{2S \log(1/\delta_2)}{N}}$$

$$\leq S\epsilon_1 + S\sqrt{\frac{2 \log(1/\delta_2)}{N}},$$

where the first inequality follows by [13] with probability at least $1 - \delta_2$, and the second inequality uses $d_h^{\Psi(h,s_h)}(s_h) \geq \frac{p_h(s_h)}{2}$ for all $s_h \in \mathcal{U}(h, \epsilon_1)$. Similarly, for any $\pi \in \Pi_\mathcal{S}$

$$\frac{1}{2} \cdot \mathbb{E}_\pi^\mathcal{P} \|\mathbb{O}_h(\cdot \mid s_h) - \widehat{\mathbb{O}}_h(\cdot \mid s_h)\|_1 \leq S\epsilon_1 + \frac{1}{2} \sum_{s_h \in \mathcal{U}(h,\epsilon_1)} d_h^\pi(s_h) \sqrt{\frac{2O \log(1/\delta_2)}{N p_h(s_h)}}$$

$$\leq S\epsilon_1 + \sum_{s_h \in \mathcal{U}(h,\epsilon_1)} d_h^{\Psi(h,s_h)}(s_h) \sqrt{\frac{2O \log(1/\delta_2)}{N p_h(s_h)}} \leq S\epsilon_1 + \sum_{s_h \in \mathcal{U}(h,\epsilon_1)} \sqrt{d_h^{\Psi(h,s_h)}(s_h)} \sqrt{\frac{2O \log(1/\delta_2)}{N}}$$

$$\leq S\epsilon_1 + \sqrt{\frac{2SO \log(1/\delta_2)}{N}},$$

where similarly the first inequality follows by [13] with probability at least $1 - \delta_2$, and the second inequality uses $d_h^{\Psi(h,s_h)}(s_h) \geq \frac{p_h(s_h)}{2}$ for all $s_h \in \mathcal{U}(h, \epsilon_1)$. Therefore, by a union bound, all the high probability events above hold with probability

$$1 - SH\delta_1 - SHAe^{-N\epsilon_1/8} - 2SAH\delta_2.$$

Therefore, we can choose $N = \widetilde{\Theta}(\frac{S^2 + SO}{\epsilon^2})$ and $\epsilon_1 = \Theta(\frac{\epsilon}{S})$, leading to the total sample complexity

$$SHA \left( N + \widetilde{\Theta} \left( \frac{S^3 AH^4}{\epsilon} \right) \right) = \widetilde{\Theta} \left( \frac{S^2 AHO + S^3 AH}{\epsilon^2} + \frac{S^4 A^2 H^5}{\epsilon} \right), \qquad \text{(H.3)}$$

which completes the proof. $\qquad \square$

Now with such a model learned in a reward-free way, we are ready to present our main result for approximate belief learning.

**Theorem H.6.** Consider a $\gamma$-observable POMDP $\mathcal{P}$ (c.f. Assumption 2.2), an $\epsilon > 0$, approximate transition and emission $\{\widehat{\mathbb{T}}_h\}_{h \in [H]}$ and $\{\widehat{\mathbb{O}}_h\}_{h \in [H]}$ learned from Algorithm 4 that ensure that for any $\pi \in \Pi^{\text{gen}}$ and $h \in [H]$:

$$\mathbb{E}_\pi^\mathcal{P} \left[ \|\mathbb{T}_h(\cdot \mid s_h, a_h) - \widehat{\mathbb{T}}_h(\cdot \mid s_h, a_h)\|_1 + \|\mathbb{O}_h(\cdot \mid s_h) - \widehat{\mathbb{O}}_h(\cdot \mid s_h)\|_1 \right] \leq \mathcal{O}\left( \frac{\epsilon}{H} \right).$$

Then we can construct in time $\text{POLY}(H, A, S, O, \frac{1}{\epsilon}, \log \frac{1}{\delta})$ a belief $\{b_h^{\text{apx}} : \mathcal{Z}_h \to \Delta(\mathcal{S})\}_{h \in [H]}$ with no further samples. In addition, if the parameter in Algorithm 4 satisfies $N = \widetilde{\Theta}(\frac{O \log(SH/\delta)}{\gamma^2 \epsilon_1})$ and our class of finite memory policies $\Pi^L$ satisfies $L \geq \widetilde{\Omega}(\gamma^{-4} \log(S/\epsilon))$, then for any $\pi \in \Pi^{\text{gen}}$ and $h \in [H]$:

$$\mathbb{E}_\pi^\mathcal{P} \|b_h(\tau_h) - b_h^{\text{apx}}(z_h)\|_1 \leq \mathcal{O}(\epsilon).$$

*Proof.* We firstly consider the following simple while important fact: for the estimated emission $\widehat{\mathbb{O}}_h$, its observability can be evaluated as

$$\|\widehat{\mathbb{O}}_h^\top (b - b')\|_1 \geq \|\mathbb{O}_h^\top (b - b')\|_1 - \|(\mathbb{O}_h^\top - \widehat{\mathbb{O}}_h^\top)(b - b')\|_1 \geq (\gamma - \|\widehat{\mathbb{O}}_h - \mathbb{O}_h\|_\infty) \|b - b'\|_1,$$

for any $b, b' \in \Delta(\mathcal{S})$ and $\|\widehat{\mathbb{O}}_h - \mathbb{O}_h\|_\infty := \max_{s_h \in \mathcal{S}} \|\mathbb{O}_h(\cdot \mid s_h) - \widehat{\mathbb{O}}_h(\cdot \mid s_h)\|_1$. Therefore, if one can ensure that the emission at *any* state $s_h$ is learned accurately in the sense that $\|\mathbb{O}_h(\cdot \mid s_h) - \widehat{\mathbb{O}}_h(\cdot \mid s_h)\|_1 \leq \frac{\gamma}{2}$, we can conclude that $\widehat{\mathbb{O}}_h$ is also $\gamma/2$-observable. However, the key challenge here is that there could exist some states $s_h$ that can only be visited *with a low probability no matter what exploration policy is used*. Therefore, emissions at such states may not be learned accurately. To address this issue, our key technique is to *redirect* the transition probability into states that cannot be explored sufficiently to some highly visited states, in a new POMDP that is close to the original one in generating the beliefs.

Specifically, first, for any $\epsilon_1 > 0$, we define

$$\mathcal{S}_h^{\text{low}} := \left\{ s_h \in \mathcal{S} \,\middle|\, \frac{N_h(s_h)}{N} < \frac{\epsilon_1}{2} \right\}, \quad \mathcal{S}_h^{\text{high}} := \mathcal{S} \setminus \mathcal{S}_h^{\text{low}}.$$

By Chernoff bound, with probability at least $1 - Se^{-N\epsilon_1/8}$, it holds that
$$\mathcal{S}_h^{\text{low}} \subseteq \{s_h \in \mathcal{S} \,|\, p_h(s_h) < \epsilon_1\},$$
where $p_h(s_h) := \max_{\pi \in \Pi_S} d_h^\pi(s_h)$. To see the reason, we notice that for any $s_h$ such that $p_h(s_h) \geq \epsilon_1$, with probability $1 - e^{-N\epsilon_1/8}$, it holds that $\frac{N_h(s_h)}{N} \geq \frac{\epsilon_1}{2}$. Therefore, the last step is by taking a union bound for all $s_h$. Now with $\mathcal{S}_h^{\text{low}}$ defined, we are ready to construct a truncated POMDP $\mathcal{P}^{\text{trunc}}$ such that for each $h \in [H]$, we define the transition as

$$\mathbb{T}_h^{\text{trunc}}(s_{h+1} \,|\, s_h, a_h) := \mathbb{T}_h(s_{h+1} \,|\, s_h, a_h) + \frac{\sum_{s_{h+1}' \in \mathcal{S}_{h+1}^{\text{low}}} \mathbb{T}_h(s_{h+1}' \,|\, s_h, a_h)}{|\mathcal{S}_{h+1}^{\text{high}}|}, \quad \forall s_h \in \mathcal{S}, s_{h+1} \in \mathcal{S}_{h+1}^{\text{high}}, a_h \in \mathcal{A},$$

$$\mathbb{T}_h^{\text{trunc}}(s_{h+1} \,|\, s_h, a_h) := 0, \quad \forall s_h \in \mathcal{S}, s_{h+1} \in \mathcal{S}_{h+1}^{\text{low}}, a_h \in \mathcal{A}.$$

Meanwhile, for the initial state distribution, we define

$$\mu_1^{\text{trunc}}(s_1) := \mu_1(s_1) + \frac{\sum_{s_1' \in \mathcal{S}_1^{\text{low}}} \mu_1(s_1')}{|\mathcal{S}_1^{\text{high}}|}, \quad \forall s_1 \in \mathcal{S}_1^{\text{high}},$$

$$\mu_1^{\text{trunc}}(s_1) := 0, \quad \forall s_1 \in \mathcal{S}_1^{\text{low}}.$$

For emission, we simply define

$$\mathbb{O}_h^{\text{trunc}}(o_h \,|\, s_h) := \mathbb{O}_h(o_h \,|\, s_h), \quad \forall h \in [H], s_h \in \mathcal{S}, o_h \in \mathcal{O}_h.$$

Finally, we define the rewards for $\mathcal{P}^{\text{trunc}}$ arbitrarily. Now we examine the total variation distance between the trajectory distributions in $\mathcal{P}$ and $\mathcal{P}^{\text{trunc}}$. For any policy $\pi \in \Pi^{\text{gen}}$, it is easy to see that

$$\mathbb{P}^{\pi,\mathcal{P}}(\overline{\tau}_h) \leq \mathbb{P}^{\pi,\mathcal{P}^{\text{trunc}}}(\overline{\tau}_h),$$

for any $\overline{\tau}_h \in \overline{\mathcal{T}}_h^{\text{high}} := \{(s_{1:h}, o_{1:h}, a_{1:h-1}) \,|\, s_{h'}' \in \mathcal{S}_{h'}^{\text{high}}, \forall h' \in [h]\}$, since some rarely visited states' probability has been redirected to the highly visited ones in $\mathcal{P}^{\text{trunc}}$. Meanwhile, it holds by a union bound that for any $h \in [H]$

$$\mathbb{P}^{\pi,\mathcal{P}}(\overline{\tau}_h \notin \overline{\mathcal{T}}_h^{\text{high}}) \leq \sum_{h' \in [h]} \mathbb{P}^{\pi,\mathcal{P}}(s_{h'} \in \mathcal{S}_{h'}^{\text{low}}) \leq HS\epsilon_1.$$

Therefore, by noticing that $\mathbb{P}^{\pi,\mathcal{P}^{\text{trunc}}}(\overline{\tau}_h) = 0$ for any $\overline{\tau}_h \notin \overline{\mathcal{T}}_h^{\text{high}}$ and $h \in [H]$, the total variation distance between the trajectory distributions in $\mathcal{P}$ and $\mathcal{P}^{\text{trunc}}$ can be bounded by

$$\sum_{\overline{\tau}_h} |\mathbb{P}^{\pi,\mathcal{P}}(\overline{\tau}_h) - \mathbb{P}^{\pi,\mathcal{P}^{\text{trunc}}}(\overline{\tau}_h)| \leq 2HS\epsilon_1. \tag{H.4}$$

On the other hand, by Equation (H.9) of Lemma H.9, we have

$$\mathbb{E}_\pi^{\mathcal{P}}[\|\boldsymbol{b}_h(\tau_h) - \boldsymbol{b}_h^{\text{trunc}}(\tau_h)\|_1] \leq 2 \sum_{\overline{\tau}_h} |\mathbb{P}^{\pi,\mathcal{P}}(\overline{\tau}_h) - \mathbb{P}^{\pi,\mathcal{P}^{\text{trunc}}}(\overline{\tau}_h)| \leq 4HS\epsilon_1. \tag{H.5}$$

With such an intermediate quantity $\mathcal{P}^{\text{trunc}}$, we define the transition of its approximate version $\widehat{\mathcal{P}}^{\text{trunc}}$ as follows: we define the transition as

$$\widehat{\mathbb{T}}_h^{\text{trunc}}(s_{h+1} \,|\, s_h, a_h) := \widehat{\mathbb{T}}_h(s_{h+1} \,|\, s_h, a_h) + \frac{\sum_{s_{h+1}' \in \mathcal{S}_{h+1}^{\text{low}}} \widehat{\mathbb{T}}_h(s_{h+1}' \,|\, s_h, a_h)}{|\mathcal{S}_{h+1}^{\text{high}}|}, \quad \forall s_h \in \mathcal{S}, s_{h+1} \in \mathcal{S}_{h+1}^{\text{high}}, a_h \in \mathcal{A},$$

$$\widehat{\mathbb{T}}_h^{\text{trunc}}(s_{h+1} \,|\, s_h, a_h) := 0, \quad \forall s_h \in \mathcal{S}, s_{h+1} \in \mathcal{S}_{h+1}^{\text{low}}, a_h \in \mathcal{A}.$$

Meanwhile, for the initial state distribution, we define

$$\widehat{\mu}_1^{\text{trunc}}(s_1) := \widehat{\mu}_1(s_1) + \frac{\sum_{s_1' \in \mathcal{S}_1^{\text{low}}} \widehat{\mu}_1(s_1')}{|\mathcal{S}_1^{\text{high}}|}, \quad \forall s_1 \in \mathcal{S}_1^{\text{high}},$$

$$\widehat{\mu}_1^{\text{trunc}}(s_1) := 0, \quad \forall s_1 \in \mathcal{S}_1^{\text{low}}.$$

For emission, we define

$$\widehat{\mathbb{O}}_h^{\mathrm{trunc}}(o_h \mid s_h) := \widehat{\mathbb{O}}_h(o_h \mid s_h), \quad \forall h \in [H], s_h \in \mathcal{S}, o_h \in \mathcal{O}_h.$$

Now we define $\widehat{\mathbb{O}}_h^{\mathrm{sub}} \in \mathbb{R}^{|\mathcal{S}_h^{\mathrm{high}}| \times O}$ to be the sub-matrix of $\widehat{\mathbb{O}}_h^{\mathrm{trunc}}$, where we only keep those rows of $\widehat{\mathbb{O}}_h^{\mathrm{trunc}}$ that correspond to the states in $\mathcal{S}_h^{\mathrm{high}}$. Similarly, we define $\mathbb{O}_h^{\mathrm{sub}} \in \mathbb{R}^{|\mathcal{S}_h^{\mathrm{high}}| \times O}$ to be the sub-matrix of $\mathbb{O}_h$, where we only keep those rows of $\mathbb{O}_h$ that correspond to the states in $\mathcal{S}_h^{\mathrm{high}}$. It is direct to see that $\mathbb{O}^{\mathrm{sub}}$ is still $\gamma$-observable. Meanwhile, we notice that

$$\|\widehat{\mathbb{O}}_h^{\mathrm{sub}} - \mathbb{O}_h^{\mathrm{sub}}\|_\infty = \max_{s_h \in \mathcal{S}_h^{\mathrm{high}}} \|\mathbb{O}_h(\cdot \mid s_h) - \widehat{\mathbb{O}}_h(\cdot \mid s_h)\|_1 \leq \max_{s_h \in \mathcal{S}_h^{\mathrm{high}}} \sqrt{\frac{O \log(SH/\delta)}{N_h(s_h)}} \leq \max_{s_h \in \mathcal{S}_h^{\mathrm{high}}} \sqrt{\frac{2O \log(SH/\delta)}{N \epsilon_1}},$$

where the first inequality is by Lemma J.9, and the second inequality is by the definition of $\mathcal{S}_h^{\mathrm{high}}$. Therefore, if we take

$$N \geq \frac{8O \log(SH/\delta)}{\gamma^2 \epsilon_1},$$

it is guaranteed that $\|\widehat{\mathbb{O}}_h^{\mathrm{sub}} - \mathbb{O}_h^{\mathrm{sub}}\|_\infty \leq \frac{\gamma}{2}$. Therefore, we conclude that $\widehat{\mathbb{O}}_h^{\mathrm{sub}}$ is also $\gamma/2$-observable.

Now we are ready to examine $\widehat{\boldsymbol{b}}_h^{\prime,\mathrm{trunc}}$. We firstly define the following POMDP $\widehat{\mathcal{P}}^{\mathrm{sub}}$, which essentially deletes all states in $\mathcal{S}_h^{\mathrm{low}}$ from the state space of $\widehat{\mathcal{P}}^{\mathrm{trunc}}$ at each step $h$, which does not affect the trajectory distribution as they were not reachable in $\widehat{\mathcal{P}}^{\mathrm{trunc}}$. Notice that the emission of $\widehat{\mathcal{P}}^{\mathrm{sub}}$ is exactly $\widehat{\mathbb{O}}_h^{\mathrm{sub}}$, implying that $\widehat{\mathcal{P}}^{\mathrm{sub}}$ is a $\gamma/2$-observable POMDP. Therefore, for policy class with finite memory $\Pi^L$ with $L \geq \widetilde{\Omega}(\gamma^{-4} \log(S/\epsilon))$, by Theorem 4.1 in [25], it is guaranteed that for any $\pi \in \Pi$,

$$\mathbb{E}_\pi^{\widehat{\mathcal{P}}^{\mathrm{sub}}} \|\widehat{\boldsymbol{b}}_h^{\mathrm{sub}}(\tau_h) - \widehat{\boldsymbol{b}}_h^{\prime,\mathrm{sub}}(z_h)\|_1 \leq \epsilon,$$

where $\widehat{\boldsymbol{b}}_h^{\mathrm{sub}}(\tau_h), \widehat{\boldsymbol{b}}_h^{\prime,\mathrm{sub}}(z_h) \in \Delta(\mathcal{S}_h^{\mathrm{high}})$. Now we claim that

$$\mathbb{E}_\pi^{\widehat{\mathcal{P}}^{\mathrm{trunc}}} \|\widehat{\boldsymbol{b}}_h^{\mathrm{trunc}}(\tau_h) - \boldsymbol{b}_h^{\mathrm{apx}}(z_h)\|_1 \leq \epsilon, \tag{H.6}$$

where we define $\boldsymbol{b}_h^{\mathrm{apx}}(z_h) \in \Delta(\mathcal{S})$ by *augmenting* $\widehat{\boldsymbol{b}}_h^{\prime,\mathrm{sub}}(z_h)$ with 0 for states from $\mathcal{S}_h^{\mathrm{low}}$. To see the reason, we notice that simulating $\widehat{\mathcal{P}}^{\mathrm{trunc}}$ is exactly equivalent to simulating $\widehat{\mathcal{P}}^{\mathrm{sub}}$, and that $\widehat{\boldsymbol{b}}_h^{\mathrm{trunc}}(\tau_h)(s_h) = 0$ for $s_h \in \mathcal{S}_h^{\mathrm{low}}$, $\widehat{\boldsymbol{b}}_h^{\mathrm{trunc}}(\tau_h)(s_h) = \widehat{\boldsymbol{b}}_h^{\mathrm{sub}}(\tau_h)(s_h)$ for $s_h \in \mathcal{S}_h^{\mathrm{high}}$.

For the total variation distance between the trajectory distributions in $\mathcal{P}^{\mathrm{trunc}}$ and $\widehat{\mathcal{P}}^{\mathrm{trunc}}$, it holds that by Lemma H.9

$$\sum_{\overline{\tau}_H} |\mathbb{P}^{\pi, \mathcal{P}^{\mathrm{trunc}}}(\overline{\tau}_H) - \mathbb{P}^{\pi, \widehat{\mathcal{P}}^{\mathrm{trunc}}}(\overline{\tau}_H)|$$

$$\leq \mathbb{E}_\pi^{\mathcal{P}^{\mathrm{trunc}}} \left[ \sum_{h \in [H]} \|\mathbb{T}_h^{\mathrm{trunc}}(\cdot \mid s_h, a_h) - \widehat{\mathbb{T}}_h^{\mathrm{trunc}}(\cdot \mid s_h, a_h)\|_1 + \|\mathbb{O}_h^{\mathrm{trunc}}(\cdot \mid s_h) - \widehat{\mathbb{O}}_h^{\mathrm{trunc}}(\cdot \mid s_h)\|_1 \right]$$

$$\leq \sum_{h \in [H]} \mathbb{E}_\pi^{\mathcal{P}^{\mathrm{trunc}}} \left[ \|\mathbb{T}_h(\cdot \mid s_h, a_h) - \widehat{\mathbb{T}}_h(\cdot \mid s_h, a_h)\|_1 + \|\mathbb{O}_h(\cdot \mid s_h) - \widehat{\mathbb{O}}_h(\cdot \mid s_h)\|_1 \right],$$

where the last step is by Lemma H.7.

Now by Equation (H.4), we have

$$\sum_h \mathbb{E}_\pi^{\mathcal{P}^{\mathrm{trunc}}} \left[ \|\mathbb{T}_h(\cdot \mid s_h, a_h) - \widehat{\mathbb{T}}_h(\cdot \mid s_h, a_h)\|_1 + \|\mathbb{O}_h(\cdot \mid s_h) - \widehat{\mathbb{O}}_h(\cdot \mid s_h)\|_1 \right]$$

$$= \sum_h 8HS\epsilon_1 + \mathbb{E}_\pi^{\mathcal{P}} \left[ \|\mathbb{T}_h(\cdot \mid s_h, a_h) - \widehat{\mathbb{T}}_h(\cdot \mid s_h, a_h)\|_1 + \|\mathbb{O}_h(\cdot \mid s_h) - \widehat{\mathbb{O}}_h(\cdot \mid s_h)\|_1 \right]$$

$$\leq \frac{\epsilon}{3} + 8H^2 S\epsilon_1,$$

where the last step is by Theorem H.5. Hence, by Lemma H.9, it holds that

$$\|\mathbb{P}^{\pi,\mathcal{P}^{\mathrm{trunc}}} - \mathbb{P}^{\pi,\widehat{\mathcal{P}}^{\mathrm{trunc}}}\|_1 \le \frac{\epsilon}{3} + 8H^2 S\epsilon_1, \tag{H.7}$$

$$\mathbb{E}^{\mathcal{P}^{\mathrm{trunc}}}_{\pi}\|\boldsymbol{b}^{\mathrm{trunc}}_h(\tau_h) - \widehat{\boldsymbol{b}}^{\mathrm{trunc}}_h(\tau_h)\|_1 \le \frac{2\epsilon}{3} + 16H^2 S\epsilon_1. \tag{H.8}$$

Finally, we are ready to prove

$$\mathbb{E}^{\mathcal{P}}_{\pi}\|\boldsymbol{b}_h(\tau_h) - \boldsymbol{b}^{\mathrm{apx}}_h(z_h)\|_1$$
$$\le \mathbb{E}^{\mathcal{P}}_{\pi}\|\boldsymbol{b}_h(\tau_h) - \boldsymbol{b}^{\mathrm{trunc}}_h(\tau_h)\|_1 + \mathbb{E}^{\mathcal{P}}_{\pi}\|\boldsymbol{b}^{\mathrm{trunc}}_h(\tau_h) - \widehat{\boldsymbol{b}}^{\mathrm{trunc}}_h(\tau_h)\|_1 + \mathbb{E}^{\mathcal{P}}_{\pi}\|\widehat{\boldsymbol{b}}^{\mathrm{trunc}}_h(\tau_h) - \boldsymbol{b}^{\mathrm{apx}}_h(z_h)\|_1$$
$$\le \mathbb{E}^{\mathcal{P}}_{\pi}\|\boldsymbol{b}_h(\tau_h) - \boldsymbol{b}^{\mathrm{trunc}}_h(\tau_h)\|_1 + \mathbb{E}^{\mathcal{P}^{\mathrm{trunc}}}_{\pi}\|\boldsymbol{b}^{\mathrm{trunc}}_h(\tau_h) - \widehat{\boldsymbol{b}}^{\mathrm{trunc}}_h(\tau_h)\|_1 + \mathbb{E}^{\widehat{\mathcal{P}}^{\mathrm{trunc}}}_{\pi}\|\widehat{\boldsymbol{b}}^{\mathrm{trunc}}_h(\tau_h) - \boldsymbol{b}^{\mathrm{apx}}_h(z_h)\|_1$$
$$\qquad + 4\|\mathbb{P}^{\pi,\mathcal{P}} - \mathbb{P}^{\pi,\mathcal{P}^{\mathrm{trunc}}}\|_1 + 2\|\mathbb{P}^{\pi,\mathcal{P}^{\mathrm{trunc}}} - \mathbb{P}^{\pi,\widehat{\mathcal{P}}^{\mathrm{trunc}}}\|_1$$
$$\le \mathcal{O}(H^2 S\epsilon_1) + \mathcal{O}(\epsilon).$$

Therefore, by setting $\epsilon_1 = \Theta\left(\frac{\epsilon}{H^2 S}\right)$, we prove our lemma. Observe that our algorithm needed no further samples. The computational complexity follows by computing belief $\boldsymbol{b}^{\mathrm{apx}}_h$ on POMDP $\widehat{\mathcal{P}}^{\mathrm{sub}}$ using finite-memory policies of size $\widetilde{\Theta}(\gamma^{-4}\log(S/\epsilon))$. For the final sample complexity, we only need to ensure $N = \widetilde{\Theta}(\frac{O\log(SH/\delta)}{\gamma^2\epsilon_1})$ in Equation (H.3), thus concluding our proof. $\qquad\square$

We conclude the proof of Theorem 5.2 by combining Theorem H.5 and Theorem H.6. $\qquad\blacksquare$

## H.1 Supporting Technical Lemmas

In the following, we provide some technical lemmas and their proofs.

**Lemma H.7.** Fix $n > 0$ and an index set $S \subseteq [n]$. For two sequences $x_{1:n}$ and $y_{1:n}$ such that $x_i, y_i \in [0,1]$ for $i \in [n]$ and $\sum_i x_i = 1, \sum_i y_i = 1$, we define

$$\widehat{x}_i = x_i + \frac{\sum_{j \in S} x_j}{n - |S|}, \quad \forall i \notin S; \qquad \widehat{x}_i = 0, \quad \forall i \in S.$$

We define $\widehat{y}_{1:n}$ similarly. Then, it holds that

$$\sum_i |x_i - y_i| \ge \sum_i |\widehat{x}_i - \widehat{y}_i|.$$

*Proof.* Note that

$$\sum_i |\widehat{x}_i - \widehat{y}_i| = \sum_{i \notin S} |\widehat{x}_i - \widehat{y}_i| = \sum_{i \notin S} \left| x_i + \frac{\sum_{j \in S} x_j}{n - |S|} - y_i - \frac{\sum_{j \in S} y_j}{n - |S|} \right|$$
$$\le (n - |S|) \left| \frac{\sum_{j \in S} x_j}{n - |S|} - \frac{\sum_{j \in S} y_j}{n - |S|} \right| + \sum_{i \notin S} |x_i - y_i|$$
$$\le \sum_i |x_i - y_i|,$$

which completes the proof. $\qquad\square$

**Lemma H.8.** Fix two finite sets $\mathcal{X}, \mathcal{Y}$ and two joint distributions $P_1, P_2 \in \Delta(\mathcal{X} \times \mathcal{Y})$. It holds that

$$-\mathbb{E}_{P_1(x)} \sum_y |P_1(y \mid x) - P_2(y \mid x)| \le \sum_{x,y} |P_1(x,y) - P_2(x,y)| - \sum_x |P_1(x) - P_2(x)|$$
$$\le \mathbb{E}_{P_1(x)} \sum_y |P_1(y \mid x) - P_2(y \mid x)|.$$

*Proof.* For the second inequality, it holds that

$$\sum_{x,y} |P_1(x,y) - P_2(x,y)| = \sum_{x,y} |P_1(x,y) - P_1(x)P_2(y\,|\,x) + P_1(x)P_2(y\,|\,x) - P_2(x,y)|$$

$$\leq \sum_{x,y} |P_1(x,y) - P_1(x)P_2(y\,|\,x)| + |P_1(x)P_2(y\,|\,x) - P_2(x,y)|$$

$$= \sum_{x,y} |P_1(x)(P_1(y\,|\,x) - P_2(y\,|\,x))| + |P_2(y\,|\,x)(P_1(x) - P_2(x))|$$

$$= \mathbb{E}_{P_1(x)} \sum_y |P_1(y\,|\,x) - P_2(y\,|\,x)| + \sum_x |P_1(x) - P_2(x)|.$$

Meanwhile, for the first inequality, it holds that

$$\sum_{x,y} |P_1(x,y) - P_2(x,y)| = \sum_{x,y} |P_1(x,y) - P_1(x)P_2(y\,|\,x) + P_1(x)P_2(y\,|\,x) - P_2(x,y)|$$

$$\geq \sum_{x,y} -|P_1(x,y) - P_1(x)P_2(y\,|\,x)| + |P_1(x)P_2(y\,|\,x) - P_2(x,y)|$$

$$= \sum_{x,y} -|P_1(x)(P_1(y\,|\,x) - P_2(y\,|\,x))| + |P_2(y\,|\,x)(P_1(x) - P_2(x))|$$

$$= \mathbb{E}_{P_1(x)} -\sum_y |P_1(y\,|\,x) - P_2(y\,|\,x)| + \sum_x |P_1(x) - P_2(x)|,$$

concluding our lemma. $\qquad\square$

**Lemma H.9.** Consider any two POMDP instances $\mathcal{P}$ and $\widehat{\mathcal{P}}$ and define the belief functions as $\boldsymbol{b}_{1:H}$ and $\widehat{\boldsymbol{b}}_{1:H}$, respectively (see Appendix C for the definition of belief functions). It holds that for any $\pi \in \Pi^{\mathrm{gen}}$

$$\left\|\mathbb{P}^{\pi,\mathcal{P}} - \mathbb{P}^{\pi,\widehat{\mathcal{P}}}\right\|_1 = \sum_{\overline{\tau}_H} |\mathbb{P}^{\pi,\mathcal{P}}(\overline{\tau}_H) - \mathbb{P}^{\pi,\widehat{\mathcal{P}}}(\overline{\tau}_H)|$$

$$\leq \|\mu_1 - \widehat{\mu}_1\|_1 + \mathbb{E}^{\mathcal{P}}_\pi \sum_{h\in[H-1]} \|\mathbb{T}_h(\cdot\,|\,s_h,a_h) - \widehat{\mathbb{T}}_h(\cdot\,|\,s_h,a_h)\|_1$$

$$+ \mathbb{E}^{\mathcal{P}}_\pi \sum_{h\in[H]} \|\mathbb{O}_h(\cdot\,|\,s_h) - \widehat{\mathbb{O}}_h(\cdot\,|\,s_h)\|_1,$$

$$\mathbb{E}^{\mathcal{P}}_\pi \|\boldsymbol{b}_h(\tau_h) - \widehat{\boldsymbol{b}}_h(\tau_h)\|_1 \leq 2\|\mu_1 - \widehat{\mu}_1\|_1 + 2\mathbb{E}^{\mathcal{P}}_\pi \sum_{h\in[H-1]} \|\mathbb{T}_h(\cdot\,|\,s_h,a_h) - \widehat{\mathbb{T}}_h(\cdot\,|\,s_h,a_h)\|_1$$

$$+ 2\mathbb{E}^{\mathcal{P}}_\pi \sum_{h\in[H]} \|\mathbb{O}_h(\cdot\,|\,s_h) - \widehat{\mathbb{O}}_h(\cdot\,|\,s_h)\|_1,$$

where we remind readers that we denote by $\overline{\tau}_H = (s_{1:H}, o_{1:H}, a_{1:H-1})$ the trajectory with states from an episode of the POMDP.

*Proof.* The first inequality also appears in [44] and we provide a simplified proof here for completeness. By Lemma H.8, it holds that

$$\sum_{\overline{\tau}_H} |\mathbb{P}^{\pi,\mathcal{P}}(\overline{\tau}_H) - \mathbb{P}^{\pi,\widehat{\mathcal{P}}}(\overline{\tau}_H)|$$

$$\leq \sum_{\overline{\tau}_{H-1}} |\mathbb{P}^{\pi,\mathcal{P}}(\overline{\tau}_{H-1}) - \mathbb{P}^{\pi,\widehat{\mathcal{P}}}(\overline{\tau}_{H-1})| + \mathbb{E}^{\mathcal{P}}_\pi \sum_{a_{H-1},s_H,o_H} \Big| \pi_{H-1}(a_{H-1}\,|\,\overline{\tau}_{H-1})\mathbb{T}_{H-1}(s_H\,|\,s_{H-1},a_{H-1})\mathbb{O}_H(o_H\,|\,s_H)$$

$$- \pi_{H-1}(a_{H-1}\,|\,\overline{\tau}_{H-1})\widehat{\mathbb{T}}_{H-1}(s_H\,|\,s_{H-1},a_{H-1})\widehat{\mathbb{O}}_H(o_H\,|\,s_H)\Big|$$

$$\leq \sum_{\overline{\tau}_{H-1}} |\mathbb{P}^{\pi,\mathcal{P}}(\overline{\tau}_{H-1}) - \mathbb{P}^{\pi,\widehat{\mathcal{P}}}(\overline{\tau}_{H-1})| + \mathbb{E}^{\mathcal{P}}_\pi \left[ \|\mathbb{T}_{H-1}(\cdot\,|\,s_{H-1},a_{H-1}) - \widehat{\mathbb{T}}_{H-1}(\cdot\,|\,s_{H-1},a_{H-1})\|_1 + \|\mathbb{O}_H(\cdot\,|\,s_H) - \widehat{\mathbb{O}}_H(\cdot\,|\,s_H)\|_1 \right],$$

where the last step is again from Lemma H.8. Therefore, by repeatedly unrolling the inequality, we proved the first result.

For the second result, we notice that by Lemma H.8, it holds that

$$\sum_{\tau_h,s_h} |\mathbb{P}_h^{\pi,\mathcal{P}}(\tau_h,s_h) - \mathbb{P}_h^{\pi,\widehat{\mathcal{P}}}(\tau_h,s_h)| \geq -\sum_{\tau_h} |\mathbb{P}_h^{\pi,\mathcal{P}}(\tau_h) - \mathbb{P}_h^{\pi,\widehat{\mathcal{P}}}(\tau_h)| + \mathbb{E}_\pi^{\mathcal{P}} \sum_{s_h} |\mathbb{P}_h^{\pi,\mathcal{P}}(s_h \,|\, \tau_h) - \mathbb{P}_h^{\pi,\widehat{\mathcal{P}}}(s_h \,|\, \tau_h)|.$$

Notice the fact that $\mathbb{P}_h^{\pi,\mathcal{P}}(s_h \,|\, \tau_h)$ does not depend on $\pi$ since it is exactly the belief $\boldsymbol{b}_h(\tau_h)(s_h)$, we conclude that

$$\mathbb{E}_\pi^{\mathcal{P}} \|\boldsymbol{b}_h(\tau_h) - \widehat{\boldsymbol{b}}_h(\tau_h)\|_1 \leq \sum_{\tau_h,s_h} |\mathbb{P}_h^{\pi,\mathcal{P}}(\tau_h,s_h) - \mathbb{P}_h^{\pi,\widehat{\mathcal{P}}}(\tau_h,s_h)| + \sum_{\tau_h} |\mathbb{P}_h^{\pi,\mathcal{P}}(\tau_h) - \mathbb{P}_h^{\pi,\widehat{\mathcal{P}}}(\tau_h)|$$

$$\leq 2 \sum_{\overline{\tau}_H} |\mathbb{P}^{\pi,\mathcal{P}}(\overline{\tau}_H) - \mathbb{P}^{\pi,\widehat{\mathcal{P}}}(\overline{\tau}_H)|, \tag{H.9}$$

where the last step comes from the fact that after marginalization, the total variation distance will not increase. By combining it with the first result, we proved the second result. $\qquad\square$

**Corollary H.10.** Consider any two POSG instances $\mathcal{G}, \widehat{\mathcal{G}}$ that satisfy Assumption C.8 and the corresponding belief functions in the form of $\mathbb{P}^{\mathcal{G}}, \mathbb{P}^{\widehat{\mathcal{G}}} : \mathcal{C}_h \to \Delta(\mathcal{P}_h \times \mathcal{S})$ for any $h \in [H]$, respectively. It holds that for any $\pi \in \Pi^{\mathrm{gen}}$

$$\mathbb{E}_\pi^{\mathcal{G}} \|\mathbb{P}^{\mathcal{G}}(\cdot,\cdot \,|\, c_h) - \mathbb{P}^{\widehat{\mathcal{G}}}(\cdot,\cdot \,|\, c_h)\|_1$$
$$\leq 2\|\mu_1 - \widehat{\mu}_1\|_1 + 2\mathbb{E}_\pi^{\mathcal{G}} \sum_{h\in[H-1]} \|\mathbb{T}_h(\cdot \,|\, s_h,a_h) - \widehat{\mathbb{T}}_h(\cdot \,|\, s_h,a_h)\|_1 + 2\mathbb{E}_\pi^{\mathcal{G}} \sum_{h\in[H]} \|\mathbb{O}_h(\cdot \,|\, s_h) - \widehat{\mathbb{O}}_h(\cdot \,|\, s_h)\|_1.$$

*Proof.* For the second result, we notice that by Lemma H.8, it holds that

$$\sum_{c_h,p_h,s_h} |\mathbb{P}_h^{\pi,\mathcal{G}}(c_h,p_h,s_h) - \mathbb{P}_h^{\pi,\widehat{\mathcal{G}}}(c_h,p_h,s_h)| \geq -\sum_{c_h} |\mathbb{P}_h^{\pi,\mathcal{P}}(c_h) - \mathbb{P}_h^{\pi,\widehat{\mathcal{G}}}(c_h)|$$
$$+ \mathbb{E}_\pi^{\mathcal{G}} \sum_{s_h,p_h} |\mathbb{P}_h^{\pi,\mathcal{G}}(s_h,p_h \,|\, c_h) - \mathbb{P}_h^{\pi,\widehat{\mathcal{G}}}(s_h,p_h \,|\, c_h)|.$$

Notice the fact that $\mathbb{P}_h^{\pi,\mathcal{G}}(s_h,p_h \,|\, c_h), \mathbb{P}_h^{\pi,\widehat{\mathcal{G}}}(s_h,p_h \,|\, c_h)$ do not depend on $\pi$ due to Assumption C.8, we conclude that

$$\mathbb{E}_\pi^{\mathcal{P}} \|\mathbb{P}^{\mathcal{G}}(\cdot,\cdot \,|\, c_h) - \mathbb{P}^{\widehat{\mathcal{G}}}(\cdot,\cdot \,|\, c_h)\|_1 \leq \sum_{c_h,p_h s_h} |\mathbb{P}_h^{\pi,\mathcal{G}}(c_h,p_h,s_h) - \mathbb{P}_h^{\pi,\widehat{\mathcal{G}}}(c_h,p_h,s_h)| + \sum_{c_h} |\mathbb{P}_h^{\pi,\mathcal{G}}(c_h) - \mathbb{P}_h^{\pi,\widehat{\mathcal{G}}}(c_h)|$$

$$\leq 2\sum_{\overline{\tau}_H} |\mathbb{P}^{\pi,\mathcal{G}}(\overline{\tau}_H) - \mathbb{P}^{\pi,\widehat{\mathcal{G}}}(\overline{\tau}_H)|, \tag{H.10}$$

where the last step again comes from the fact that after marginalization, the total variation distance will not increase. By combining it with Lemma H.9, we proved the second result. $\qquad\square$

**Lemma H.11.** For any reward function $r_{1:H}$ of $\mathcal{P}$ with $r_h : \mathcal{S} \times \mathcal{A} \to \mathbb{R}$ for any $h \in [H]$, it holds that

$$\max_{\pi\in\Pi^{\mathrm{gen}}} v^{\mathcal{P}}(\pi) \leq \max_{\pi\in\Pi_{\mathcal{S}}} v^{\mathcal{P}}(\pi).$$

*Proof.* Denote $\pi^\star \in \Pi_{\mathcal{S}}$ to be the optimal policy obtained by running value iteration only on the state space $\mathcal{S}$ for $\mathcal{P}$. Now we are ready to prove the following argument for any $\pi \in \Pi^{\mathrm{gen}}$ inductively:

$$Q_h^{\pi^\star,\mathcal{P}}(s_h,a_h) \geq Q_h^{\pi,\mathcal{P}}(s_{1:h},o_{1:h},a_{1:h}).$$

| | | Asymmetric optimistic NPG | Expert policy distillation | Asymmetric Q-learning | Vanilla AAC |
|---|---|---|---|---|---|
| Deterministic POMDP | Case 1 | $3.32 \pm 0.66$ | $\mathbf{3.33 \pm 0.62}$ | $3.04 \pm 0.58$ | $3.25 \pm 0.65$ |
| | Case 2 | $7.1 \pm 0.48$ | $\mathbf{7.26 \pm 0.68}$ | $6.15 \pm 0.85$ | $6.41 \pm 0.91$ |
| | Case 3 | $3.04 \pm 0.23$ | $\mathbf{3.25 \pm 0.33}$ | $3.09 \pm 0.39$ | $3.1 \pm 0.38$ |
| | Case 4 | $6.51 \pm 0.6$ | $\mathbf{6.54 \pm 0.58}$ | $6.16 \pm 0.48$ | $5.87 \pm 0.58$ |
| Block MDP | Case 1 | $3.31 \pm 0.46$ | $\mathbf{3.37 \pm 0.41}$ | $3.03 \pm 0.4$ | $3.08 \pm 0.43$ |
| | Case 2 | $6.36 \pm 0.52$ | $\mathbf{6.67 \pm 0.54}$ | $5.74 \pm 0.43$ | $5.7 \pm 0.46$ |
| | Case 3 | $3.2 \pm 0.26$ | $\mathbf{3.37 \pm 0.22}$ | $3.14 \pm 0.31$ | $2.97 \pm 0.32$ |
| | Case 4 | $6.01 \pm 0.32$ | $\mathbf{6.44 \pm 0.36}$ | $5.58 \pm 0.32$ | $5.33 \pm 0.19$ |

Table 2: Rewards of different approaches for POMDPs under the deterministic filter condition.

It is easy to see the argument holds for $h = H$. Fix state-action pair $(s_h, a_h) \in \mathcal{S} \times \mathcal{A}$ and trajectory $(s_{1:h-1}, o_{1:h}, a_{1:h-1})$, we note that:

$$
\begin{aligned}
Q_h^{\pi^\star, \mathcal{P}}(s_h, a_h) &= r_h(s_h, a_h) + \mathbb{E}_{s_{h+1} \sim \mathbb{T}_h(\cdot \,|\, s_h, a_h)} \left[ \max_{a_{h+1}} Q_{h+1}^{\pi^\star, \mathcal{P}}(s_{h+1}, a_{h+1}) \right] \\
&\geq r_h(s_h, a_h) + \mathbb{E}_{\substack{s_{h+1} \sim \mathbb{T}_h(\cdot \,|\, s_h, a_h), \\ o_{h+1} \sim \mathbb{O}_{h+1}(\cdot | s_{h+1})}} \left[ \max_{a_{h+1}} Q_{h+1}^{\pi, \mathcal{P}}(s_{1:h+1}, o_{1:h+1}, a_{1:h+1}) \right] \\
&\geq r_h(s_h, a_h) + \mathbb{E}_{\substack{s_{h+1} \sim \mathbb{T}_h(\cdot \,|\, s_h, a_h), \\ o_{h+1} \sim \mathbb{O}_{h+1}(\cdot | s_{h+1})}} \left[ \mathbb{E}_{a_{h+1} \sim \pi_{h+1}(\cdot \,|\, s_{1:h+1}, o_{1:h+1}, a_{1:h})} Q_{h+1}^{\pi, \mathcal{P}}(s_{1:h+1}, o_{1:h+1}, a_{1:h+1}) \right] \\
&= Q_h^{\pi, \mathcal{P}}(s_{1:h}, o_{1:h}, a_{1:h}),
\end{aligned}
$$

where the first inequality comes from the inductive hypothesis. $\qquad\square$

# I  Missing Details in Section 6

**POMDP under deterministic filter condition.** We first evaluate our algorithms on POMDPs with certain structures, i.e., the deterministic conditions. In particular, we generate POMDPs, where either the transition dynamics are deterministic or the emission ensures decodability. We test three of our approaches, expert policy distillation, asymmetric optimistic natural policy gradient. We summarize our results in Table 2, where the four cases corresponds to POMDPs with $(S = A = 2, O = 3, H = 5)$, $(S = A = 2, O = 3, H = 10)$, $(S = 3, A = 2, O = 4, H = 5)$, $(S = 3, A = 2, O = 4, H = 5)$, and we can see that our approach based on expert distillation outperforms all the other methods, which is consistent with the fact that such methods have exploited the special structures of the POMDPs achieving both polynomial sample and computational complexity.

**General POMDPs.** Here we also evaluate our methods for general randomly generated POMDPs without any structures. Hence, we compare the baselines with asymmetric optimistic natural policy gradient and asymmetric optimistic value iteration (i.e., the single-agent version of Algorithm 5). In Figure 2, we show the performance of different algorithms in POMDPs of different size, where the four cases corresponds to POMDPs with $(S = A = O = 2, H = 5)$, $(S = A = O = 2, H = 10)$, $(S = O = 3, A = 2, H = 10)$, $(S = A = 3, O = 2, H = 10)$, and our approaches achieves the highest rewards with small number of episodes.

**Implementation details.** For each problem setting, we generated 20 POMDPs randomly and report the average performance and its standard deviation for each algorithms. For our algorithms based on privileged value learning methods, we find that using a finite memory of 3 already provides strong performance. For our algorithms based privileged policy learning, we instantiate the MDP learning algorithm with the fully observable optimistic natural policy gradient algorithm [74]. Meanwhile, for both the decoder learning and belief learning, we find that just utilizing all the historic trajectories gives us reasonable performance without additional samples. For baselines, the hyperparameters $\alpha$ for $Q$-value update and step size for the policy update are tuned by grid search, where $\alpha$ controls the update of temporal difference learning (recall the update rule of temporal difference learning as

$Q \leftarrow (1 - \alpha)Q + \alpha Q^{\text{target}}$). For asymmetric $Q$-learning, we use $\epsilon$-greedy exploration, where we use the seminal decreasing rate $\epsilon_t = \frac{H+1}{H+t}$. Finally, all simulations are conducted on a personal laptop with Apple M1 CPU and 16 GB memory.

**Empirical insights and interpretation of the experimental results.** To understand intuitively why our approach outperforms those baseline algorithms, we notice the key difference in the value and policy update style between our approaches and vanilla asymmetric actor critic and asymmetric $Q$-learning. The baselines often roll-out the polices, collect trajectories, and only update the value and the policies on the states/history the trajectories have visited. Therefore, ideally, to learn a good policy for baselines, the number of trajectories to collect is at least as large as the history size, which is indeed exponential in the horizon $H$. This is known as curse of history for partially observable RL. In contrast, in our algorithms, we explicitly estimate the empirical transition and emissions, which is indeed of polynomial size. Thus, the sample complexity avoids suffering from the potential exponential dependency of horizon or the length of the finite memory. Finally, we acknowledge that the baselines are developed to handle complex, high-dimensional deep RL problems, while scaling our methods to deep RL benchmarks requires non-trivial engineering efforts.

# J  Missing Details in Section 7

**Proof of Proposition 7.1:**

Given the condition of Definition 3.2, the function $\phi_h$ that satisfies the condition of Proposition 7.1 can be constructed recursively as follows for any reachable $\tau_h$

$$\phi_h(\tau_h) := \psi_h(\phi_{h-1}(\tau_{h-1}), a_{h-1}, o_h),$$

and $\phi_1(o_1) := \psi_1(o_1)$. Therefore, we can prove by induction that by belief update rule

$$\mathbb{P}^{\mathcal{P}}(s_h \,|\, \tau_h) = U_h(b^{\phi_{h-1}(\tau_{h-1})}; a_{h-1}, o_h) = b^{\psi_h(\phi_{h-1}(\tau_{h-1}), a_{h-1}, o_h)},$$

where the last step is by the definition of our $\phi_h$. Therefore, we have $\mathbb{P}^{\mathcal{P}}(s_h = \psi_h(\phi_{h-1}(\tau_{h-1}), a_{h-1}, o_h) \,|\, \tau_h) = 1$.

For the other direction, it is similar to the proof of Lemma C.1 in [19] for $m$-step decodable POMDP. Here we prove it for completeness. For any reachable trajectory $\tau_h \in \mathcal{T}_h$, it holds by the belief update and induction that

$$\mathbb{P}^{\mathcal{P}}(s_h \,|\, \tau_h) = U_h(b^{\phi_{h-1}(\tau_{h-1})}, a_{h-1}, o_h) = \mathbb{P}^{\mathcal{P}}(s_h \,|\, s_{h-1} = \phi_{h-1}(\tau_{h-1}), a_{h-1}, o_h).$$

Meanwhile, since we know $\mathbb{P}^{\mathcal{P}}(\cdot \,|\, \tau_h)$ is a one-hot vector, we can construct $\psi_h$ such that $\psi_h(s_{h-1}, a_{h-1}, o_h)$ is the unique $s_h$ that makes $\mathbb{P}^{\mathcal{P}}(s_h \,|\, \tau_h) > 0$ with $s_{h-1} = \phi_{h-1}(\tau_{h-1})$. If this procedure does not complete the definition of $\psi$ for some $(s_{h-1}, a_{h-1}, o_h)$, it implies that either $s_{h-1}$ is not reachable or $o_h$ is not reachable given $s_{h-1}$, i.e., $\mathbb{P}^{\mathcal{P}}(o_h \,|\, s_{h-1}, a'_{h-1}) = 0$ for any $a'_{h-1} \in \mathcal{A}$, thus recovering the conditions of Definition 3.2. ∎

**Proof of Proposition 7.3:**

Note that for any given problem instance of a POMDP, we can construct a POSG by adding a dummy agent that has the observation being the exact state at each time step, and has only one dummy action that does not affect the transition or reward. Therefore, even the local private information $p_{i,h}$ of the dummy agent can decode the underlying state, and hence $(c_h, p_h)$ reveals the underlying state. Therefore, the corresponding POSG with the dummy agent satisfies the condition in Proposition 7.3. Meanwhile, it is direct to see that any CCE of the POSG is an optimal policy for the original POMDP. Now by the `PSPACE-hardness` of planning for POMDPs [66], we proved our proposition. ∎

**Theorem J.1.** Suppose the POSG $\mathcal{G}$ satisfies Definition 7.2. For any $\pi \in \Pi_{\mathcal{S}}$ and (potentially stochastic) decoding functions $\widehat{g} = \{\widehat{g}_{i,h}\}_{i \in [n], h \in [H]}$ with $\widehat{g}_{i,h} : \mathcal{C}_h \times \mathcal{P}_{i,h} \to \Delta(\mathcal{S})$ for each $i \in [n], h \in [H]$, it holds that

$$\text{NE/CCE-gap}(\pi^{\widehat{g}}) - \text{NE/CCE-gap}(\pi) \leq 2nH^2 \max_{i \in [n]} \max_{u_i \in \Pi_i} \max_{j \in [n], h \in [H]} \mathbb{P}^{u_i \times \pi_{-i}, \mathcal{G}}(s_h \neq \widehat{g}_{j,h}(c_h, p_{j,h}))$$

$$\text{CE-gap}(\pi^{\widehat{g}}) - \text{CE-gap}(\pi) \leq 2nH^2 \max_{i \in [n]} \max_{m_i \in \mathcal{M}_i} \max_{j \in [n], h \in [H]} \mathbb{P}^{(m_i \diamond \pi_i) \odot \pi_{-i}, \mathcal{G}}(s_h \neq \widehat{g}_{j,h}(c_h, p_{j,h}))$$

where $\pi^{\widehat{g}}$ is the distilled policy of $\pi$ through the decoding functions $\widehat{g}$, where at step $h$, each agent $i$ firstly individually decodes the state by sampling $s_h \sim \widehat{g}_{i,h}(\cdot \,|\, c_h, p_{i,h})$, and then acts according to the expert $\pi_{i,h}$, where in the following discussions, we slightly abuse the notation and regard $\widehat{g}_{i,h}(c_h, p_{i,h})$ as a random variable following the distribution of $\widehat{g}_{i,h}(\cdot \,|\, c_h, p_{i,h})$. In other words, the decoding process does not need correlations among the agents.

*Proof.* For notational simplicity, we write $v_i$ instead of $v_i^{\mathcal{G}}$ when the underlying model is clear from the context. Firstly, we consider any deterministic decoding function $\widehat{\phi} = \{\widehat{\phi}_{i,h}\}_{i \in [n], h \in [H]}$ with $\widehat{\phi}_{i,h} : \mathcal{C}_h \times \mathcal{P}_{i,h} \to \mathcal{S}$ for each $i \in [n], h \in [H]$, and note the following that for any $i \in [n]$,

$$v_i(\pi) - v_i(\pi^{\widehat{\phi}}) = \mathbb{E}_\pi^{\mathcal{G}}[R] - \mathbb{E}_{\pi^{\widehat{\phi}}}^{\mathcal{G}}[R]$$

$$= \mathbb{E}_\pi^{\mathcal{G}}[R\mathbb{1}[\forall j \in [n], h \in [H] : s_h = \widehat{\phi}_{j,h}(c_h, p_{j,h})]] - \mathbb{E}_{\pi^{\widehat{\phi}}}^{\mathcal{G}}[R\mathbb{1}[\forall j \in [n], h \in [H] : s_h = \widehat{\phi}_{j,h}(c_h, p_{j,h})]]$$

$$\quad + \mathbb{E}_\pi^{\mathcal{G}}[R\mathbb{1}[\exists j \in [n], h \in [H] : s_h \neq \widehat{\phi}_{j,h}(c_h, p_{j,h})]] - \mathbb{E}_{\pi^{\widehat{\phi}}}^{\mathcal{G}}[R\mathbb{1}[\exists j \in [n], h \in [H] : s_h \neq \widehat{\phi}_{j,h}(c_h, p_{j,h})]]$$

$$= \mathbb{E}_\pi^{\mathcal{G}}[R\mathbb{1}[\exists j \in [n], h \in [H] : s_h \neq \widehat{\phi}_{j,h}(c_h, p_{j,h})]] - \mathbb{E}_{\pi^{\widehat{\phi}}}^{\mathcal{G}}[R\mathbb{1}[\exists j \in [n], h \in [H] : s_h \neq \widehat{\phi}_{j,h}(c_h, p_{j,h})]],$$

where we define $R := \sum_h r_{i,h}(s_h, a_h)$, and the last step is by the definition of $\pi^{\widehat{\phi}}$

$$v_i(\pi) - v_i(\pi^{\widehat{\phi}}) \leq H\mathbb{P}^{\pi, \mathcal{G}}(\exists j \in [n], h \in [H] : s_h \neq \widehat{\phi}_{j,h}(c_h, p_{j,h})).$$

Therefore, by noticing the fact that $\widehat{g}$ is equivalent to a mixture of deterministic decoding functions, where at the beginning of each episode, one can firstly independently sample the outcome for each $(c_h, p_{j,h})$ for $j \in [n]$ and $h \in [H]$, we conclude that

$$v_i(\pi) - v_i(\pi^{\widehat{g}}) = v_i(\pi) - \mathbb{E}_{\widehat{\phi} \sim \widehat{g}} v_i(\pi^{\widehat{\phi}}) \leq H\mathbb{E}_{\widehat{\phi} \sim \widehat{g}}\mathbb{P}^{\pi, \mathcal{G}}(\exists j \in [n], h \in [H] : s_h \neq \widehat{\phi}_{j,h}(c_h, p_{j,h}))$$

$$= H\mathbb{P}^{\pi, \mathcal{G}}(\exists j \in [n], h \in [H] : s_h \neq \widehat{g}_{j,h}(c_h, p_{j,h})).$$

Now we prove our result for NE and CCE first. Due to similar arguments for evaluating $v_i(\pi) - v_i(\pi^{\widehat{\phi}})$, for any $u_i \in \Pi_i \cup \Pi_{\mathcal{S},i}$, it holds that

$$v_i(u_i \times \pi_{-i}^{\widehat{\phi}}) - v_i(u_i \times \pi_{-i}) \leq \mathbb{E}_{u_i \times \pi_{-i}^{\widehat{\phi}}}^{\mathcal{G}}[R\mathbb{1}[\exists j \in [n] \setminus \{i\}, h \in [H] : s_h \neq \widehat{\phi}_{j,h}(c_h, p_{j,h})]]$$

$$\leq H\mathbb{P}^{u_i \times \pi_{-i}^{\widehat{\phi}}, \mathcal{G}}(\exists j \in [n] \setminus \{i\}, h \in [H] : s_h \neq \widehat{\phi}_{j,h}(c_h, p_{j,h})).$$

We notice the following fact that

$$\mathbb{P}^{u_i \times \pi_{-i}, \mathcal{G}}(\forall j \in [n] \setminus \{i\}, h \in [H] : s_h = \widehat{\phi}_{j,h}(c_h, p_{j,h})) = \sum_{\overline{\tau}_H \in \overline{\mathcal{T}}_H(\widehat{\phi})} \mathbb{P}^{u_i \times \pi_{-i}, \mathcal{G}}(\overline{\tau}_H),$$

where we define $\overline{\mathcal{T}}_H(\widehat{\phi}) := \{\overline{\tau}_H \in \overline{\mathcal{T}}_H \,|\, \forall j \in [n] \setminus \{i\}, h \in [H] : s_h = \widehat{\phi}_{j,h}(c_h, p_{j,h})\}$. By definition of $\pi^{\widehat{\phi}}$, it holds that

$$\forall \overline{\tau}_H \in \overline{\mathcal{T}}_H(\widehat{\phi}) : \mathbb{P}^{u_i \times \pi_{-i}, \mathcal{G}}(\overline{\tau}_H) = \mathbb{P}^{u_i \times \pi_{-i}^{\widehat{\phi}}, \mathcal{G}}(\overline{\tau}_H).$$

Therefore, we have

$$\mathbb{P}^{u_i \times \pi_{-i}, \mathcal{G}}(\forall j \in [n] \setminus \{i\}, h \in [H] : s_h = \widehat{\phi}_{j,h}(c_h, p_{j,h})) = \mathbb{P}^{u_i \times \pi_{-i}^{\widehat{\phi}}, \mathcal{G}}(\forall j \in [n] \setminus \{i\}, h \in [H] : s_h = \widehat{\phi}_{j,h}(c_h, p_{j,h})).$$

Correspondingly, it holds that

$$\mathbb{P}^{u_i \times \pi_{-i}, \mathcal{G}}(\exists j \in [n] \setminus \{i\}, h \in [H] : s_h \neq \widehat{\phi}_{j,h}(c_h, p_{j,h})) = \mathbb{P}^{u_i \times \pi_{-i}^{\widehat{\phi}}, \mathcal{G}}(\exists j \in [n] \setminus \{i\}, h \in [H] : s_h \neq \widehat{\phi}_{j,h}(c_h, p_{j,h})),$$

which implies that

$$v_i(u_i \times \pi_{-i}^{\widehat{\phi}}) - v_i(u_i \times \pi_{-i}) \leq H\mathbb{P}^{u_i \times \pi_{-i}^{\widehat{\phi}}, \mathcal{G}}(\exists j \in [n] \setminus \{i\}, h \in [H] : s_h \neq \widehat{\phi}_{j,h}(c_h, p_{j,h}))$$

$$= H\mathbb{P}^{u_i \times \pi_{-i}, \mathcal{G}}(\exists j \in [n] \setminus \{i\}, h \in [H] : s_h \neq \widehat{\phi}_{j,h}(c_h, p_{j,h})).$$

Again by the fact that $\widehat{g}$ is equivalent to a mixture of deterministic decoding functions, it holds

$$v_i(u_i \times \pi_{-i}^{\widehat{g}}) - v_i(u_i \times \pi_{-i}) = \mathbb{E}_{\widehat{\phi} \sim \widehat{g}} v_i(u_i \times \pi_{-i}^{\widehat{\phi}}) - v_i(u_i \times \pi_{-i})$$
$$\leq H \mathbb{E}_{\widehat{\phi} \sim \widehat{g}} \mathbb{P}^{u_i \times \pi_{-i}, \mathcal{G}}(\exists j \in [n] \setminus \{i\}, h \in [H] : s_h \neq \widehat{\phi}_{j,h}(c_h, p_{j,h}))$$
$$= H \mathbb{P}^{u_i \times \pi_{-i}, \mathcal{G}}(\exists j \in [n] \setminus \{i\}, h \in [H] : s_h \neq \widehat{g}_{j,h}(c_h, p_{j,h})). \tag{J.1}$$

Now we are ready to evaluate the NE/CCE-gap of policy $\pi^{\widehat{g}}$ as follows:

$$\text{NE/CCE-gap}(\pi^{\widehat{g}}) - \text{NE/CCE-gap}(\pi)$$
$$\leq \max_{i \in [n]} \left( \max_{u_i \in \Pi_i} v_i(u_i \times \pi_{-i}^{\widehat{g}}) - \max_{u_i \in \Pi_{\mathcal{S},i}} v_i(u_i \times \pi_{-i}) \right) + H \mathbb{P}^{\pi, \mathcal{G}}(\exists j \in [n], h \in [H] : s_h \neq \widehat{g}_{j,h}(c_h, p_{j,h})).$$

Now we notice that $\max_{u_i \in \Pi_{\mathcal{S},i}} v_i(u_i \times \pi_{-i}) = \max_{u_i \in \Pi_i} v_i(u_i \times \pi_{-i})$ since $\Pi_{\mathcal{S},i} \subseteq \Pi_i$ by the deterministic filter condition Definition 3.2 and $\pi_{-i}$ is a Markov policy. Therefore, we conclude that

$$\text{NE/CCE-gap}(\pi^{\widehat{g}}) - \text{NE/CCE-gap}(\pi)$$
$$\leq \max_{i \in [n]} \left( \max_{u_i \in \Pi_i} v_i(u_i \times \pi_{-i}^{\widehat{g}}) - \max_{u_i \in \Pi_i} v_i(u_i \times \pi_{-i}) \right) + H \mathbb{P}^{\pi, \mathcal{G}}(\exists j \in [n], h \in [H] : s_h \neq \widehat{g}_{j,h}(c_h, p_{j,h}))$$
$$\leq \max_{i \in [n]} \left( \max_{u_i \in \Pi_i} \left( v_i(u_i \times \pi_{-i}^{\widehat{g}}) - v_i(u_i \times \pi_{-i}) \right) \right) + H \mathbb{P}^{\pi, \mathcal{G}}(\exists j \in [n], h \in [H] : s_h \neq \widehat{g}_{j,h}(c_h, p_{j,h}))$$
$$\leq \max_{i \in [n]} \left( \max_{u_i \in \Pi_i} H \mathbb{P}^{u_i \times \pi_{-i}, \mathcal{G}}(\exists j \in [n] \setminus \{i\}, h \in [H] : s_h \neq \widehat{g}_{j,h}(c_h, p_{j,h})) \right) + H \mathbb{P}^{\pi, \mathcal{G}}(\exists j \in [n], h \in [H] : s_h \neq \widehat{g}_{j,h}(c_h, p_{j,h}))$$
$$\leq 2H \max_{i \in [n], u_i \in \Pi_i} \sum_{j \in [n]} \sum_{h} \mathbb{P}^{u_i \times \pi_{-i}, \mathcal{G}}(s_h \neq \widehat{g}_{j,h}(c_h, p_{j,h}))$$
$$\leq 2n H^2 \max_{i \in [n], u_i \in \Pi_i, j \in [n], h \in [H]} \mathbb{P}^{u_i \times \pi_{-i}, \mathcal{G}}(s_h \neq \widehat{g}_{j,h}(c_h, p_{j,h})),$$

where the second last step is by a union bound, thus proving our result for NE and CCE.

For CE, consider any strategy modification $m_i \in \mathcal{M}_i \cup \mathcal{M}_{\mathcal{S},i}$, it holds that

$$\text{CE-gap}(\pi^{\widehat{g}}) - \text{CE-gap}(\pi)$$
$$\leq \max_{m_i \in \mathcal{M}_i} v_i((m_i \diamond \pi_i^{\widehat{g}}) \odot \pi_{-i}^{\widehat{g}}) - \max_{m_i \in \mathcal{M}_{\mathcal{S},i}} v_i((m_i \diamond \pi_i) \odot \pi_{-i}) + H \mathbb{P}^{\pi, \mathcal{G}}(\exists j \in [n], h \in [H] : s_h \neq \widehat{g}_{j,h}(c_h, p_{j,h})).$$

Now we notice that $\max_{m_i \in \mathcal{M}_{\mathcal{S},i}} v_i((m_i \diamond \pi_i) \odot \pi_{-i}) = \max_{m_i \in \mathcal{M}_{\mathcal{S},i}} v_i((m_i \diamond \pi_i) \odot \pi_{-i})$ since $\mathcal{M}_{\mathcal{S},i} \subseteq \mathcal{M}_i$ by the deterministic filter condition Definition 7.2 and Lemma J.2. Therefore, we conclude that

$$\text{CE-gap}(\pi^{\widehat{g}}) - \text{CE-gap}(\pi)$$
$$\leq \max_{i \in [n]} \max_{m_i \in \mathcal{M}_i} v_i((m_i \diamond \pi_i^{\widehat{g}}) \odot \pi_{-i}^{\widehat{g}}) - \max_{m_i \in \mathcal{M}_i} v_i((m_i \diamond \pi_i) \odot \pi_{-i}) + H \mathbb{P}^{\pi, \mathcal{G}}(\exists j \in [n], h \in [H] : s_h \neq \widehat{g}_{j,h}(c_h, p_{j,h}))$$
$$\leq \max_{i \in [n]} \max_{m_i \in \mathcal{M}_i} \left( v_i((m_i \diamond \pi_i^{\widehat{g}}) \odot \pi_{-i}^{\widehat{g}}) - v_i((m_i \diamond \pi_i) \odot \pi_{-i}) \right) + H \mathbb{P}^{\pi, \mathcal{G}}(\exists j \in [n], h \in [H] : s_h \neq \widehat{g}_{j,h}(c_h, p_{j,h}))$$
$$\leq \max_{i \in [n]} \max_{m_i \in \mathcal{M}_i} H \mathbb{P}^{(m_i \diamond \pi_i) \odot \pi_{-i}, \mathcal{G}}(\exists j \in [n] \setminus \{i\}, h \in [H] : s_h \neq \widehat{g}_{j,h}(c_h, p_{j,h}))$$
$$\qquad + H \mathbb{P}^{\pi, \mathcal{G}}(\exists j \in [n], h \in [H] : s_h \neq \widehat{g}_{j,h}(c_h, p_{j,h}))$$
$$\leq 2H \max_{i \in [n], m_i \in \mathcal{M}_i} \sum_{j \in [n]} \sum_{h} \mathbb{P}^{(m_i \diamond \pi_i) \odot \pi_{-i}, \mathcal{G}}(s_h \neq \widehat{g}_{j,h}(c_h, p_{j,h}))$$
$$\leq 2n H^2 \max_{i \in [n], m_i \in \mathcal{M}_i, h \in [H], j \in [n]} \mathbb{P}^{(m_i \diamond \pi_i) \odot \pi_{-i}, \mathcal{G}}(s_h \neq \widehat{g}_{j,h}(c_h, p_{j,h}))$$

where the third step is due to the same reason as Equation (J.1). $\qquad \square$

**Lemma J.2.** For any $\pi \in \Pi_{\mathcal{S}}$, it holds for any reward function and $i \in [n]$ that

$$\max_{m_i \in \mathcal{M}_i^{\text{gen}}} v_i((m_i \diamond \pi_i) \odot \pi_{-i}) = \max_{m_i \in \mathcal{M}_{\mathcal{S},i}} v_i((m_i \diamond \pi_i) \odot \pi_{-i}).$$

*Proof.* Denote $m_i^\star \in \text{argmax}_{m_i \in \mathcal{M}_i^{\text{gen}}} v_i((m_i \diamond \pi_i) \odot \pi_{-i})$ and $\widehat{m}_i^\star \in \text{argmax}_{m_i \in \mathcal{M}_{\mathcal{S},i}} v_i((m_i \diamond \pi_i) \odot \pi_{-i})$. Now we shall prove that $V_{i,h}^{(m_i^\star \diamond \pi_i) \odot \pi_{-i}, \mathcal{G}}(s_{1:h}, o_{1:h}, a_{1:h-1}) \leq V_{i,h}^{(\widehat{m}_i^\star \diamond \pi_i) \odot \pi_{-i}, \mathcal{G}}(s_h)$ inductively for each $h \in [H]$. Note that it holds for $h = H + 1$. Now we consider the following

$$V_{i,h}^{(m_i^\star \diamond \pi_i) \odot \pi_{-i}, \mathcal{G}}(s_{1:h}, o_{1:h}, a_{1:h-1})$$

$$= \mathbb{E}_{\substack{a_h \sim \pi_h(\cdot \mid s_h) \\ s_{h+1} \sim \mathbb{T}_h(\cdot \mid s_h, m_{i,h}^\star(s_{1:h}, o_{1:h}, a_{1:h-1}, a_{i,h}), a_{-i,h}) \\ o_{h+1} \sim \mathbb{O}_{h+1}(\cdot \mid s_{h+1})}} \left\{ r_h(s_h, m_{i,h}^\star(s_{1:h}, o_{1:h}, a_{1:h-1}, a_{i,h}), a_{-i,h}) \right.$$

$$\left. + V_{i,h+1}^{(m_i^\star \diamond \pi_i) \odot \pi_{-i}, \mathcal{G}}(s_{1:h+1}, o_{1:h+1}, a_{1:h-1}, m_{i,h}^\star(s_{1:h}, o_{1:h}, a_{1:h-1}, a_{i,h}), a_{-i,h}) \right\}$$

$$\leq \mathbb{E}_{\substack{a_h \sim \pi_h(\cdot \mid s_h) \\ s_{h+1} \sim \mathbb{T}_h(\cdot \mid s_h, m_{i,h}^\star(s_{1:h}, o_{1:h}, a_{1:h-1}, a_{i,h}), a_{-i,h}) \\ o_{h+1} \sim \mathbb{O}_{h+1}(\cdot \mid s_{h+1})}} \left\{ r_h(s_h, m_{i,h}^\star(s_{1:h}, o_{1:h}, a_{1:h-1}, a_{i,h}), a_{-i,h}) + V_{i,h+1}^{(\widehat{m}_i^\star \diamond \pi_i) \odot \pi_{-i}, \mathcal{G}}(s_{h+1}) \right\}$$

$$\leq \mathbb{E}_{\substack{a_h \sim \pi_h(\cdot \mid s_h) \\ s_{h+1} \sim \mathbb{T}_h(\cdot \mid s_h, \widehat{m}_{i,h}^\star(s_h, a_{i,h}), a_{-i,h}) \\ o_{h+1} \sim \mathbb{O}_{h+1}(\cdot \mid s_{h+1})}} \left[ r_h(s_h, \widehat{m}_{i,h}^\star(s_h, a_{i,h}), a_{-i,h}) + V_{i,h+1}^{(\widehat{m}_i^\star \diamond \pi_i) \odot \pi_{-i}, \mathcal{G}}(s_{h+1}) \right]$$

$$= V_{i,h}^{(\widehat{m}_i^\star \diamond \pi_i) \odot \pi_{-i}, \mathcal{G}}(s_h),$$

where the second inequality follows from the inductive hypothesis and the third step is by the definition of $\widehat{m}_i^\star \in \text{argmax}_{m_i \in \mathcal{M}_{\mathcal{S},i}} v_i((m_i \diamond \pi_i) \odot \pi_{-i})$. $\square$

**Lemma J.3.** Given an approximate POSG $\widehat{\mathcal{G}}$ that satisfies Assumption C.8 with approximate transitions and emissions being $\{\widehat{\mathbb{T}}_h, \widehat{\mathbb{O}}_h\}_{h \in [H]}$, we define the approximate decoding function $\widehat{g}$ to be

$$\widehat{g}_{i,h}(s_h \mid c_h, p_{i,h}) := \mathbb{P}^{\widehat{\mathcal{G}}}(s_h \mid c_h, p_{i,h}),$$

for each $h \in [H], s_h \in \mathcal{S}, c_h \in \mathcal{C}_h, p_{i,h} \in \mathcal{P}_{i,h}$. Then it holds that for any $\pi \in \Pi_{\mathcal{S}}$,

$$\max_{i \in [n], u_i \in \Pi_i, j \in [n], h \in [H]} \mathbb{P}^{u_i \times \pi_{-i}, \mathcal{G}}(s_h \neq \widehat{g}_{j,h}(c_h, p_{j,h}))$$

$$\leq \max_{i \in [n], u_i \in \Pi_{\mathcal{S},i}} \mathbb{E}_{u_i \times \pi_{-i}}^{\mathcal{G}} \left[ \sum_{h \in [H-1]} \|\mathbb{T}_h(\cdot \mid s_h, a_h) - \widehat{\mathbb{T}}_h(\cdot \mid s_h, a_h)\|_1 + \sum_{h \in [H]} \|\mathbb{O}_h(\cdot \mid s_h) - \widehat{\mathbb{O}}_h(\cdot \mid s_h)\|_1 \right],$$

and

$$\max_{i \in [n], m_i \in \mathcal{M}_i, j \in [n], h \in [H]} \mathbb{P}^{(m_i \diamond \pi_i) \odot \pi_{-i}, \mathcal{G}}(s_h \neq \widehat{g}_{j,h}(c_h, p_{j,h}))$$

$$\leq \max_{i \in [n], m_i \in \mathcal{M}_{\mathcal{S},i}} \mathbb{E}_{(m_i \diamond \pi_i) \odot \pi_{-i}}^{\mathcal{G}} \left[ \sum_{h \in [H]} \|\mathbb{T}_h(\cdot \mid s_h, a_h) - \widehat{\mathbb{T}}_h(\cdot \mid s_h, a_h)\|_1 + \sum_{h \in [H-1]} \|\mathbb{O}_h(\cdot \mid s_h) - \widehat{\mathbb{O}}_h(\cdot \mid s_h)\|_1 \right].$$

*Proof.* Note for any $i \in [n], u_i \in \Pi_i, j \in [n], h \in [H]$, it holds

$$\mathbb{P}^{u_i \times \pi_{-i}, \mathcal{G}}(s_h \neq \widehat{g}_{j,h}(c_h, p_{j,h})) = \frac{1}{2} \mathbb{E}_{u_i \times \pi_{-i}}^{\mathcal{G}} \sum_{s_h} \left| \mathbb{P}^{\mathcal{G}}(s_h \mid c_h, p_{j,h}) - \mathbb{P}^{\widehat{\mathcal{G}}}(s_h \mid c_h, p_{j,h}) \right|,$$

due to the condition in Definition 7.2. Meanwhile,

$$\frac{1}{2}\mathbb{E}^{\mathcal{G}}_{u_i \times \pi_{-i}} \sum_{s_h} \left| \mathbb{P}^{\mathcal{G}}(s_h \mid c_h, p_{j,h}) - \mathbb{P}^{\widehat{\mathcal{G}}}(s_h \mid c_h, p_{j,h}) \right|$$

$$\leq \sum_{s_h, c_h, p_{j,h}} \left| \mathbb{P}^{u_i \times \pi_{-i}, \mathcal{G}}(s_h, c_h, p_{j,h}) - \mathbb{P}^{u_i \times \pi_{-i}, \widehat{\mathcal{G}}}(s_h, c_h, p_{j,h}) \right|$$

$$\leq \sum_{\overline{\tau}_h} \left| \mathbb{P}^{u_i \times \pi_{-i}, \mathcal{G}}(\overline{\tau}_h) - \mathbb{P}^{u_i \times \pi_{-i}, \widehat{\mathcal{G}}}(\overline{\tau}_h) \right|$$

$$\leq \sum_{\overline{\tau}_H} \left| \mathbb{P}^{u_i \times \pi_{-i}, \mathcal{G}}(\overline{\tau}_H) - \mathbb{P}^{u_i \times \pi_{-i}, \widehat{\mathcal{G}}}(\overline{\tau}_H) \right|$$

$$\leq \mathbb{E}^{\mathcal{G}}_{u_i \times \pi_{-i}} \left[ \sum_{h \in [H-1]} \|\mathbb{T}_h(\cdot \mid s_h, a_h) - \widehat{\mathbb{T}}_h(\cdot \mid s_h, a_h)\|_1 + \sum_{h \in [H]} \|\mathbb{O}_h(\cdot \mid s_h) - \widehat{\mathbb{O}}_h(\cdot \mid s_h)\|_1 \right],$$

where the first inequality is by the first inequality in Lemma J.7, the second and third inequalities are due to the fact that TV distance does not increase after marginalization, and the last inequality is by Lemma H.9. Since $\pi_{-i}$ is a fixed and fully-observable Markov policy, by Lemma H.11, we have

$$\mathbb{E}^{\mathcal{G}}_{u_i \times \pi_{-i}} \left[ \sum_{h \in [H-1]} \|\mathbb{T}_h(\cdot \mid s_h, a_h) - \widehat{\mathbb{T}}_h(\cdot \mid s_h, a_h)\|_1 + \sum_{h \in [H]} \|\mathbb{O}_h(\cdot \mid s_h) - \widehat{\mathbb{O}}_h(\cdot \mid s_h)\|_1 \right]$$

$$\leq \max_{u_i \in \Pi_{\mathcal{S},i}} \mathbb{E}^{\mathcal{G}}_{u_i \times \pi_{-i}} \left[ \sum_{h \in [H-1]} \|\mathbb{T}_h(\cdot \mid s_h, a_h) - \widehat{\mathbb{T}}_h(\cdot \mid s_h, a_h)\|_1 + \sum_{h \in [H]} \|\mathbb{O}_h(\cdot \mid s_h) - \widehat{\mathbb{O}}_h(\cdot \mid s_h)\|_1 \right],$$

thus proving the first result of our lemma.

For the second result of our lemma, it can be proved similarly that for any $i \in [n], m_i \in \mathcal{M}_i, j \in [n], h \in [H]$,

$$\mathbb{P}^{(m_i \diamond \pi_i) \odot \pi_{-i}, \mathcal{G}}(s_h \neq \widehat{g}_{j,h}(c_h, p_{j,h}))$$

$$\leq \mathbb{E}^{\mathcal{G}}_{(m_i \diamond \pi_i) \odot \pi_{-i}} \left[ \sum_{h \in [H-1]} \|\mathbb{T}_h(\cdot \mid s_h, a_h) - \widehat{\mathbb{T}}_h(\cdot \mid s_h, a_h)\|_1 + \sum_{h \in [H]} \|\mathbb{O}_h(\cdot \mid s_h) - \widehat{\mathbb{O}}_h(\cdot \mid s_h)\|_1 \right].$$

By Lemma J.2, we proved the second result. □

**Theorem J.4.** Fix any $\epsilon, \delta \in (0,1)$ and $\pi \in \Pi_{\mathcal{S}}$. Algorithm 5 can learn a decoding function $\widehat{g}$ such that with probability $1 - \delta$

$$\max_{i \in [n], u_i \in \Pi_i, j \in [n], h \in [H]} \mathbb{P}^{u_i \times \pi_{-i}, \mathcal{G}}(s_h \neq \widehat{g}_{j,h}(c_h, p_{j,h})) \leq \epsilon,$$

with total sample complexity $\widetilde{\mathcal{O}}(\frac{nS^2 AHO + nS^3 AH}{\epsilon^2} + \frac{S^4 A^2 H^5}{\epsilon})$ and computational complexity $\text{POLY}(S, A, H, O, \frac{1}{\epsilon}), \log \frac{1}{\delta}$.

*Proof.* With the help of Lemma J.3, it suffices to prove

$$\max_{i \in [n], u_i \in \Pi_{\mathcal{S},i}} \mathbb{E}^{\mathcal{G}}_{u_i \times \pi_{-i}} \left[ \sum_{h \in [H]} \|\mathbb{T}_h(\cdot \mid s_h, a_h) - \widehat{\mathbb{T}}_h(\cdot \mid s_h, a_h)\|_1 + \|\mathbb{O}_h(\cdot \mid s_h) - \widehat{\mathbb{O}}_h(\cdot \mid s_h)\|_1 \right] \leq \epsilon.$$

The following proof procedure follows similarly to that of Theorem H.5. For each $h \in [H]$ and $s_h \in \mathcal{S}$, we define

$$p_h(s_h) = \max_{i \in [n], u_i \in \Pi_{\mathcal{S},i}} d_h^{u_i \times \pi_{-i}}(s_h).$$

Fix $\epsilon_1, \delta_1 > 0$, we define $\mathcal{U}(h, \epsilon_1) = \{s_h \in \mathcal{S} \mid p_h(s_h) \geq \epsilon_1\}$. By [37], one can learn a policy $\{\Psi_i(h, s_h)\}_{i \in [n]}$ with sample complexity $\widetilde{\mathcal{O}}(\frac{S^2 A_i H^4}{\epsilon_1})$ such that $\max_{i \in [n]} d_h^{\Psi_i(h, s_h) \times \pi_{-i}}(s_h) \geq$

$\frac{p_h(s_h)}{2}$ for each $s_h \in \mathcal{U}(h, \epsilon_1)$ with probability $1 - n \cdot \delta_1$. Now we assume this event holds for any $h \in [H]$ and $s_h \in \mathcal{U}(h, \epsilon_1)$. For each $s_h \in \mathcal{S}$ and $a_h \in \mathcal{A}$, we have executed each policy $\{\Psi_i(h, s_h) \times \pi_{-i}\}_{i \in [n]}$ for the first $h - 1$ steps followed by an action $a_h \in \mathcal{A}$ for $N$ episodes and denote the total number of episodes that $s_h$ and $a_h$ are visited as $N_h(s_h, a_h)$, and $N_h(s_h) = \sum_{a \in \mathcal{A}} N_h(s_h, a)$. Then, with probability $1 - e^{-N\epsilon_1/8}$, we have $N_h(s_h, a_h) \geq \frac{Np_h(s_h)}{2}$ by Chernoff bound. Now conditioned on this event, we are ready to evaluate the following for any $i \in [n]$, and $u_i \in \Pi_{\mathcal{S},i}$:

$$\mathbb{E}^{\mathcal{G}}_{u_i \times \pi_{-i}} \|\mathbb{T}_h(\cdot \mid s_h, a_h) - \widehat{\mathbb{T}}_h(\cdot \mid s_h, a_h)\|_1 = \sum_{s_h, a_h} d_h^{u_i \times \pi_{-i}}(s_h)(u_i \times \pi_{-i})_h(a_h \mid s_h)\|\mathbb{T}_h(\cdot \mid s_h, a_h) - \widehat{\mathbb{T}}_h(\cdot \mid s_h, a_h)\|_1$$

$$\leq 2 \cdot S\epsilon_1 + \sum_{s_h \in \mathcal{U}(h,\epsilon_1), a_h} d_h^{u_i \times \pi_{-i}}(s_h)[u_i \times \pi_{-i}]_h(a_h \mid s_h)\sqrt{\frac{S\log(1/\delta_2)}{N_h(s_h, a_h)}}$$

$$\leq 2 \cdot S\epsilon_1 + \sum_{s_h \in \mathcal{U}(h,\epsilon_1), a_h} d_h^{u_i \times \pi_{-i}}(s_h)[u_i \times \pi_{-i}]_h(a_h \mid s_h)\sqrt{\frac{2S\log(1/\delta_2)}{Np_h(s_h)}}$$

$$\leq 2 \cdot S\epsilon_1 + \sum_{s_h} \sqrt{d_h^{u_i \times \pi_{-i}}(s_h)}\sqrt{\frac{2S\log(1/\delta_2)}{N}}$$

$$\leq 2 \cdot S\epsilon_1 + S\sqrt{\frac{2\log(1/\delta_2)}{N}},$$

where the second step is by Lemma J.8, and the last step is by Cauchy-Schwarz inequality. Similarly,

$$\mathbb{E}^{\mathcal{G}}_{u_i \times \pi_{-i}} \|\mathbb{O}_h(\cdot \mid s_h) - \widehat{\mathbb{O}}_h(\cdot \mid s_h)\|_1 = \sum_{s_h} d_h^{u_i \times \pi_{-i}}(s_h)\|\mathbb{O}_h(\cdot \mid s_h) - \widehat{\mathbb{O}}_h(\cdot \mid s_h)\|_1$$

$$\leq 2 \cdot S\epsilon_1 + \sum_{s_h \in \mathcal{U}(h,\epsilon_1)} d_h^{u_i \times \pi_{-i}}(s_h)\sqrt{\frac{O\log(1/\delta_2)}{N_h(s_h)}}$$

$$\leq 2 \cdot S\epsilon_1 + \sum_{s_h \in \mathcal{U}(h,\epsilon_1)} d_h^{u_i \times \pi_{-i}}(s_h)\sqrt{\frac{2O\log(1/\delta_2)}{Np_h(s_h)}}$$

$$\leq 2 \cdot S\epsilon_1 + \sum_{s_h \in \mathcal{U}(h,\epsilon_1)} \sqrt{d_h^{u_i \times \pi_{-i}}(s_h)}\sqrt{\frac{2O\log(1/\delta_2)}{N}}$$

$$\leq 2 \cdot S\epsilon_1 + \sqrt{\frac{2SO\log(1/\delta_2)}{N}},$$

where the second step is by Lemma J.9, and the last step is by Cauchy-Schwarz inequality. Therefore, by a union bound, all high probability events hold with probability

$$1 - SHn\delta_1 - SHAe^{-N\epsilon_1/8} - 2SAH\delta_2.$$

Therefore, we can choose $N = \widetilde{\Theta}(\frac{S^2 + SO}{\epsilon^2})$ and $\epsilon_1 = \Theta(\frac{\epsilon}{S})$, leading to the total sample complexity

$$SHA\left(nN + \widetilde{\Theta}\left(\frac{S^3 AH^4}{\epsilon}\right)\right) = \widetilde{\Theta}\left(\frac{nS^2 AHO + nS^3 AH}{\epsilon^2} + \frac{S^4 A^2 H^5}{\epsilon}\right),$$

which completes the proof. $\qquad \square$

Note that although our Algorithm 5 and Theorem J.4 are stated for NE/CCE, it can also handle CE with simple modifications, where the key observation is that the strategy modification $m_i \in \mathcal{M}_{\mathcal{S},i}$ can also be regarded as a Markov policy in an *extended* MDP marginalized by $\pi_{-i}$ as defined below.

**Definition J.5.** Fix $\pi \in \Pi_{\mathcal{S}}$. We define $\mathcal{M}_i^{\text{extended}}(\pi)$ to be an MDP for agent $i$, where for each $h \in [H]$, the state is $(s_h, a_{i,h})$, the action is some modified action $a'_{i,h}$, the transition is defined as $\mathbb{T}_h^{\text{extended}}(s_{h+1}, a_{i,h+1} \mid s_h, a_{i,h}, a'_{i,h}) :=$

$\mathbb{E}_{a_{-i,h} \sim \pi_h(\cdot \,|\, s_h, a_{i,h})}[\mathbb{T}_h(s_{h+1} \,|\, s_h, a'_{i,h}, a_{-i,h})\pi_{h+1}(a_{i,h+1} \,|\, s_{h+1})]$, where we slightly abuse the notation of $\pi_h(a_{-i,h} \,|\, s_h, a_{i,h})$ and $\pi_h(a_{i,h} \,|\, s_h)$ by defining them as the posterior and marginal distributions induced by the joint distribution $\pi_h(a_h \,|\, s_h)$. Similarly, the reward is given by $r_h^{\text{extended}}(s_h, a_{i,h}, a'_{i,h}) := \mathbb{E}_{a_{-i,h} \sim \pi_h(\cdot \,|\, s_h, a_{i,h})}[r_h(s_h, a'_{i,h}, a_{-i,h})]$.

With the help of such an extended MDP, we can develop Algorithm 6, which is a CE version of Algorithm 5 with the following guarantees.

**Theorem J.6.** Fix any $\epsilon, \delta \in (0,1)$ and $\pi \in \Pi_{\mathcal{S}}$. Algorithm 6 can learn a decoding function $\widehat{g}$ such that

$$\max_{i \in [n], m_i \in \mathcal{M}_i, j \in [n], h \in [H]} \mathbb{P}^{(m_i \diamond \pi_i) \odot \pi_{-i}, \mathcal{G}}(s_h \neq \widehat{g}_{j,h}(c_h, p_{j,h})) \leq \epsilon,$$

with total sample complexity $\widetilde{\mathcal{O}}\left(\frac{nS^2 A^3 HO + nS^3 A^4 H}{\epsilon^2} + \frac{S^4 A^6 H^5}{\epsilon}\right)$ and computational complexity $\text{POLY}(S, A, H, O, \frac{1}{\epsilon})$.

*Proof.* Due to the construction of $\mathcal{M}_i^{\text{extended}}(\pi)$, the proof of Theorem J.4 readily applies, where the only difference is that the state space of $\mathcal{M}_i^{\text{extended}}(\pi)$ is now $SA_i$, larger than that of $\mathcal{M}(\pi_{-i})$ by a factor of $A_i$ thus proving our theorem. $\square$

We next introduce and prove several supporting lemmas used before.

**Lemma J.7.** Suppose we can sample from a joint distribution $P \in \Delta(\mathcal{X} \times \mathcal{Y})$ for some finite $\mathcal{X}$, $\mathcal{Y}$ i.i.d. Then we can learn an approximate distribution $Q \in \Delta(\mathcal{X} \times \mathcal{Y})$ with sample complexity $\Theta\left(\frac{|\mathcal{X}||\mathcal{Y}| + \log 1/\delta}{\epsilon^2}\right)$ such that

$$\mathbb{E}_{x \sim P} \sum_{y \in \mathcal{Y}} |P(y \,|\, x) - Q(y \,|\, x)| \leq 2 \sum_{x \in \mathcal{X}, y \in \mathcal{Y}} |P(x,y) - Q(x,y)| \leq \epsilon,$$

with probability $1 - \delta$.

*Proof.* Note the following holds

$$\sum_{x \in \mathcal{X}, y \in \mathcal{Y}} |P(x,y) - Q(x,y)| = \sum_{x \in \mathcal{X}, y \in \mathcal{Y}} |P(x,y) - P(x)Q(y \,|\, x) + P(x)Q(y \,|\, x) - Q(x,y)|$$

$$\geq \sum_{x \in \mathcal{X}, y \in \mathcal{Y}} |P(x,y) - P(x)Q(y \,|\, x)| - |P(x)Q(y \,|\, x) - Q(x,y)|.$$

Therefore, we have

$$\mathbb{E}_{x \sim P} \sum_{y \in \mathcal{Y}} |P(y \,|\, x) - Q(y \,|\, x)| \leq \sum_{x \in \mathcal{X}, y \in \mathcal{Y}} |P(x,y) - Q(x,y)| + \sum_{x \in \mathcal{X}, y \in \mathcal{Y}} Q(y \,|\, x)|P(x) - Q(x)|$$

$$\leq 2 \sum_{x \in \mathcal{X}, y \in \mathcal{Y}} |P(x,y) - Q(x,y)|. \tag{J.2}$$

By the sample complexity of learning discrete distributions [13], we can learn $Q$ such that $\sum_{x \in \mathcal{X}, y \in \mathcal{Y}} |P(x,y) - Q(x,y)| \leq \epsilon$ in sample complexity $\Theta\left(\frac{|\mathcal{X}||\mathcal{Y}| + \log 1/\delta}{\epsilon^2}\right)$ with probability $1 - \delta$. Thus, we proved our lemma. $\square$

**Lemma J.8** (Concentration on transition). Fix $\delta > 0$ and dataset $\{\bar{\tau}_H^k\}_{k \in [N]}$ sampled from $\mathcal{P}$ (or $\mathcal{G}$ in the multi-agent setting) under policy $\pi \in \Pi^{\text{gen}}$. We define for each $h \in [H]$, $(s_h, a_h, s_{h+1}) \in \mathcal{S} \times \mathcal{A} \times \mathcal{S}$

$$N_h(s_h, a_h) = \sum_{k \in [N]} \mathbb{1}[s_h^k = s_h, a_h^k = a_h], \qquad N_h(s_h, a_h, s_{h+1}) = \sum_{k \in [N]} \mathbb{1}[s_h^k = s_h, a_h^k = a_h, s_{h+1}^k = s_h].$$

Then, with probability at least $1 - \delta$, it holds that for any $k \in [K], h \in [H], s_h \in \mathcal{S}, a_h \in \mathcal{A}$:

$$\|\mathbb{T}_h(\cdot \,|\, s_h, a_h) - \widehat{\mathbb{T}}_h(\cdot \,|\, s_h, a_h)\|_1 \leq C_1 \sqrt{\frac{S \log(SAHK/\delta)}{\max\{N_h(s_h, a_h), 1\}}},$$

for some absolute constant $C_1 > 0$, where we define $\widehat{\mathbb{T}}_h(s_{h+1} \,|\, s_h, a_h) = \frac{N_h(s_h, a_h, s_{h+1})}{\max\{N_h(s_h, a_h), 1\}}$.

*Proof.* This is done by firstly bounding $\|\mathbb{T}_h(\cdot \mid s_h, a_h) - \widehat{\mathbb{T}}_h(\cdot \mid s_h, a_h)\|_1$ for specific $k, h, s_h, a_h$ according to [13], and then taking union bound for all $k \in [K], h \in [H], s_h \in \mathcal{S}, a_h \in \mathcal{A}$. $\qquad \square$

**Lemma J.9** (Concentration on emission). Fix $\delta > 0$ and dataset $\{\bar{\tau}_H^k\}_{k \in [N]}$ sampled from $\mathcal{P}$ (or $\mathcal{G}$ in the multi-agent setting) under some policy $\pi \in \Pi^{\text{gen}}$. We define for each $h \in [H], (s_h, o_h) \in \mathcal{S} \times \mathcal{O}$

$$N_h(s_h, o_h) = \sum_{k \in [N]} \mathbb{1}[s_h^k = s_h, o_h^k = o_h], \qquad N_h(s_h) = \sum_{k \in [N]} \mathbb{1}[s_h^k = s_h].$$

Then, with probability at least $1 - \delta$, it holds that

$$\|\mathbb{O}_h(\cdot \mid s_h) - \widehat{\mathbb{O}}_h(\cdot \mid s_h)\|_1 \le C_2 \sqrt{\frac{O \log(SHK/\delta)}{\max\{N_h^k(s_h), 1\}}},$$

for some absolute constant $C_2 > 0$, where we define $\widehat{\mathbb{O}}_h(o_h \mid s_h) = \frac{N_h(s_h, o_h)}{\max\{N_h(s_h), 1\}}$.

*Proof.* This is done by firstly bounding $\|\mathbb{O}_h(\cdot \mid s_h) - \widehat{\mathbb{O}}_h(\cdot \mid s_h)\|_1$ for specific $k, h, s_h$ according to [13], and then taking union bound for all $k \in [K], h \in [H], s_h \in \mathcal{S}$. $\qquad \square$

Now we switch to proving the guarantees for Algorithm 7.

**Lemma J.10.** Fix $\delta > 0$. With probability $1 - \delta$, it holds that for any $k \in [K], h \in [H], s_h \in \mathcal{S}$:

$$\sum_{o_{h+1}} \left| \mathbb{P}^{\mathcal{G}}(o_{h+1} \mid s_h, a_h) - \widehat{\mathbb{J}}_h^k(o_{h+1} \mid s_h, a_h) \right| \le C_3 \sqrt{\frac{O \log(SHKA/\delta)}{N_h^k(s_h, a_h)}},$$

where $\widehat{\mathbb{J}}_h^k$ is defined in Algorithm 7.

*Proof.* This is done by firstly bounding $\sum_{o_{h+1}} \left| \mathbb{P}^{\mathcal{G}}(o_{h+1} \mid s_h, a_h) - \widehat{\mathbb{J}}_h^k(o_{h+1} \mid s_h, a_h) \right|$ for specific $k, h, s_h, a_h$ according to [13], and then taking union bound for all $k \in [K], h \in [H], s_h \in \mathcal{S}, a_h \in \mathcal{A}$. $\qquad \square$

From now on, we shall use the bonus

$$b_h^k(s_h, a_h) = \min \left\{ C_3(H - h)\sqrt{\frac{O \log(SAHK/\delta)}{\max\{N_h^k(s_h, a_h), 1\}}}, 2(H - h) \right\} \qquad \text{(J.3)}$$

in Algorithm 7, for some absolute constant $C_3 > 0$.

Before presenting our technical analysis, we define the following notation for the ease of presentation. We define the following approximate value functions for any policy $\pi \in \Pi$ in a backward way for $h \in [H]$ when given some approximate belief in the form of $\{\widehat{P}_h : \widehat{\mathcal{C}}_h \to \Delta(\mathcal{P}_h \times \mathcal{S})\}_{h \in [H]}$ as discussed in Section 7.2:

$$\widehat{V}_{i,h}^{\pi,\mathcal{G}}(c_h) := \mathbb{E}_{s_h, p_h \sim \widehat{P}_h(\cdot, \cdot \mid \widehat{c}_h)} \mathbb{E}_{\omega_h, \{a_{j,h} \sim \pi_{j,h}(\cdot \mid \omega_{j,h}, c_h, p_{j,h})\}_{j \in [n]}}$$
$$\mathbb{E}_{s_{h+1} \sim \mathbb{T}_h(\cdot \mid s_h, a_h), o_{h+1} \sim \mathbb{O}_{h+1}(\cdot \mid s_{h+1})} \left[ r_{i,h}(s_h, a_h) + V_{i,h+1}^{\pi,\mathcal{G}}(c_{h+1}) \right],$$

$$\widehat{Q}_{i,h}^{\pi,\mathcal{G}}(c_h, \gamma_h) := \mathbb{E}_{s_h, p_h \sim \widehat{P}_h(\cdot, \cdot \mid \widehat{c}_h)} \mathbb{E}_{\{a_{j,h} \sim \gamma_{j,h}(\cdot \mid p_{j,h})\}_{j \in [n]}}$$
$$\mathbb{E}_{s_{h+1} \sim \mathbb{T}_h(\cdot \mid s_h, a_h), o_{h+1} \sim \mathbb{O}_{h+1}(\cdot \mid s_{h+1})} \left[ r_{i,h}(s_h, a_h) + V_{i,h+1}^{\pi,\mathcal{G}}(c_{h+1}) \right],$$

for each $(i, c_h) \in [n] \times \mathcal{C}_h$ and $\gamma_h \in \Gamma_h$, where we define $\widehat{V}_{i,H+1}^{\pi,\mathcal{G}}(c_{H+1}) = 0$.

Intuitively, this definition of $\widehat{V}_{i,h}^{\pi,\mathcal{G}}(c_h)$ mimics the Bellman equation of the ground-truth value function $V_{i,h}^{\pi,\mathcal{G}}(c_h)$ by replacing the ground-truth belief $\mathbb{P}^{\mathcal{G}}(s_h, p_h \mid c_h)$ by $\widehat{P}_h(s_h, p_h \mid \widehat{c}_h)$. Next, we point out the following quantitative bound when using $\widehat{V}_{i,h}^{\pi,\mathcal{G}}(c_h)$ to approximate $V_{i,h}^{\pi,\mathcal{G}}(c_h)$.

**Lemma J.11.** For any $\pi', \pi \in \Pi$, it holds that

$$\mathbb{E}_{\pi'}^{\mathcal{G}} \left| V_{i,h}^{\pi,\mathcal{G}}(c_h) - \widehat{V}_{i,h}^{\pi,\mathcal{G}}(c_h) \right| \leq (H-h+1)^2 \epsilon_{\text{belief}},$$

where

$$\epsilon_{\text{belief}} := \max_{h \in [H]} \max_{\pi \in \Pi} \mathbb{E}_{\pi}^{\mathcal{G}} \left\| \mathbb{P}^{\mathcal{G}}(\cdot, \cdot \mid c_h) - \widehat{P}_h(\cdot, \cdot \mid \widehat{c}_h) \right\|_1.$$

*Proof.* It follows directly by combining Lemma 4 and Lemma 8 of [51]. $\qquad\square$

Meanwhile, note that although in Algorithm 7, the value functions we maintain have input $\widehat{c}_h$ instead of $c_h$ for computational efficiency, we extend the definitions of those value functions to also accept $c_h$ as inputs as follows (with a slight abuse of notation):

$$Q_{i,h}^{\text{high},k}(c_h, \gamma_h) := Q_{i,h}^{\text{high},k}(\widehat{c}_h, \gamma_h), \quad Q_{i,h}^{\text{high},k}(c_h, p_h, s_h, a_h) := Q_{i,h}^{\text{high},k}(\widehat{c}_h, p_h, s_h, a_h), \quad V_{i,h}^{\text{high},k}(c_h) := V_{i,h}^{\text{high},k}(\widehat{c}_h)$$
$$Q_{i,h}^{\text{low},k}(c_h, \gamma_h) := Q_{i,h}^{\text{low},k}(\widehat{c}_h, \gamma_h), \quad Q_{i,h}^{\text{low},k}(c_h, p_h, s_h, a_h) := Q_{i,h}^{\text{low},k}(\widehat{c}_h, p_h, s_h, a_h), \quad V_{i,h}^{\text{low},k}(c_h) := V_{i,h}^{\text{low},k}(\widehat{c}_h),$$

where we recall that $\widehat{c}_h = \text{Compress}_h(c_h)$.

**Lemma J.12** (Optimism 1 for NE/CCE). With probability $1 - \delta$, for any $k \in [K]$, for Algorithm 7, it holds that for any $i \in [n]$, $\pi'_i \in \Pi_i$, $h \in [H]$

$$Q_{i,h}^{\text{high},k}(\widehat{c}_h, \gamma_h) \geq \widehat{Q}_{i,h}^{\pi'_i \times \pi^k_{-i}, \mathcal{G}}(c_h, \gamma_h), \qquad V_{i,h}^{\text{high},k}(\widehat{c}_h) \geq \widehat{V}_{i,h}^{\pi'_i \times \pi^k_{-i}, \mathcal{G}}(c_h),$$

where we recall that $\widehat{c}_h = \text{Compress}_h(c_h)$.

*Proof.* We will prove by backward induction. Obviously, it holds for $h = H + 1$. Now we assume the lemma holds for $h + 1$, then by definition

$$Q_{i,h}^{\text{high},k}(c_h, \gamma_h) = \mathbb{E}_{s_h, p_h \sim \widehat{P}_h(\cdot, \cdot \mid \widehat{c}_h)} \mathbb{E}_{\{a_{j,h} \sim \gamma_{j,h}(\cdot \mid p_{j,h})\}_{j \in [n]}} \left[ Q_{i,h}^{\text{high},k}(c_h, p_h, s_h, a_h) \right]$$

$$= \mathbb{E}_{s_h, p_h \sim \widehat{P}_h(\cdot, \cdot \mid \widehat{c}_h)} \mathbb{E}_{\{a_{j,h} \sim \gamma_{j,h}(\cdot \mid p_{j,h})\}_{j \in [n]}} \min\{ r_{i,h}(s_h, a_h) + b_h^{k-1}(s_h, a_h)$$
$$+ \mathbb{E}_{o_{h+1} \sim \widehat{\mathbb{J}}_h^{k-1}(\cdot \mid s_h, a_h)} \left[ V_{i,h+1}^{\text{high},k}(c_{h+1}) \right], H - h + 1 \}$$

$$\geq \mathbb{E}_{s_h, p_h \sim \widehat{P}_h(\cdot, \cdot \mid \widehat{c}_h)} \mathbb{E}_{\{a_{j,h} \sim \gamma_{j,h}(\cdot \mid p_{j,h})\}_{j \in [n]}}$$
$$\min\left\{ r_{i,h}(s_h, a_h) + b_h^{k-1}(s_h, a_h) + \mathbb{E}_{o_{h+1} \sim \widehat{\mathbb{J}}_h^{k-1}(\cdot \mid s_h, a_h)} \left[ \widehat{V}_{i,h+1}^{\pi'_i \times \pi^k_{-i}, \mathcal{G}}(c_{h+1}) \right], H - h + 1 \right\},$$

where the last step is by inductive hypothesis. Now note that for any $(s_h, p_h, a_h)$, we have

$$b_h^{k-1}(s_h, a_h) + \mathbb{E}_{o_{h+1} \sim \widehat{\mathbb{J}}_h^{k-1}(\cdot \mid s_h, a_h)} \left[ \widehat{V}_{i,h+1}^{\pi'_i \times \pi^k_{-i}, \mathcal{G}}(c_{h+1}) \right]$$

$$\geq b_h^{k-1}(s_h, a_h) - (H-h) \|\widehat{\mathbb{J}}_h^{k-1}(\cdot \mid s_h, a_h) - \mathbb{P}^{\mathcal{G}}(\cdot \mid s_h, a_h)\|_1 + \mathbb{E}_{s_{h+1} \sim \mathbb{T}_h(\cdot \mid s_h, a_h), o_{h+1} \sim \mathbb{O}_{h+1}(\cdot \mid s_{h+1})} \left[ \widehat{V}_{i,h+1}^{\pi'_i \times \pi^k_{-i}, \mathcal{G}}(c_{h+1}) \right]$$

$$\geq \mathbb{E}_{s_{h+1} \sim \mathbb{T}_h(\cdot \mid s_h, a_h), o_{h+1} \sim \mathbb{O}_{h+1}(\cdot \mid s_{h+1})} \left[ \widehat{V}_{i,h+1}^{\pi'_i \times \pi^k_{-i}, \mathcal{G}}(c_{h+1}) \right],$$

where we notice $\mathbb{P}^{\mathcal{G}}(o_{h+1} \mid s_h, a_h) = \sum_{s_{h+1}} \mathbb{O}_{h+1}(o_{h+1} \mid s_{h+1}) \mathbb{T}_h(s_{h+1} \mid s_h, a_h)$ for the first inequality, and the second inequality comes from the construction of our bonus $b_h^{k-1}(s_h, a_h)$ in Equation (J.3) and Lemma J.10. Meanwhile, by the definition of value functions, it holds that $\mathbb{E}_{s_{h+1} \sim \mathbb{T}_h(\cdot \mid s_h, a_h), o_{h+1} \sim \mathbb{O}_{h+1}(\cdot \mid s_{h+1})} \left[ \widehat{V}_{i,h+1}^{\pi'_i \times \pi^k_{-i}, \mathcal{G}}(c_{h+1}) \right] \leq H - h$. Therefore, we have

$$\min\left\{ r_{i,h}(s_h, a_h) + b_h^{k-1}(s_h, a_h) + \mathbb{E}_{o_{h+1} \sim \widehat{\mathbb{J}}_h^{k-1}(\cdot \mid s_h, a_h)} \left[ V_{i,h+1}^{\pi'_i \times \pi^k_{-i}, \mathcal{G}}(c_{h+1}) \right], H - h + 1 \right\}$$

$$\geq r_{i,h}(s_h, a_h) + \mathbb{E}_{s_{h+1} \sim \mathbb{T}_h(\cdot \mid s_h, a_h), o_{h+1} \sim \mathbb{O}_{h+1}(\cdot \mid s_{h+1})} \left[ \widehat{V}_{i,h+1}^{\pi'_i \times \pi^k_{-i}, \mathcal{G}}(c_{h+1}) \right].$$

Now we conclude

$$Q_{i,h}^{\text{high},k}(c_h, \gamma_h)$$
$$\geq \mathbb{E}_{s_h, p_h \sim \widehat{P}_h(\cdot, \cdot \mid \widehat{c}_h)} \mathbb{E}_{\{a_{j,h} \sim \gamma_{j,h}(\cdot \mid p_{j,h})\}_{j \in [n]}} \mathbb{E}_{s_{h+1} \sim \mathbb{T}_h(\cdot \mid s_h, a_h), o_{h+1} \sim \mathbb{O}_{h+1}(\cdot \mid s_{h+1})} [r_{i,h}(s_h, a_h)$$
$$+ \widehat{V}_{i,h+1}^{\pi'_i \times \pi^k_{-i}, \mathcal{G}}(c_{h+1})]$$
$$= \widehat{Q}_{i,h}^{\pi'_i \times \pi^k_{-i}, \mathcal{G}}(c_h, \gamma_h).$$

By definition, we have $Q_{i,h}^{\text{high},k}(c_h, \gamma_h) = Q_{i,h}^{\text{high},k}(\widehat{c}_h, \gamma_h)$, thus proving $Q_{i,h}^{\text{high},k}(\widehat{c}_h, \gamma_h) \geq \widehat{Q}_{i,h}^{\pi'_i \times \pi^k_{-i}, \mathcal{G}}(c_h, \gamma_h)$. Now for the value function, note that

$$V_{i,h}^{\text{high},k}(c_h) = \mathbb{E}_{\omega_h} Q_{i,h}^{\text{high},k}(c_h, \{\pi^k_{j,h}(\cdot \mid \omega_{j,h}, \widehat{c}_h, \cdot)\}_{j \in [n]})$$
$$\geq \mathbb{E}_{\omega'_h} \mathbb{E}_{\omega_h} Q_{i,h}^{\text{high},k}(c_h, \pi'_{i,h}(\cdot \mid \omega'_{i,h}, c_h, \cdot), \{\pi^k_{j,h}(\cdot \mid \omega_{j,h}, \widehat{c}_h, \cdot)\}_{j \in [n] \setminus \{i\}})$$
$$\geq \mathbb{E}_{\omega'_h} \mathbb{E}_{\omega_h} \widehat{Q}_{i,h}^{\pi'_i \times \pi^k_{-i}, \mathcal{G}}(c_h, \pi'_{i,h}(\cdot \mid \omega'_{i,h}, c_h, \cdot), \{\pi^k_{j,h}(\cdot \mid \omega_{j,h}, \widehat{c}_h, \cdot)\}_{j \in [n] \setminus \{i\}})$$
$$= \widehat{V}_{i,h}^{\pi'_i \times \pi^k_{-i}, \mathcal{G}}(c_h),$$

where the first step is by the property of Bayesian CCE, and the second step is by $Q_{i,h}^{\text{high},k}(c_h, \gamma_h) \geq \widehat{Q}_{i,h}^{\pi'_i \times \pi^k_{-i}, \mathcal{G}}(c_h, \gamma_h)$ for any $\gamma_h \in \Gamma_h$ as proved above. Again by definition, we proved $V_{i,h}^{\text{high},k}(\widehat{c}_h) = V_{i,h}^{\text{high},k}(c_h) \geq \widehat{V}_{i,h}^{\pi'_i \times \pi^k_{-i}, \mathcal{G}}(c_h)$. $\qquad \square$

**Lemma J.13** (Optimism 1 for CE). With probability $1 - \delta$, for any $k \in [K]$, for Algorithm 7, it holds that for any $i \in [n]$, $m_i \in \mathcal{M}_i$, $h \in [H]$

$$Q_{i,h}^{\text{high},k}(\widehat{c}_h, \gamma_h) \geq \widehat{Q}_{i,h}^{(m_i \diamond \pi^k_i) \odot \pi^k_{-i}, \mathcal{G}}(c_h, \gamma_h)$$
$$V_{i,h}^{\text{high},k}(\widehat{c}_h) \geq \widehat{V}_{i,h}^{(m_i \diamond \pi^k_i) \odot \pi^k_{-i}, \mathcal{G}}(c_h).$$

*Proof.* We will prove by backward induction. Obviously, it holds for $h = H + 1$. Now we assume the lemma holds for $h + 1$. Now we notice that by definition,

$$Q_{i,h}^{\text{high},k}(c_h, \gamma_h) = \mathbb{E}_{s_h, p_h \sim \widehat{P}_h(\cdot, \cdot \mid \widehat{c}_h)} \mathbb{E}_{\{a_{j,h} \sim \gamma_{j,h}(\cdot \mid p_{j,h})\}_{j \in [n]}} \left[ Q_{i,h}^{\text{high},k}(c_h, p_h, s_h, a_h) \right]$$
$$= \mathbb{E}_{s_h, p_h \sim \widehat{P}_h(\cdot, \cdot \mid \widehat{c}_h)} \mathbb{E}_{\{a_{j,h} \sim \gamma_{j,h}(\cdot \mid p_{j,h})\}_{j \in [n]}} \min\{r_{i,h}(s_h, a_h) + b_h^{k-1}(s_h, a_h)$$
$$+ \mathbb{E}_{o_{h+1} \sim \widehat{\mathbb{J}}_h^{k-1}(\cdot \mid s_h, a_h)} \left[ V_{i,h+1}^{\text{high},k}(c_{h+1}) \right], H - h + 1\}$$
$$\geq \mathbb{E}_{s_h, p_h \sim \widehat{P}_h(\cdot, \cdot \mid \widehat{c}_h)} \mathbb{E}_{\{a_{j,h} \sim \gamma_{j,h}(\cdot \mid p_{j,h})\}_{j \in [n]}}$$
$$\min \left\{ r_{i,h}(s_h, a_h) + b_h^{k-1}(s_h, a_h) + \mathbb{E}_{o_{h+1} \sim \widehat{\mathbb{J}}_h^{k-1}(\cdot \mid s_h, a_h)} \left[ \widehat{V}_{i,h+1}^{(m_i \diamond \pi^k_i) \odot \pi^k_{-i}, \mathcal{G}}(c_{h+1}) \right], H - h + 1 \right\},$$

where the last step is by inductive hypothesis. Now note that for any $s_h, p_h, a_h$, we have

$$b_h^{k-1}(s_h, a_h) + \mathbb{E}_{o_{h+1} \sim \widehat{\mathbb{J}}_h^{k-1}(\cdot \mid s_h, a_h)} \left[ \widehat{V}_{i,h+1}^{(m_i \diamond \pi^k_i) \odot \pi^k_{-i}, \mathcal{G}}(c_{h+1}) \right]$$
$$\geq b_h^{k-1}(s_h, a_h) - (H - h) \|\widehat{\mathbb{J}}_h^{k-1}(\cdot \mid s_h, a_h) - \mathbb{P}^{\mathcal{G}}(\cdot \mid s_h, a_h)\|_1$$
$$+ \mathbb{E}_{s_{h+1} \sim \mathbb{T}_h(\cdot \mid s_h, a_h), o_{h+1} \sim \mathbb{O}_{h+1}(\cdot \mid s_{h+1})} \left[ \widehat{V}_{i,h+1}^{(m_i \diamond \pi^k_i) \odot \pi^k_{-i}, \mathcal{G}}(c_{h+1}) \right]$$
$$\geq \mathbb{E}_{s_{h+1} \sim \mathbb{T}_h(\cdot \mid s_h, a_h), o_{h+1} \sim \mathbb{O}_{h+1}(\cdot \mid s_{h+1})} \left[ \widehat{V}_{i,h+1}^{(m_i \diamond \pi^k_i) \odot \pi^k_{-i}, \mathcal{G}}(c_{h+1}) \right],$$

where we notice $\mathbb{P}^{\mathcal{G}}(o_{h+1} \mid s_h, a_h) = \sum_{s_{h+1}} \mathbb{O}_{h+1}(o_{h+1} \mid s_{h+1}) \mathbb{T}_h(s_{h+1} \mid s_h, a_h)$ for the first inequality, and the second inequality comes from the construction of our bonus $b_h^{k-1}(s_h, a_h)$ in Equation (J.3) and Lemma J.10. Meanwhile, by the definition of value functions, it holds that

$$\mathbb{E}_{s_{h+1}\sim\mathbb{T}_h(\cdot\,|\,s_h,a_h),o_{h+1}\sim\mathbb{O}_{h+1}(\cdot\,|\,s_{h+1})}\left[\widehat{V}_{i,h+1}^{(m_i\diamond\pi_i^k)\odot\pi_{-i}^k,\mathcal{G}}(c_{h+1})\right]\leq H-h.\text{ Therefore, we have}$$

$$
\min\{r_{i,h}(s_h,a_h)+b_h^{k-1}(s_h,a_h)
$$
$$
+\mathbb{E}_{s_{h+1},o_{h+1}\sim\widehat{\mathbb{J}}_h^{k-1}(\cdot,\cdot\,|\,s_h,a_h)}\left[V_{i,h+1}^{(m_i\diamond\pi_i^k)\odot\pi_{-i}^k,\mathcal{G}}(c_{h+1})\right],H-h+1\}
$$
$$
\geq r_{i,h}(s_h,a_h)+\mathbb{E}_{s_{h+1}\sim\mathbb{T}_h(\cdot\,|\,s_h,a_h),o_{h+1}\sim\mathbb{O}_{h+1}(\cdot\,|\,s_{h+1})}\left[\widehat{V}_{i,h+1}^{(m_i\diamond\pi_i^k)\odot\pi_{-i}^k,\mathcal{G}}(c_{h+1})\right].
$$

Now we conclude

$$Q_{i,h}^{\mathrm{high},k}(c_h,\gamma_h)$$
$$\geq \mathbb{E}_{s_h,p_h\sim\widehat{P}_h(\cdot,\cdot\,|\,\widehat{c}_h)}\mathbb{E}_{\{a_{j,h}\sim\gamma_{j,h}(\cdot\,|\,p_{j,h})\}_{j\in[n]}}$$
$$\mathbb{E}_{s_{h+1}\sim\mathbb{T}_h(\cdot\,|\,s_h,a_h),o_{h+1}\sim\mathbb{O}_{h+1}(\cdot\,|\,s_{h+1})}\left[r_{i,h}(s_h,a_h)+\widehat{V}_{i,h+1}^{(m_i\diamond\pi_i^k)\odot\pi_{-i}^k,\mathcal{G}}(c_{h+1})\right]$$
$$=\widehat{Q}_{i,h}^{(m_i\diamond\pi_i^k)\odot\pi_{-i}^k,\mathcal{G}}(c_h,\gamma_h).$$

By definition, we have $Q_{i,h}^{\mathrm{high},k}(c_h,\gamma_h)=Q_{i,h}^{\mathrm{high},k}(\widehat{c}_h,\gamma_h)$, thus proving $Q_{i,h}^{\mathrm{high},k}(\widehat{c}_h,\gamma_h)\geq\widehat{Q}_{i,h}^{(m_i\diamond\pi_i^k)\odot\pi_{-i}^k,\mathcal{G}}(c_h,\gamma_h)$. Now for the value function, note that

$$V_{i,h}^{\mathrm{high},k}(c_h)=\mathbb{E}_{\omega_h}Q_{i,h}^{\mathrm{high},k}(c_h,\{\pi_{j,h}^k(\cdot\,|\,\omega_{j,h},\widehat{c}_h,\cdot)\}_{j\in[n]})$$
$$\geq\mathbb{E}_{\omega_h}Q_{i,h}^{\mathrm{high},k}(c_h,\{\pi_{j,h}^k(\cdot\,|\,\omega_{j,h},\widehat{c}_h,\cdot)\}_{j\in[n]\setminus\{i\}},(m_{i,h}\diamond\pi_{i,h}^k)(\cdot\,|\,\omega_{i,h},\widehat{c}_h,\cdot))$$
$$\geq\mathbb{E}_{\omega_h}\widehat{Q}_{i,h}^{(m_i\diamond\pi_i^k)\odot\pi_{-i}^k,\mathcal{G}}(c_h,\{\pi_{j,h}^k(\cdot\,|\,\omega_{j,h},\widehat{c}_h,\cdot)\}_{j\in[n]\setminus\{i\}},(m_{i,h}\diamond\pi_{i,h}^k)(\cdot\,|\,\omega_{i,h},\widehat{c}_h,\cdot))$$
$$=\widehat{V}_{i,h}^{(m_i\diamond\pi_i^k)\odot\pi_{-i}^k,\mathcal{G}}(c_h),$$

where the first step is by the property of Bayesian CE, and the second step is by $Q_{i,h}^{\mathrm{high},k}(c_h,\gamma_h)\geq\widehat{Q}_{i,h}^{(m_i\diamond\pi_i^k)\odot\pi_{-i}^k,\mathcal{G}}(c_h,\gamma_h)$ for any $\gamma_h\in\Gamma_h$. Again by definition, we proved $V_{i,h}^{\mathrm{high},k}(\widehat{c}_h)=V_{i,h}^{\mathrm{high},k}(c_h)\geq\widehat{V}_{i,h}^{(m_i\diamond\pi_i^k)\odot\pi_{-i}^k,\mathcal{G}}(c_h)$. $\hfill\square$

**Lemma J.14** (Pessimism). With probability $1-\delta$, for any $k\in[K]$, for Algorithm 7, it holds that for any $i\in[n]$, $h\in[H]$

$$Q_{i,h}^{\mathrm{low},k}(\widehat{c}_h,\gamma_h)\leq\widehat{Q}_{i,h}^{\pi^k,\mathcal{G}}(c_h,\gamma_h)$$
$$V_{i,h}^{\mathrm{low},k}(\widehat{c}_h)\leq\widehat{V}_{i,h}^{\pi^k,\mathcal{G}}(c_h).$$

*Proof.* We prove by backward induction on $h$. Obviously, the lemma holds for $h=H+1$. Now we assume the lemma holds for $h+1$. Similar to the proof of the previous lemma, we note by inductive hypothesis

$$Q_{i,h}^{\mathrm{low},k}(c_h,\gamma_h)\leq\mathbb{E}_{s_h,p_h\sim\widehat{P}_h(\cdot,\cdot\,|\,\widehat{c}_h)}\mathbb{E}_{\{a_{j,h}\sim\gamma_{j,h}(\cdot\,|\,p_{j,h})\}_{j\in[n]}}$$
$$\max\left\{r_{i,h}(s_h,a_h)-b_h^{k-1}(s_h,a_h)+\mathbb{E}_{s_{h+1},o_{h+1}\sim\widehat{\mathbb{J}}_h^{k-1}(\cdot,\cdot\,|\,s_h,a_h)}\left[\widehat{V}_{i,h+1}^{\pi^k,\mathcal{G}}(c_{h+1})\right],0\right\},$$

where for any $s_h,p_h,a_h$, we have

$$-b_h^{k-1}(s_h,a_h)+\mathbb{E}_{o_{h+1}\sim\widehat{\mathbb{J}}_h^{k-1}(\cdot\,|\,s_h,a_h)}\left[V_{i,h+1}^{\mathrm{low},k}(c_{h+1})\right]$$
$$\leq -b_h^{k-1}(s_h,a_h)+(H-h)\|\widehat{\mathbb{J}}_h^{k-1}(\cdot\,|\,s_h,a_h)-\mathbb{P}^{\mathcal{G}}(\cdot\,|\,s_h,a_h)\|_1$$
$$+\mathbb{E}_{s_{h+1}\sim\mathbb{T}_h(\cdot\,|\,s_h,a_h),o_{h+1}\sim\mathbb{O}_{h+1}(\cdot\,|\,s_{h+1})}\left[\widehat{V}_{i,h+1}^{\pi^k,\mathcal{G}}(c_{h+1})\right]$$
$$\leq\mathbb{E}_{s_{h+1}\sim\mathbb{T}_h(\cdot\,|\,s_h,a_h),o_{h+1}\sim\mathbb{O}_{h+1}(\cdot\,|\,s_{h+1})}\left[\widehat{V}_{i,h+1}^{\pi^k,\mathcal{G}}(c_{h+1})\right],$$

where the last step again comes from the construction of our bonus in Equation (J.3) and Lemma J.10. Therefore, we conclude

$$Q_{i,h}^{\text{low},k}(c_h, \gamma_h) \leq \mathbb{E}_{s_h, p_h \sim \widehat{P}_h(\cdot, \cdot \mid \widehat{c}_h)} \mathbb{E}_{\{a_{j,h} \sim \gamma_{j,h}(\cdot \mid p_{j,h})\}_{j \in [n]}}$$
$$\mathbb{E}_{o_{h+1} \sim \widehat{\mathbb{J}}_h^{k-1}(\cdot \mid s_h, a_h)} \left[ r_{i,h}(s_h, a_h) + \widehat{V}_{i,h+1}^{\pi^k, \mathcal{G}}(c_{h+1}) \right]$$
$$= \widehat{Q}_{i,h}^{\pi^k, \mathcal{G}}(c_h, \gamma_h).$$

Similarly, for value function, it holds that

$$V_{i,h}^{\text{low},k}(c_h) = \mathbb{E}_{\omega_h} Q_{i,h}^{\text{low},k}(c_h, \{\pi_{j,h}^k(\cdot \mid \omega_{j,h}, \widehat{c}_h, \cdot)\}_{j \in [n]})$$
$$\leq \mathbb{E}_{\omega_h} \widehat{Q}_{i,h}^{\pi^k, \mathcal{G}}(c_h, \{\pi_{j,h}^k(\cdot \mid \omega_{j,h}, \widehat{c}_h, \cdot)\}_{j \in [n]}) = \widehat{V}_{i,h}^{\pi^k, \mathcal{G}}(c_h),$$

thus proving our lemma. $\qquad\square$

**Theorem J.15** (NE/CCE version). With probability $1 - \delta$, Algorithm 7 enjoys the regret guarantee of

$$\sum_{k \in [K]} \max_{i \in [n]} \left( \max_{\pi_i' \in \Pi_i} V_{i,1}^{\pi_i' \times \pi_{-i}^k, \mathcal{G}}(c_1^k) - V_{i,1}^{\pi^k, \mathcal{G}}(c_1^k) \right) \leq \mathcal{O}(KH^2 \epsilon_{\text{belief}} + H^2 \sqrt{SAOK \log(SAHK/\delta)} + H^2 SA \sqrt{O \log(SAHK/\delta)}).$$

Correspondingly, this implies that one can learn an $(\epsilon + H^2 \epsilon_{\text{belief}})$-NE if $\mathcal{G}$ is zero-sum and $(\epsilon + H^2 \epsilon_{\text{belief}})$-CCE if $\mathcal{G}$ is general-sum with sample complexity $\mathcal{O}(\frac{H^4 SAO \log(SAHO/\delta)}{\epsilon^2})$ and computation complexity $\text{POLY}(S, \max_{h \in [H]} |\widehat{\mathcal{C}}_h|, \max_{h \in [H]} |\mathcal{P}_h|, H, \frac{1}{\epsilon}, \log \frac{1}{\delta})$.

*Proof.* Note for any given $i \in [n]$ and $\pi_i' \in \Pi_i$, by Lemma J.12 and Lemma J.14, it holds

$$\max_{\pi_i' \in \Pi_i} V_{i,h}^{\pi_i' \times \pi_{-i}^k, \mathcal{G}}(c_h^k) - V_{i,h}^{\pi^k, \mathcal{G}}(c_h^k) \leq V_{i,h}^{\text{high},k}(c_h^k) - V_{i,h}^{\text{low},k}(c_h^k).$$

Therefore, it suffices to bound $V_{i,h}^{\text{high},k}(c_h^k) - V_{i,h}^{\text{low},k}(c_h^k)$:

$$V_{i,h}^{\text{high},k}(c_h^k) - V_{i,h}^{\text{low},k}(c_h^k)$$
$$= \mathbb{E}_{s_h, p_h \sim \widehat{P}_h(\cdot, \cdot \mid \widehat{c}_h^k)} \mathbb{E}_{\omega_h} \left[ Q_{i,h}^{\text{high},k}(c_h^k, \{\pi_{j,h}^k(\cdot \mid \cdot, \widehat{c}_h^k, \cdot)\}_{j \in [n]}) - Q_{i,h}^{\text{low},k}(c_h^k, \{\pi_{j,h}^k(\cdot \mid \cdot, \widehat{c}_h^k, \cdot)\}_{j \in [n]}) \right]$$
$$\leq \mathbb{E}_{s_h, p_h \sim \mathbb{P}^{\mathcal{G}}(\cdot, \cdot \mid c_h^k)} \mathbb{E}_{\omega_h} \left[ Q_{i,h}^{\text{high},k}(c_h^k, \{\pi_{j,h}^k(\cdot \mid \cdot, \widehat{c}_h^k, \cdot)\}_{j \in [n]}) - Q_{i,h}^{\text{low},k}(c_h^k, \{\pi_{j,h}^k(\cdot \mid \cdot, \widehat{c}_h^k, \cdot)\}_{j \in [n]}) \right] + (H - h + 1)\epsilon_h(c_h^k)$$
$$\leq \mathbb{E}_{s_h, p_h \sim \mathbb{P}^{\mathcal{G}}(\cdot, \cdot \mid c_h^k)} \mathbb{E}_{\omega_h} \mathbb{E}_{\{a_{j,h} \sim \pi_{j,h}^k(\cdot \mid \omega_{j,h}, \widehat{c}_h^k, p_{j,h})\}_{j \in [n]}} \left[ Q_{i,h}^{\text{high},k}(c_h^k, p_h, s_h, a_h) - Q_{i,h}^{\text{low},k}(c_h^k, p_h, s_h, a_h) \right] + (H - h + 1)\epsilon_h(c_h^k)$$
$$\leq Z_{k,h}^1 + Q_{i,h}^{\text{high},k}(c_h^k, p_h^k, s_h^k, a_h^k) - Q_{i,h}^{\text{low},k}(c_h^k, p_h^k, s_h^k, a_h^k) + (H - h + 1)\epsilon_h(c_h^k)$$
$$\leq Z_{k,h}^1 + \mathbb{E}_{o_{h+1} \sim \widehat{\mathbb{J}}_h^{k-1}(\cdot \mid s_h^k, a_h^k)} \left[ V_{i,h+1}^{\text{high},k}(c_{h+1}) - V_{i,h+1}^{\text{low},k}(c_{h+1}) \right] + (H - h + 1)\epsilon_h(c_h^k) + 2b_h^{k-1}(s_h^k, a_h^k)$$
$$\leq Z_{k,h}^1 + \mathbb{E}_{o_{h+1} \sim \mathbb{P}^{\mathcal{G}}(\cdot \mid s_h^k, a_h^k)} \left[ V_{i,h+1}^{\text{high},k}(c_{h+1}) - V_{i,h+1}^{\text{low},k}(c_{h+1}) \right] + (H - h + 1)\epsilon_h(c_h^k) + 3b_h^{k-1}(s_h^k, a_h^k)$$
$$\leq Z_{k,h}^1 + Z_{k,h}^2 + V_{i,h+1}^{\text{high},k}(c_{h+1}^k) - V_{i,h+1}^{\text{low},k}(c_{h+1}^k) + (H - h + 1)\epsilon_h(c_h^k) + 3b_h^{k-1}(s_h^k, a_h^k),$$

where we define the two Martingale difference sequences as follows

$$Z_{k,h}^1 := \mathbb{E}_{s_h, p_h \sim \mathbb{P}^{\mathcal{G}}(\cdot, \cdot \mid c_h^k)} \mathbb{E}_{\omega_h} \mathbb{E}_{\{a_{j,h} \sim \pi_{j,h}^k(\cdot \mid \omega_{j,h}, \widehat{c}_h^k, p_{j,h})\}_{j \in [n]}} \left[ Q_{i,h}^{\text{high},k}(c_h^k, p_h, s_h, a_h) - Q_{i,h}^{\text{low},k}(c_h^k, p_h, s_h, a_h) \right]$$
$$- \left( Q_{i,h}^{\text{high},k}(c_h^k, p_h^k, s_h^k, a_h^k) - Q_{i,h}^{\text{low},k}(c_h^k, p_h^k, s_h^k, a_h^k) \right)$$
$$Z_{k,h}^2 := \mathbb{E}_{o_{h+1} \sim \mathbb{P}^{\mathcal{G}}(\cdot \mid s_h^k, a_h^k)} \left[ V_{i,h+1}^{\text{high},k}(c_{h+1}) - V_{i,h+1}^{\text{low},k}(c_{h+1}) \right] - \left( V_{i,h+1}^{\text{high},k}(c_{h+1}^k) - V_{i,h+1}^{\text{low},k}(c_{h+1}^k) \right),$$

and the error of the belief is defined as

$$\epsilon_h(c_h^k) := \|\widehat{P}_h(\cdot, \cdot \mid \widehat{c}_h^k) - \mathbb{P}^{\mathcal{G}}(\cdot, \cdot \mid c_h^k)\|_1.$$

Since $|Z_{k,h}^1| \leq H$, $|Z_{k,h}^2| \leq H$, and $\epsilon_h(c_h^k) \leq 2$, by Azuma-Hoeffding bound, we conclude with probability $1 - 3\delta$, the following holds

$$\sum_{k,h} Z_{k,h}^1 \leq \mathcal{O}(H\sqrt{HK \log \frac{1}{\delta}}), \qquad \sum_{k,h} Z_{k,h}^2 \leq \mathcal{O}(H\sqrt{HK \log \frac{1}{\delta}}),$$

$$\sum_{k,h} \epsilon_h(c_h^k) \leq \sum_k \mathbb{E}_{\pi^k}^{\mathcal{G}}\left[\sum_h \epsilon_h(c_h)\right] + \mathcal{O}(\sqrt{HK \log \frac{1}{\delta}}) \leq KH\epsilon_{\text{belief}} + \mathcal{O}(\sqrt{HK \log \frac{1}{\delta}}).$$

Meanwhile, by the pigeonhole principle, it holds that

$$\sum_{k,h} b_h^{k-1}(s_h^k, a_h^k) \leq H\sqrt{O \log(SAHK/\delta)} \sum_{k,h} \frac{1}{\sqrt{\max\{1, N_h^{k-1}(s_h^k, a_h^k)\}}}$$

$$\leq \mathcal{O}\left(H\sqrt{O \log(SAHK/\delta)}(H\sqrt{SAK} + HSA)\right).$$

Now by Lemma J.12 and Lemma J.14 and putting everything together, we conclude

$$\sum_{k \in [K]} \max_{i \in [n]} \left(\max_{\pi_i' \in \Pi_i} \widehat{V}_{i,1}^{\pi_i' \times \pi_{-i}, \mathcal{G}}(c_1^k) - \widehat{V}_{i,1}^{\pi, \mathcal{G}}(c_1^k)\right) \leq KH^2\epsilon_{\text{belief}} + \mathcal{O}(H^2\sqrt{SAOK \log(SAHK/\delta)} + H^2SA\sqrt{O \log(SAHK/\delta)}).$$

Now by Lemma J.11, we proved the regret guarantees as follows

$$\sum_{k \in [K]} \max_{i \in [n]} \left(\max_{\pi_i' \in \Pi_i} V_{i,1}^{\pi_i' \times \pi_{-i}^k, \mathcal{G}}(c_1^k) - V_{i,1}^{\pi^k, \mathcal{G}}(c_1^k)\right)$$

$$\leq \mathcal{O}(KH^2\epsilon_{\text{belief}} + H^2\sqrt{SAOK \log(SAHK/\delta)} + H^2SA\sqrt{O \log(SAHK/\delta)}).$$

For the PAC guarantees, since we define $k^\star \in \arg\min_{i \in [n], k \in [K]} V_{i,1}^{\text{high}, k}(c_1^k) - V_{i,1}^{\text{low}, k}(c_1^k)$, we have

$$\text{CCE-gap}(\pi^{k^\star}) \leq \mathcal{O}(H^2\epsilon_{\text{belief}}) + \max_{i \in [n]} \left(V_{i,h}^{\text{high}, k^\star}(c_1^{k^\star}) - V_{i,h}^{\text{low}, k^\star}(c_1^{k^\star})\right)$$

$$\leq \mathcal{O}(H^2\epsilon_{\text{belief}}) + \left(\frac{1}{K} \sum_{k \in [K]} V_{i,h}^{\text{high}, k}(c_1^k) - V_{i,h}^{\text{low}, k}(c_1^k)\right)$$

$$\leq \mathcal{O}(H^2\epsilon_{\text{belief}} + H^2\sqrt{SAO \log(SAHK/\delta)/K} + \frac{H^2SA}{K}\sqrt{O \log(SAHK/\delta)}).$$

Finally, for two-player zero-sum games, we denote $\widehat{\pi}^{k^\star}$ to be the marginalized policy of $\pi^{k^\star}$. Then we have

$$\text{NE-gap}(\widehat{\pi}^{k^\star}) \leq \text{CCE-gap}(\pi^{k^\star}),$$

thus concluding our theorem. $\qquad \square$

**Theorem J.16** (CE version). With probability $1 - \delta$, Algorithm 7 enjoys the regret guarantee of

$$\sum_{k \in [K]} \max_{i \in [n]} \left(\max_{m_i' \in \mathcal{M}_i} V_{i,1}^{(m_i' \diamond \pi_i^k) \odot \pi_{-i}^k, \mathcal{G}}(c_1^k) - V_{i,1}^{\pi^k, \mathcal{G}}(c_1^k)\right)$$

$$\leq \mathcal{O}(KH^2\epsilon_{\text{belief}} + H^2\sqrt{SAOK \log(SAHK/\delta)} + H^2SA\sqrt{O \log(SAHK/\delta)}).$$

Correspondingly, this implies that one can learn an $(\epsilon + H^2\epsilon_{\text{belief}})$-CE with sample complexity $\mathcal{O}(\frac{H^4SAO \log(SAHO/\delta)}{\epsilon^2})$ and computation complexity $\text{POLY}(S, \max_{h \in [H]} |\widehat{\mathcal{C}}_h|, \max_{h \in [H]} |\mathcal{P}_h|, H, \frac{1}{\epsilon}, \log \frac{1}{\delta})$

*Proof.* Then proof follows as that of Theorem J.15, where we only need to change the first step of the proof as

$$\widehat{V}_{i,h}^{\pi_i' \times \pi_{-i}^k, \mathcal{G}}(c_h^k) - \widehat{V}_{i,h}^{\pi^k, \mathcal{G}}(c_h^k) \leq V_{i,h}^{\text{high}, k}(c_h^k) - V_{i,h}^{\text{low}, k}(c_h^k),$$

by Lemma J.13 and Lemma J.14, and the remaining steps are exactly the same. $\qquad \square$

**Lemma J.17** (Adapted from Theorem H.5). Algorithm 8 can learn the approximate POMDP with transition $\widehat{\mathbb{T}}_{1:H}$ and emission $\widehat{\mathbb{O}}_{1:H}$ such that for any policy $\pi \in \Pi^{\text{gen}}$ and $h \in [H]$

$$\mathbb{E}_\pi^{\mathcal{G}}\left[\|\mathbb{T}_h(\cdot \,|\, s_h, a_h) - \widehat{\mathbb{T}}_h(\cdot \,|\, s_h, a_h)\|_1 + \|\mathbb{O}_h(\cdot \,|\, s_h) - \widehat{\mathbb{O}}_h(\cdot \,|\, s_h)\|_1\right] \leq \mathcal{O}(\epsilon),$$

using sample complexity $\widetilde{\mathcal{O}}\left(\frac{S^2 AHO + S^3 AH}{\epsilon^2} + \frac{S^4 A^2 H^5}{\epsilon}\right)$ with probability $1 - \delta$.

*Proof.* Note that Algorithm 8 is essentially treating the POSG $\mathcal{G}$ as a centralized MDP and running Algorithm 4, where the only modifications we make in Algorithm 8 is that we take the controller set (see some examples of the controller set in Appendix C.3) into considerations when learning the models. Specifically, for the transition $\widehat{\mathbb{T}}_h$, what we estimate is only $\widehat{\mathbb{T}}_h(s_{h+1} \,|\, s_h, a_{\mathcal{I}_h, h})$ instead of $\widehat{\mathbb{T}}_h(s_{h+1} \,|\, s_h, a_h)$, where $\mathcal{T}_h \subseteq [n]$ is the controller set. Therefore, the sample complexity of Algorithm 8 will not be worse than that of Algorithm 4. $\qquad\square$

### Proof of Theorem 7.7:

Note that the proof idea essentially resembles that of Theorem H.6, where we construct the model $\mathcal{G}^{\text{trunc}}$ for $\mathcal{G}$ in exactly the same way as constructing $\mathcal{P}^{\text{trunc}}$ for $\mathcal{P}$. Therefore, by Corollary H.10, we have

$$\mathbb{E}_\pi^{\mathcal{G}}[\|\mathbb{P}^{\mathcal{G}}(\cdot, \cdot \,|\, c_h) - \mathbb{P}^{\mathcal{G}^{\text{trunc}}}(\cdot, \cdot \,|\, c_h)\|_1] \leq 2 \sum_{\overline{\tau}_H} |\mathbb{P}^{\pi, \mathcal{G}}(\overline{\tau}_H) - \mathbb{P}^{\pi, \mathcal{G}^{\text{trunc}}}(\overline{\tau}_H)| \leq 4HS\epsilon_1.$$

Meanwhile, we can construct $\widehat{\mathcal{G}}^{\text{trunc}}$ and $\widehat{\mathcal{G}}^{\text{sub}}$ using exactly the same way as for $\widehat{\mathcal{P}}^{\text{trunc}}$ and $\widehat{\mathcal{P}}^{\text{sub}}$, where $\widehat{\mathcal{G}}^{\text{sub}}$ is a $\gamma/2$-observable POSG.

Now, according to [51], for all the examples in Appendix C.3, there exists a compression function that maps $c_h$ to $\widehat{c}_h$ such that the size of the compressed common information is quasi-polynomial, i.e., $\widehat{C}_h \leq (AO)^{C\gamma^{-4} \log \frac{SH}{\epsilon_2}}$ for some absolute constant $C$ and $\epsilon_2 \in (0, 1)$, and the corresponding approximate belief $\{\widehat{P}_h : \widehat{\mathcal{C}}_h \to \Delta(\mathcal{S} \times \mathcal{P}_h)\}_{h \in [H]}$ satisfies that

$$\mathbb{E}_\pi^{\widehat{\mathcal{G}}^{\text{sub}}}\|\mathbb{P}^{\widehat{\mathcal{G}}^{\text{sub}}}(\cdot, \cdot \,|\, c_h) - \widetilde{P}_h(\cdot, \cdot \,|\, \widehat{c}_h)\|_1 \leq \epsilon_2.$$

Therefore, we can do the same augmentation for $\widetilde{P}_h$ on states from $\mathcal{S}_h^{\text{low}}$ to construct the approximate belief $\widehat{P}_h$ as in the proof of Theorem H.6, and the remaining steps follow from the proof of Theorem H.6. This will lead to a total of polynomial-time and polynomial-sample complexities. $\qquad\blacksquare$

### J.1 Background on Bayesian Games

The Bayesian game is a generalization of normal-form games in partially observable settings. Specifically, a Bayesian game is specified as $(n, \{\mathcal{A}_i\}_{i \in [n]}, \{\Theta_i\}_{i \in [n]}, \{r_i\}_{i \in [n]}, \mu)$, where $n$ is the number of players, $\mathcal{A}_i$ is the actor space, $\Theta_i$ is the type space, $r_i : \times_{i \in [n]}(\Theta_i \times \mathcal{A}_i) \to [0, 1]$ is the reward function, and $\mu$ is the prior distribution of the joint type. At the beginning of the game, a type $\theta = (\theta_i)_{i \in [n]}$ is drawn from the prior distribution $\mu \in \Delta(\Theta)$. Then each agent $i$ gets its own type $\theta_i$ and takes the action $a_i$. We define a pure strategy of an agent as $s_i \in \text{ST}_i := \{\Theta_i \to \mathcal{A}_i\}$. We define $J_i(s_i, s_{-i})$ to be the expected rewards for agent $i$, given the pure joint strategy $(s_i, s_{-i})$.

By definition, $J_i(s_i, s_{-i})$ can be evaluated as

$$J_i(s_i, s_{-i}) := \mathbb{E}_{\theta \sim \mu} r_i(\theta, s_i(\theta_i), s_{-i}(\theta_{-i})).$$

**Bayesian NE.** We define $\gamma^\star \in \times_{i \in [n]} \Delta(\text{ST}_i)$ is an $\epsilon$-NE is it satisfies that

$$\mathbb{E}_{s \sim \gamma^\star} J_i(s) \geq \mathbb{E}_{s \sim \gamma^\star} J_i(s_i', s_{-i}) - \epsilon, \quad \forall i \in [n], s_i' \in \text{ST}_i.$$

**Bayesian CCE.** We say a distribution of joint strategies $\gamma^\star \in \Delta(\times_{i \in [n]} \text{ST}_i)$ to be a $\epsilon$-Bayesian CCE if it satisfies

$$\mathbb{E}_{s \sim \gamma^\star} J_i(s) \geq \mathbb{E}_{s \sim \gamma^\star} J_i(s_i', s_{-i}) - \epsilon, \quad \forall i \in [n], s_i' \in \text{ST}_i.$$

**(Agent-form) Bayesian CE.** We say a distribution of joint strategies $\gamma^\star \in \Delta(\times_{i \in [n]} \text{ST}_i)$ to be an $\epsilon$-agent-form Bayesian CE if it satisfies

$$\mathbb{E}_{s \sim \gamma^\star} J_i(s) \geq \mathbb{E}_{s \sim \gamma^\star} J_i(m_i \diamond s_i, s_{-i}) - \epsilon, \quad \forall i \in [n], m_i' \in \mathcal{M}_i,$$

where $\mathcal{M}_i = \{\Theta_i \times \mathcal{A}_i \to \mathcal{A}_i\}$ is the space for strategy modification, where $m_i$ modifies $s_i$ as follows: given current type $\theta_i$ and the recommended action $a_i$, the strategy modification changes the action to the another action $m_i(\theta_i, a_i)$.

Note that Bayesian NE for zero-sum games, and (agent-form) Bayesian CE/CCE are all tractable solution concepts and can be computed with polynomial computational complexity, e.g., [29, 32, 23].

## K  Concluding Remarks and Limitations

In this paper, we aim to understand the provable benefits of privileged information for partially observable RL problems under two empirically successful paradigms, *expert distillation* [14, 64, 58] and *asymmetric actor-critic* [68, 7, 3], which represent privileged *policy* and privileged *value* learning, respectively, with an emphasis on studying *both* the computational and sample efficiencies of the algorithms. Our results (as summarized in Table 1) showed that privileged information does improve learning efficiency in a series of known POMDP subclasses. One potential limitation of our work is that we only focused on the case with *exact* state information. It remains to explore whether such an assumption can be further relaxed, e.g., when privileged state information may be biased, partially observable, or delayed, as usually happens in practice, and how our theoretical results may be affected. Meanwhile, as an initial theoretical study, we have been primarily focusing on the tabular settings (except Appendix G), and it would be interesting to extend the results to function-approximation settings to handle massively large state, action, and observation spaces in practice.

