# OpenReview forum: "Provable Partially Observable Reinforcement Learning with Privileged Information"
_NeurIPS.cc/2024/Conference — NeurIPS 2024 poster_

### Official Review · Reviewer_tn5m · 2024-07-12

**Soundness:** 3
**Presentation:** 3
**Contribution:** 2
**Rating:** 5
**Confidence:** 3

**Summary:**

This paper offers valuable insights into the use of privileged information in partially observable reinforcement learning (POMDP), presenting theoretical advancements and practical algorithms. However, challenges remain in ensuring the effectiveness of expert distillation, applicability to real-world problems, and bridging the gap between training and test environments. Addressing these limitations is essential for advancing the practical implementation of Privileged Policy Learning.

**Strengths:**

1. The paper formalizes expert distillation, highlighting its limitations in finding near-optimal policies in observable POMDPs. This formalization provides a structured approach to understanding and improving existing empirical methods.

2. The introduction of the deterministic filter condition broadens the scope of tractable POMDP models and ensures that expert distillation can achieve polynomial sample and computational complexities.

3. The investigation into multi-agent RL (MARL) frameworks with privileged information and the proposed algorithms ensure polynomial complexities.

**Weaknesses:**

1.	Expert policies in pratical problems may not be optimal [1-3]. This raises concerns about the effectiveness of Privileged Policy Learning. If the expert policies are suboptimal, the student policies distilled from them may inherit these inefficiencies, potentially leading to suboptimal performance in practical applications.

2.	A significant limitation highlighted is the dependency on access to the state of the POMDP environment. In most practical scenarios, the internal states are not observable, restricting the application of the proposed methods. The deterministic filter condition, while theoretically valuable, may not always be applicable in real-world problems where state observability is a significant challenge.

3.	The paper discusses the difficulty of obtaining the internal state in actual problems, creating a gap between the training environment (where privileged information is available) and the test environment (where it is not). This discrepancy can lead to performance degradation when the policies trained with privileged information are deployed in real-world scenarios where such information is unavailable.

[1] Wu, Yueh-Hua, et al. "Imitation learning from imperfect demonstration." International Conference on Machine Learning. PMLR, 2019.

[2] Tian, Yuandong, Qucheng Gong, and Yu Jiang. "Joint policy search for multi-agent collaboration with imperfect information." Advances in neural information processing systems 33 (2020): 19931-19942.

[3] Tang, Yu, et al. "Hierarchical reinforcement learning from imperfect demonstrations through reachable coverage-based subgoal filtering." Knowledge-Based Systems 294 (2024): 111736.

**Questions:**

The paper does not discuss the tightness of the derived bounds or whether they meet the needs of practical problems.

**Limitations:**

Lack of sufficient experimental verification

---

> ### Author Rebuttal · Authors · 2024-08-07
>
> We thank the reviewer for the valuable feedback and believe there are several important misunderstandings we would like to clarify. We address the reviewer's concerns and questions below:
>
> ---
>
> ## Regarding the potential sub-optimality of the expert
> **Firstly, the focus of our paper is to study how to distill a student policy from a given (**even the perfect**) expert policy provably.** We agree that in practice, not having a perfect expert might be an issue. But we believe this is some perspective **orthogonal** to the focus of our paper. **In fact, training a better/optimal expert policy is the focus of RL in MDPs, which has been studied extensively with many existing works.** In contrast, we studied RL in POMDPs with privileged information. Among the **first attempts** at theoretically understanding privileged information, we started with two most commonly used algorithmic paradigms: Expert Distillation and Asymmetric Actor-Critic, with many empirical successes as justifications (see Appendix B for a detailed discussion), and **in the most fundamental and basic setting**. We believe this will serve as the foundation for further studying the extended setting mentioned by the reviewer. We do believe extensions to the case without “optimal” expert policy is an important future work to explore. We will make sure to refer to these papers when motivating and discussing such generalizations.
>
> ---
>
> **More importantly, our analyses and results can be readily  generalized to the case when the expert policy is sub-optimal.** Specifically, for any $\epsilon>0$, if the expert is only $\epsilon$-optimal, i.e., the performance difference between the expert and the optimal policy is $\epsilon$, the distilled student policy's performance gap compared with the optimal policy will also only increases by at most $\epsilon$, through a direct triangle inequality argument. If needed, we can add this corollary in the next version of our paper.
>
> ---
>
> ## Regarding the access to the state
> **Depending on the access to the state is exactly the setting that we wanted to understand better theoretically, and the setting that has been widely used in practice.** This was referred to as settings with "privileged information" in the literature and our paper. The cases when such a state is *not accessible* are *not* the theme of our work here, and will have to suffer from the known computational and statistical hardness in general.
>
> Meanwhile, the deterministic filter condition is **weaker** than many existing “statistically tractable” POMDP models, and also allows us to incorporate “computationally” considerations when carefully incorporated with privileged information, i.e., our main results. Empirically, it can also serve as a good criteria for us to determine whether the empirical paradigm of expert distillation will suffer from sub-optimality or not. We will add clarifications in the revision to make these points clearer.
>
> ---
>
> ## Regarding the gap between training and testing
> We firstly acknowledge that the performance *gap* between training and testing is the major challenging of RL in POMDPs with privileged information, which is **exactly we hope and managed to address in our paper**. In other words, all the guarantees (in particular, Theorem 4.5, 5.3) of our algorithms are showing that the **deployed policy** is an optimal one for the POMDP (with **partial observations** as input), instead of the training policy (allowing **privileged state information** as input). Hence, our results exactly showed that **such a performance gap can be conquered with our more carefully algorithm design and analyses** (while the naive versions of the algorithms from existing empirical works can fail, see our Sec 3).
>
> ---
>
> ## Regarding the tightness of the bound
> We thank the reviewer for bring up this important question and discuss the tightness of our bound here.
> - For our results in Sec 4, we actually achieve **polynomial** sample complexity, which matches the best results for sample complexity [31, 39]. For the time/computational complexity, we also achieve the **polynomial** complexity, thus not improvable either (note that even planning in MDP is a P-complete problem).
> - For our results in Sec 5, we still achieve **polynomial** sample complexity, thus matching the best sample complexity as before. More importantly, we also achieve **quasi-polynomial** time/computational complexity, which is shown to be tight for the *observable POMDP* setting we are considering (see [22]).
>
> ---
>
> ## Regarding the experimental evaluation
>
>
> We **have provided new experiments** for more instances of POMDPs of larger sizes. We refer to the uploaded pdf for detailed results.
>
> ---
>
> ## Regarding the additional references
>
>
> We thank the reviewer for bringing more related literature [1, 2, 3] to our attention. We will definitely include them in our related work section in our revision.
>
> ---
>
> We hope our responses have addressed the reviewer’s concerns, and would be more than happy to answer any other questions the reviewer may have. Please do not hesitate to let us know if there are any other explanations/updates needed that may help re-evaluate our paper.

---

> ### Author Response · Authors · 2024-08-13
> **Have we addressed your concerns?**
>
> Dear Reviewer tn5m,
>
> We hope the reviewer is doing well! Since the discussion period is ending very soon, we would like to kindly ask whether our further responses have adequately addressed your remaining concerns?
>
> We understand that the reviewer has the very valid concern of whether RL in POMDP with privileged information is a meaningful setting. We admit there might be other paradigms that will complement this privileged information setting like the wonderful literature shared by the reviewer (we will make sure to include those references in our revision!). Meanwhile, we also would like to point out that our focus is a **theoretical understanding** of such a valid and popular empirical paradigm that is already extensively employed in empirical applications. Therefore, discussion whether the POMDP with privileged information setting is valid is still a quite important question, but beyond the focus of our paper.
>
> Finally, we would like thanks again for the reviewer's patience and efforts dedicated to reviewing our paper and looking forward to your replies!

---

> > ### Comment · Reviewer_tn5m · 2024-08-14
> > **Response**
> >
> > Thank for the detailed responses that address my concerns. I thus raise the score.

---

### Official Review · Reviewer_9C6t · 2024-07-12

**Soundness:** 4
**Presentation:** 3
**Contribution:** 3
**Rating:** 6
**Confidence:** 3

**Summary:**

This paper has several contributions. The authors examine using privileged information for partial observability in RL. The paper looks at two empirical paradigms, expert distillation and asymmetric actor-critic, analyzing their computational and sample efficiency under specific conditions.

In addition, the authors introduce a belief-weighted optimistic asymmetric actor-critic algorithm, providing insights into MARL with CTDE.

**Strengths:**

- There is a considerable amount of summarization and formalization, such as for the expert distillation paradigm, which is appreciated.
- Theoretical results with the proposed deterministic filter condition is a novel contribution.
- The belief-state learning contribution could be interesting for future works.

**Weaknesses:**

- Like most theoretical papers, examination of existing empirical paradigms  and evaluation of proposed method is rather lacking, while understandable, is still a weakness.
- The deterministic filter condition, if I understand correctly, is weaker than previous conditions but still rather restrictive compared to real-world tasks.
- More toy examples could be helpful along side theory work.

**Questions:**

- What is the purpose of developing the deterministic filter condition? Can you give a motivating example?

**Limitations:**

The authors are upfront about their assumptions; however, I would like to mention that reliance on specific assumptions for computational tractability might limit the generalizability of the results.

---

> ### Author Rebuttal · Authors · 2024-08-07
>
> We thank the reviewer for the valuable feedback. Please see our responses below:
>
>
> ## Regarding the examination of existing empirical paradigms
>
>
> We agree with the reviewer that understanding existing empirical paradigms is important for theory-oriented research. We believe our paper indeed examined the theoretical perspectives of the **two most important empirical paradigms that utilized privileged information** in RL, by first identifying its suboptimality (Prop. 3.1), inefficiency (Prop. 3.3), and then proposing **new** provable algorithms (Sec 4 and Sec 5). Meanwhile, examining the experimental perspectives of those empirical paradigms is indeed not the main focus of our work but is an important future work we believe.
>
> ---
>
> ## Regarding the deterministic filter condition
> To further clarify, our deterministic filter condition (Definition 3.2) serves two key purposes:
> - **Unification** of existing tractable POMDP models: It unifies several important models of tractable POMDPs, including deterministic POMDPs, Block MDPs, and k-decodable POMDPs (see Appendix E for details and novel examples).
>
>
> - Criteria for the empirical paradigm of **Expert Policy Distillation**: It represents a general class of structured POMDPs with provable efficiency for the empirical paradigm of *Expert Distillation*.
>
> ---
>
> ## Regarding more experimental results
>
>
> We have conducted more extensive experimental evaluations on POMDPs of larger sizes. The pdf results are attached to the global author rebuttal
>
> ---
>
> We hope our responses have addressed the reviewer’s concerns, and would be more than happy to answer any other questions the reviewer may have. Please do not hesitate to let us know if there are any other explanations/updates needed that may help re-evaluate our paper.

---

> ### Author Response · Authors · 2024-08-13
> **Have we addressed your concerns?**
>
> Dear Reviewer 9C6t,
> We hope that you are doing well recently! Since the discussion period is ending very soon, we would like to kindly ask whether our responses have adequately addressed your concerns?
>
> We understand that your main concerns are also on our Def 3.2 (deterministic filter condition). We would like to summarize again for our responses.
>
> - On the one hand, Def 3.2 (deterministic filter condition) is already our best efforts at trying to unify existing problems. It also further extends the boundary of our current knowledge for tractable POMDP problems.
>
> - More importantly, we also admit the potential limitations of such a condition. Therefore, in Sec.5, we do not assume this condition anymore but rather only require that observations are not totally uninformative (i.e., ruling out the $\gamma=0$ case in Assumption C.8, which is a quite weak assumption in our opinion). The only sacrifice is that the computation complexity becomes quasi-polynomial instead of polynomial. However, such quasi-polynomial dependency shown to be unimprovable either [22]
>
> Finally, we would like thanks again for the reviewer's patience and efforts dedicated to reviewing our paper and looking forward to your replies!

---

> > ### Comment · Reviewer_9C6t · 2024-08-13
> >
> > I thank the authors for their clarification.  It seems that other reviewers and I are uncertain of how much value this theoretical work would bring. I agree that as far as I can see, there is no straightforward significance. My score remains the same since I'm optimistic of the theoretical results providing insight for future research.

---

> ### Author Response · Authors · 2024-08-13
> **Thanks for your support!**
>
> **We greatly thank the reviewer for appreciating our potential theoretical insights for future research and being NOT concerned with the value of theoretical works!** Meanwhile, we also thank the reviewer for bringing up the **value of the theoretical work** and would like also to clarify for you and (potentially) other reviewers to help further understand our paper in a high-level way.
>
> In terms of **theoretical value**, we believe it has been emphasized and summarized in our Table 1 and Figure 1, which gives a complete landscape for how our approach with privileged information has offered **strict and significant** theoretical benefits in a wide variety of POMDPs. Therefore, we believe our theoretical results are sufficient enough.
>
> In terms of **empirical value**, we also provide some potential points here
>
> 1. **We gave a rigorous criteria for using privileged policy learning/expert distillation in specific problems**. This will remind the practitioners to examine how well their applications satisfy our criteria (exactly or approximately) first before really applying such a method to certain problems. Therefore, it can not only serve as a theoretical condition but also as an empirical guidance, particularly considering that the limitations of privileged policy learning/expert distillation have not been not very well understood.
>
> 2. **We revealed the potential inefficiency of vanilla asymmetric-actor critic, and highlighted the importance of the advanced decoupled belief learning + policy optimization pipeline.** Notably, this rigorously answered why it is meaningful to pursue different and advanced variants for vanilla asymmetric-actor-critic algorithms to get better efficiency.
>
> In summary, our Sec. 3, explicitly named as ***revisiting empirical paradigms*** is exactly trying to highlight/justify the value of our later theoretical results for practice. We believe different empirical practitioners can find different values from it according to what empirical paradigm they are adopting and what problem they are trying to solve.
>
> **Finally, we want to sincerely thank you again for being optimistic of the theoretical results providing insight for future research! As theoreticians, we are also making efforts for developing empirically-relevant theory (like what we are trying to do in the current submission)!**

---

### Official Review · Reviewer_AXXx · 2024-07-12

**Soundness:** 3
**Presentation:** 1
**Contribution:** 3
**Rating:** 4
**Confidence:** 2

**Summary:**

This paper presents a novel theoretical characterization of certain kinds of
POMDP's which admit efficient learning. First, related characterizations are
explored and theoretical results show that these classes of POMDP's suffer from
certain drawbacks when trying to learn policies. Based on this analysis, a new
class of POMDP's is defined using the "deterministic filter condition".
This characterization of POMDP's is used to develop a novel learning algorithm
based on a decomposition of POMDP policy learning into belief learning and a
fully-observable policy learning step. Theoretical results show that policy
learning in POMDP's satisfying the deterministic filter condition condition is
tractable given access to privileged information at training time. The paper
also presents an extension of the proposed algorithm to handle partially
observable multi-agent RL.

**Strengths:**

- This paper considers a problem of practical importance, namely how to learn
  policies when there is more information available at training time than test
  time.
- The extension to multi-agent RL is a useful inclusion.
- Giving some theoretical failings of existing approaches helps motivate the
  work.
- The theoretical analysis of the proposed algorithm is extensive.

**Weaknesses:**

- Some important sections are deferred to the appendix (related work,
  conclusion, and the discussion of experiments). I understand that space in the
  main paper is quite limited but I think it's important to include these
  sections. Maybe instead some theoretical results could be deferred to the
  appendix, especially those in section 3 concerning the shortcomings of
  existing approaches.
- More generally, so much of the discussion in the paper is in the appendix that
  the main body of the paper is quite hard to read.
- The central definition 3.2 could be explained a bit more (see questions).
- The legends on the graphs in the experimental section cover a lot of the
  relevant information in the graph.

**Questions:**

I'm not sure that my understanding of 3.2 is correct. The intuition given in the
paper is that it corresponds to allowing $\gamma$ to go to 1 in assumption C.8.
But to my understanding C.8 encodes a certain kind of approximate observability
and letting $\gamma$ go to one recovers a fully-observable environment. (I
believe this is related to the restriction that $b^s$ is a one-hot vector in
definition 3.2). But if the environment is observable then there's no need to
consider POMDP's anymore, regular MDP's should be sufficient. Could you clarify
this definition a bit more, and especially how it is distinct from assuming that
the emission function can be inverted?

**Limitations:**

Limitations of the proposed approach are discussed, although this discussion is left to the appendix.

---

> ### Author Rebuttal · Authors · 2024-08-07
>
> We thank the reviewer for the valuable feedback. We believe there are several important misunderstandings we want to clarify. Please see our responses below:
>
> ---
>
> ## Response to Point 1 of weakness
> We are thankful for the suggestions on the organization of the paper. We will re-organize the paper by moving some motivation parts to the appendix, and moving back some more related work and experiment discussions.
>
> ---
>
> ## Response to Point 2 of weakness
> We shall explain more about Definition 3.2 in the following responses to the questions. Meanwhile, we will add more discussions on Definition 3.2 in the main paper.
>
> ---
>
> ## Response to Point 3 of weakness
> We have replotted the figures for better readability. Please see the attached PDF in our global author rebuttal.
>
> ---
>
> ## Response to Questions
> Firstly, the reviewer is correct that when $\gamma$ tends to one, the condition implies a reversible emission matrix. However, **we only use this extreme example as a motivation.** This only corresponds to one extreme case of us, i.e., Block MDP (Example E.2).
>
>
> **More importantly, our condition (Def 3.2) is much more general than this relatively trivial case.** For example, we can also impose **no assumptions** on the emission **by allowing the emission to be totally uninformative/unobservable ($\gamma=0$)**, while only requiring the latent state transition to be deterministic (see Example E.1). In general, our (newly identified) condition represents a **joint effect** of transition determinism and observation informativeness, which is **of independent interest** in the POMDP literature. The $\gamma=1$ case and the deterministic latent state transition case just serve as *two extreme examples* of our condition. There are certainly more examples in between, e.g., Example E.3 and the red shade area in Fig. 1.
>
>
> **Intuitive explanations about the deterministic filter.** Firstly, the terminology of a *filter* refers to the process of *estimating the underlying unknown state using a sequence of actions and observation* (recursively). Therefore, our condition only assumes that if at step $h-1$, the *estimation* of the current states is not random, then the state estimation for the next step $h$ after taking the new action $a_{h-1}$ and receiving the new observation $o_h$ is also non-random. **Therefore, it is direct to see that requiring $o_h$ to be informative such that it can be inverted to/decode the state at step $h$ is only sufficient, but not necessary.**
>
> ---
>
> We hope our responses have addressed the reviewer’s concerns, and would be more than happy to answer any other questions the reviewer may have. Please do not hesitate to let us know if there are any other explanations/updates needed that may help re-evaluate our paper.

---

> > ### Comment · Reviewer_AXXx · 2024-08-12
> >
> > Thank you for the explanation. I am less concerned about definition 3.2 now, although it still seems quite restrictive to me. I have raised my score slightly (3 -> 4). I am still concerned about the restrictiveness of 3.2, combined with the relatively small experiments (even the new experiments) and the amount of work which is deferred to the appendix.

---

> ### Author Response · Authors · 2024-08-12
> **Thanks for your replies!**
>
> We appreciate the reviewer's feedback and are glad to hear that the previous concern regarding whether Definition 3.2 implies decodability has been resolved. Regarding the remaining concerns, we believe they can be addressed with a few additional clarifications, which we apologize to be unable to detail fully in our previous rebuttal.
>
> ---
>
> ### Concerning the Restrictiveness of Definition 3.2
>
> 1. **Firstly, Definition 3.2 is not an artificial condition disconnected from existing literature, but rather an effort to unify and further relax existing classes of POMDPs that have been extensively studied.** Specifically, this condition is broader and encompasses the following well-known classes of POMDPs:
>
>    - **Deterministic POMDP:** This is discussed in our Example E.1 and has been studied in a line of research [1, 2, 3].
>    - **Block MDP:** This is addressed in our Example E.2 and has been studied in a separate line of research [4, 5].
>    - $k$**-decodable POMDP:** This is covered in our Example E.3 and has been the focus of yet another research thread [6, 7].
>
>    These examples have already been extensively studied, and the structural assumptions they entail are considered reasonable and valuable rather than restrictive. **Thus, we believe that our condition, which unifies these seemingly unrelated assumptions and further relaxes them, should not be regarded as restrictive, especially since these stronger assumptions have been thoroughly examined.**
>
> 2. **Secondly, Definition 3.2 pertains only to the first half of our results in Section 4, while the second half of our results in Section 5 does not rely on it anymore.** Specifically, we acknowledge that Definition 3.2 may not always be satisfied in real-world applications. **Therefore, in Section 5, we assume only that the observation is not entirely uninformative, meaning we rule out the** $\gamma = 0$ **case in Assumption C.8 without making any other assumptions.** The trade-off is that the computational complexity increases from polynomial to quasi-polynomial, but this is shown to be unimprovable even in the easier problem of planning [8].
>
>
> ---
>
> ### Regarding the Experiments
>
> - Firstly, we emphasize that our primary goal is to develop solid theory rather than specific empirical algorithms. **In fact, none of the theoretical studies we previously cited [1-7] that focus on POMDPs include any experiments (except for [7] on simple environment)**. **Nonetheless, we acknowledge the importance of proof-of-concept experiments and have conducted corresponding validations**. Therefore, we think this should be viewed as a strength rather than a weakness.
> - **Regarding the problem size, our problems are not significantly smaller even compared with those examined in the empirical literature on POMDP planning.** For example, the Tiger-grid problem, one of the most famous examples in the POMDP planning literature, has 36 states, 5 actions, 17 observations, and a shorter horizon than ours (a discount factor of 0.95 implies an effective horizon of 20). Notably, experiments of this scale are conducted for the relatively easier problem of planning. Thus, we believe our experiments on the much more challenging learning problems are sufficient for a theory-oriented paper.
>
>
> ---
>
> ### Regarding the Deferred Results in the Appendix
>
> We do admit that a long appendix is necessary to present our results sufficiently and accurately. However, we believe that with one additional page in the final revision, along with the reviewer's wonderful suggestions, we can address all remaining concerns. Specifically:
>
> - We will move the results related to revisiting empirical paradigms to the appendix. Based on the reviewer's feedback, we believe that maintaining the core messages of these results in the main text is sufficient.
> - We will restore the motivating examples and explanations for Definition 3.2 to the main paper. We believe this will address most of the confusion expressed by all reviewers.
>
> Therefore, we believe these straightforward adjustments will make our presentation much clearer and avoid further confusion. Finally, we hope that our paper will be evaluated primarily on its contributions rather than on these easily fixable presentation issues.
>
> ---
>
> We are grateful to the reviewer for their dedicated efforts in reviewing our paper and for engaging in the discussion period, which has undoubtedly helped improve our work! We are looking forward to your further feedbacks!

---

> > ### Author Response · Authors · 2024-08-12
> > **References**
> >
> > [1] Jin, Chi, et al. "Sample-efficient reinforcement learning of undercomplete pomdps." Advances in Neural Information Processing Systems 33 (2020): 18530-18539.
> >
> > [2] Uehara, Masatoshi, et al. "Provably efficient reinforcement learning in partially observable dynamical systems." Advances in Neural Information Processing Systems 35 (2022): 578-592.
> >
> > [3] Uehara, Masatoshi, et al. "Computationally efficient pac rl in pomdps with latent determinism and conditional embeddings." International Conference on Machine Learning. PMLR, 2023.
> >
> > [4] Krishnamurthy, Akshay, Alekh Agarwal, and John Langford. "Pac reinforcement learning with rich observations." Advances in Neural Information Processing Systems 29 (2016).
> >
> > [5] Jiang, Nan, et al. "Contextual decision processes with low bellman rank are pac-learnable." International Conference on Machine Learning. PMLR, 2017.
> >
> > [6] Efroni, Yonathan, et al. "Provable reinforcement learning with a short-term memory." International Conference on Machine Learning. PMLR, 2022.
> >
> > [7] Guo, Jiacheng, et al. "Provably efficient representation learning with tractable planning in low-rank pomdp." International Conference on Machine Learning. PMLR, 2023.
> >
> > [8] Golowich, Noah, Ankur Moitra, and Dhruv Rohatgi. "Planning in observable pomdps in quasipolynomial time." arXiv preprint arXiv:2201.04735 (2022).

---

> ### Author Response · Authors · 2024-08-13
> **Have we addressed your remaining concerns?**
>
> We hope the reviewer is doing well! Since the discussion period is ending very soon, we would like to kindly ask whether our further responses have adequately addressed your remaining concerns? To briefly summarize and complement our further response above
> - For the restrictiveness: we believe reviewer zFhw have shared similar concerns and confusions as you before. After understanding how our condition has been used to generalize existing, extensively-studied problems and obtain stronger theoretical guarantees, reviewer zFhw has revised the evaluation from 3 to 5. Therefore, we believe the discussions there can be very informative, and if the reviewer is also interested in some more detailed technical discussions, we also kindly refer to our further response to reviewer zFhw (Response to Q.2 there).
> - For the problem size, we would like to complement that there is a large body of traditional literature on studying planning in POMDP, which also focuses on tabular POMDPs that could have the same order of size as us (as we mentioned in the response above). To connect to the modern deep RL literature, we have also made significant theoretical efforts. **In particular, we have shown that when the observation space is continuous/infinite, our algorithm just needs a simple classification oracle** (the entire Sec G discusses this setting). Therefore, we believe our algorithm can also scale to more complex applications, which can be a good future direction.
>
> Finally, we would like thanks again for the reviewer's patience and efforts dedicated to reviewing our paper! We do apologize if we have caused some confusion for the reviewer in the original submission and promise to revise it accordingly!

---

### Official Review · Reviewer_zFhw · 2024-07-16

**Soundness:** 2
**Presentation:** 1
**Contribution:** 2
**Rating:** 5
**Confidence:** 4

**Summary:**

The submission addresses both statistically and computationally efficient reinforcement learning (RL) in Partially Observable Markov Decision Processes (POMDPs). The training phase has access to hidden states, while the goal is to learn the optimal policy at test time without such access. The authors propose a sample and computation-efficient algorithm under a deterministic filter stability condition.

**Strengths:**

- The proposed setting is interesting and has significant potential for practical applications.

- The paper makes a commendable effort to cover various potential cases, enhancing the applicability of the problem settings.

**Weaknesses:**

**High-Level Critic**

- More space should have been dedicated to elaborating on the real technical question that the paper aims to resolve. With access to hidden states, the statistical hardness is virtually eliminated, so the focus could have been purely on the computational aspect. The poly-sample result is not surprising, and with the $\gamma$-observable assumption of (Golowich et al., 2022), quasi-polynomial complexity does not seem surprising either.

- The privileged information combined with the deterministic filter condition is very strong. It is unclear why Definition 3.2 was needed given the strong assumption of privileged information.

- The proposed algorithm does not seem as practical as advertised. Beyond the tabular setting, it is unclear how to implement it with general function classes, which are common in deep RL.


The current manuscript needs significant improvement in terms of writing, as detailed below.

- At the beginning of the introduction, it is confusing whether the paper is about RL in POMDPs with more information or about multi-agent RL in POMDPs. The two components investigated in this paper are orthogonal to each other, and the main result seems to be more about the former.

- The contribution of the paper needs to be more clearly articulated in the introduction. My understanding is that the paper is about (1) an algorithm that achieves both sample and computational efficiency with some new technical conditions, and (2) an extension of the proposed condition and algorithm to the multi-agent setting.

- The sentence in Lines 64-65 is confusing. What do we aim to achieve with actor-critic?

- Proposition 3.1 pertains to a specific algorithm, not to the general hardness of the problem.

- Jargon such as privileged, expert distillation, or teacher-student teaching may not help much in motivating the setting of this paper, as these are not the applications the paper pursued. It would be simpler if the problem were motivated from a theoretical perspective, focusing on technical challenges in existing works and some relevance to practice in the context of actor-critic implementation.
The decoding function $g$ in Lemma 4.3 is unclear. Definition 3.2 defines a posterior probability given $(s_{h-1}, a_{h-1}, o_h)$, but Lemma 4.3 assumes the posterior is almost deterministic. This could have been clearly stated during the problem setup. This assumption is essentially similar to the block MDP.

**Questions:**

- Definition 3.2: Does $\gamma$-observability or $\gamma$-separated PSR imply Definition 3.2?

- When (Golowich et al., 2022) already achieved quasi-computational complexity for planning, what does Theorem 5.3 improve upon it? How are the previous results in Section 4 related to this result?

**Limitations:**

There are no experiments on practical benchmarks other than synthetic small tabular POMDPs.

---

> ### Author Rebuttal · Authors · 2024-08-07
>
> We thank the reviewer for the valuable feedback, and noticed that there were several important misunderstandings we would like to clarify.
>
> ## Response to Point 1 of high-level critic:
> **For sample complexity, we respectfully disagree with the claim that with privileged information, statistical hardness is virtually eliminated.** This was exactly the point and technical contributions of the references [36,26,67] that only addressed the “sample complexity” improvement. **The key challenge is that the “output” of the learning process is still a “partially observable” policy (with partial observations as input), not one from the MDPs**.
>
> **For computation complexity, our results are not a direct combination of the poly sample learning result + quasi-poly time planning result.** Having a quasi-poly-time planning algorithm with model knowledge certainly does **NOT** necessarily imply that learning without model knowledge can also be made in quasi-poly time. In fact, the **time/computational complexity** of **learning** without model knowledge and **purely planning** with model knowledge can have **strict gaps** sometimes even in simpler models. For example, recently [24] showed that learning in block MDPs (without privileged information) is computationally (strictly) harder than supervised learning problems, despite that planning with model knowledge in block MDP is computationally easy. **Intuitively, it is the sampling errors in the learning setting that can make the computation nature of the problem fundamentally harder.** This is in the same spirit as the famous problem of *learning parity with noise*, which is also computationally hard, although solving this problem given model knowledge is trivial.
>
> **Back to our setting, access to privileged information never "assumes this challenge of sampling error away".** Therefore, our positive results on the computational complexity are **not** a direct consequence of [21, 22]. In particular, **both** the **algorithms** and **techniques** for showing "poly-sample complexities" in our paper are fundamentally different from those in [21] when there is no privileged information.
>
> ## Response to Point 2 of high-level critic:
>
> **Why deterministic filter + privileged information?**
>
> - **Empirical motivations.** Our work aims to understand practical privileged-information-based algorithms, leading to the identification of the Deterministic Filter condition after recognizing the limitations of Expert Distillation.
> - **Privileged information alone imposes no assumptions on the POMDP model itself, implying its PSPACE-hardness computationally (inherited from planning).** This is because **planning with model knowledge** is **easier** than **learning with privileged information**, where one can essentially **simulate** the problem of learning with privileged information when knowing the model knowledge.
> - **Unifications of known problem classes (see Appendix E).**
>
> ## Response to Point 3 of high-level critic:
>
> Firstly, we did **not** claim our algorithms are highly practical. Our focus was to understand existing practical paradigms, Expert Distillation, and Asymmetric Actor-Critic, in the most fundamental tabular setting, with some abstractions for theoretical analysis.
>
> **More importantly, we did have function approximation results (lines 275-277, and full results in Appendix G), where we only relied on a simple classification oracle, which is indeed quite compatible with current Deep RL implementations.**
>
> ## Response to improvements in terms of writing:
>
> **Clarifications for contributions (single-agent v.s. multi-agent):** Our main contribution is understanding empirical paradigms using privileged information and then addressing their limitations. **More importantly, results for single-agent POMDP and multi-agent POSG are both under this privileged information setup, following the same algorithmic principle.** Therefore, they are **NOT** orthogonal.
>
> **Line 45-46 (goal of actor-critic):** Asymmetric Actor-Critic is already a strong empirical algorithm, and our goal is to firstly **identify its limitations** (Prop 3.3) and then further develop both computationally and statistically efficient versions of it (Sec 5).
>
> **Regarding Proposition 3.1:** We hope to firstly **understand** the limitation of existing empirical paradigms through Proposition 3.1. Then we further use it to **motivate** a (unifying) subclass of POMDPs (Def. 3.2) under which we **managed to prove** that the paradigm of Expert Policy Distillation **can be provably efficient** (Sec 4).
>
> **Response to the last point:** The benefits of privileged information are well-studied empirically. **Those terminologies are not what we invented but are already quite standard in empirical works.**
>
> Finally, we **strongly disagree** that this assumption is "essentially similar to the block MDP". **We have provided several examples in lines 919-929, including block MDP, but significantly going beyond it.** In fact, one example of Def 3.2 can include the POMDP with latent deterministic transition **without any assumptions on the emission**.
>
> ## Response to Questions:
>
> **Firstly, neither observability or well-conditioned/regular/separated PSR condition implies Def 3.2.** To the best of our knowledge, such conditions instantiated to POMDP mainly rule out the case of *uninformative* observations. In contrast, Def 3.2 in extreme cases, can require **no assumptions on emission**.
>
> **Secondly, Thm. 5.3 improves (Golowich et al., 2022) by achieving both polynomial sample and quasi-polynomial time complexity (see more details in our response to Point 1).**
>
> **Finally, results in Sec 4 are "in parallel" with those in Sec 5.** In Sec 4, we have studied another class of POMDPs, i.e., those satisfying Def 3.2, **instead of assuming observability as in Sec. 5**. Since Def 3.2 and $\gamma$-observability **do not imply each other**, the results of Sec 4 and Sec 5 **do not imply each other, either**.

---

> ### Comment · Reviewer_zFhw · 2024-08-12
> **Thank you for the response**
>
> I thank the authors for clarifying some of my concerns. However, there are several points that still not very convincing to me:
>
>
> 1. You mention the computational hardness for learning (statistically known to be tractable) POMDPs even with access to hidden states (privileged information). It would be very helpful for me to understand this challenge by answering the following question:
>
> *Why cannot we simply learn the model through sufficiently long (but polynomial) reward-free exploration episodes (but also learn the reward and emission models), and output the result of quasi-poly planning on the estimated model?*
>
>
> 2. It seems that Definition 3.2 lacks some key intuitive explanations. Why are there no examples of $\gamma$-observability provided in Appendix E, and why is this case addressed separately in Section 5? Furthermore, what are the specific differences in the final guarantees (Theorem 4.5) between Examples E.3 and E.4?

---

> ### Author Response · Authors · 2024-08-13
> **Response to further questions of Reviewer zFhw**
>
> We thank the reviewer for the further questions and respond as follows.
>
> ---
>
> ### Response to Q.1
>
> 1. Firstly, the **computational hardness** for learning (statistically known to be tractable) POMDPs even with access to hidden states we have stated before is for the setting with **only privileged information** and **without any structural assumptions on the POMDP model.** We use it to reply to your high-level critic as follows
>     > The privileged information combined with the deterministic filter condition is very strong. It is unclear why Definition 3.2 was needed given the strong assumption of privileged information.
>
> 2.  Secondly, we only claimed that such hardness of learning problems can potentially exist in the **standard learning setting without privileged information**. **Therefore, we use it to justify why privileged information, a well-motivated empirical paradigm, also offers strict theoretical benefits in terms of computation complexity.**
>
> 3. Thirdly, we **really appreciate the reviewer's great question regarding whether reward-free suffices**. In fact, **it was our initial thought as well**. However, we note that **naively extending** the reward-free techniques from MDP to the POMDP by also learning the emission fails, as detailed below.
>
>     To briefly review, the key idea for standard reward-free exploration in MDP is to estimate the transition given some reward-free dataset $D$ at step $h\in[H]$
> $$
> \hat{\mathbb{T}}\_h(s^\prime\mid s, a)=\frac{N\_h(s, a, s^\prime)}{N\_h(s, a)},
> $$
> where $N_h(s, a, s^\prime)$ and $N_h(s, a)$ denote the count of such state-action triplets/tuples in the reward-free dataset $D$. Correspondingly, we can get the guarantee of
> $$
> V_1^{\pi, (\mathbb{T}, r)}(s_1)-V_1^{\pi, (\hat{\mathbb{T}}, r)}(s_1)\le \epsilon,
> $$
> for any reward function $r$ and policy $\pi$, where $V_1^{\pi, (\mathbb{T}, r)}(s_1)$ denotes the value of policy $\pi$ in the MDP specified by $(\mathbb{T}, r)$, and similarly, the definition extends to $V_1^{\pi, (\hat{\mathbb{T}}, r)}(s_1)$. Therefore, if we can find an optimal solution for the estimated model $(\hat{\mathbb{T}}, r)$, it is also an approximately optimal solution for the true MDP $(\mathbb{T}, r)$.
>
> Back to POMDP, we can indeed estimate the model by
> $$
> \hat{\mathbb{T}}_h(s^\prime\mid s, a)=\frac{N_h(s, a, s^\prime)}{N_h(s, a)},
> $$
> $$
> \hat{\mathbb{O}}_h(o\mid s)=\frac{N_h(s, o)}{N_h(s)},
> $$
> where again those $N_h$ denote the counts of appearance of the corresponding quantities in the reward-free dataset $D$. By a similar analysis, we can get a similar reward-free POMDP guarantees of
> $$
> V_1^{\pi, (\mathbb{T}, \mathbb{O}, r)}(s_1)-V_1^{\pi, (\hat{\mathbb{T}}, \hat{\mathbb{O}}, r)}(s_1)\le \epsilon,
> $$
> for any reward function $r$ and policy $\pi$, where $V_1^{\pi, (\mathbb{T}, \mathbb{O}, r)}(s_1)$ denotes the value of policy $\pi$ in the POMDP specified by $(\mathbb{T}, \mathbb{O}, r)$, and similarly the definition extends to $V_1^{\pi, (\hat{\mathbb{T}}, \hat{\mathbb{O}}, r)}(s_1)$. Therefore, if we can find an optimal solution for the approximate POMDP $(\hat{\mathbb{T}},\hat{\mathbb{O}}, r)$, it is also an approximately optimal solution for the true POMDP. Up to now, the analysis could be similar to that of reward-free exploration in MDP.
>
> However, even if the original POMDP satisfies $\gamma$-observability, $(\hat{\mathbb{T}},\hat{\mathbb{O}}, r)$ **is not necessarily a $\gamma$-observable POMDP. This is true even if we run the reward-free process for (polynomial) long enough episodes**, since the maximum visitation probability for certain states can be exponentially small. This will affect the estimation accuracy for corresponding rows of the emission $\hat{\mathbb{O}}$, potentially breaking its $\gamma$-observability (to **rigorously see why and how, we kindly refer to the proof of our Theorem H.5**). In other words, although such states that are inherently hard to visit do not affect the **value performance bounds**, or planning in the approximate **MDP** (since any MDP is computationally tractable), they do affect the tractability of planning in the approximate POMDP.
>
> To briefly introduce the key idea to circumvent such an issue, **we introduce a new terminal state  $s^{\text{exit}}$ and try to redirect the probabilities transitioning to those hard-to-explore states to this terminal state** and **carefully re-define the transition/emission correspondingly to ensure the *misspecified* model is $\gamma^\prime$-observable**, while making sure $\frac{\gamma^\prime}{\gamma}\ge \mathcal{O}(1)$. **Note that none of such construction or analysis along the way is proposed/needed in the standard MDP reward-free exploration framework** since it only needs to care about the **value performance bound**, rather than the **computation tractability in the approximate model.**

---

> ### Author Response · Authors · 2024-08-13
> **Response to further questions of Reviewer zFhw (Cont'd)**
>
> 4. **More importantly, we would like to emphasize again that our motivation is to understand existing practice and develop corresponding provable algorithms based on it, while the reward-free exploration + planning framework (using the algorithm from (Golowich et al., '23)) is certainly disconnected from this goal.** Specifically, our starting point was to analyze the popular pipeline of belief learning + policy optimization (**asymmetric actor-critic**). Note that **our novel techniques in Point 3 are just for one instantiation of the first step of belief learning under the specific $\gamma$-observability assumption and online exploration setting.** In practice, even if observability is not satisfied, there are many effective, empirical methods for the belief learning oracle that can be used, our policy optimization algorithm **decoupled from** the belief learning step are **still effective**. In contrast, such reward-free exploration + planning framework is restricted to the $\gamma$-observable POMDP setting only, and are different from emirical practice.
>
> 5. Finally, we would like to remind the reviewer that what the reviewer focuses on is just half of our results (Sec. 5), while in **Sec. 4, we do not need any reward-free techniques or observability assumption**, and the corresponding algorithms are also much more natural. Even in the half of the results that the reviewer focused on, our goal is not just developing **an** algorithm to achieve poly sample and quasi-poly computation complexity for the specific observable POMDP with online exploration setting at the same time, but analyzing (the flaws and enhancements of) **the** algorithmic paradigms used in practice.
>
> ---
>
> ### Response to Q.2
>
> 1. **Key intuitive explanations of Def 3.2.** Firstly, the terminology of a filter refers to the process of estimating the underlying unknown state using a sequence of actions and observation (recursively). Therefore, our condition states that if at step $h$, the estimation of the current states $s_h$ is not random, then the state estimation for the next step $h+1$ after taking the new action $a_h$ and receiving the new observation $o_{h+1}$ is also non-random.
>
> 2. **Example of $\gamma$-observable POMDP.** We apologize for not further explaining observability. This was because we thought it is one of the standard assumptions in RL theory for addressing POMDPs. Notably, another useful/related assumption is weakly-revealing assumption [39], which is indeed also equivalent to $\gamma$-observability (up to some problem-dependent factors). For specific examples, we point out that a sufficient condition is that the emission matrix has **full row-rank**, and some simple/natural examples can be found in Example B.1 of [22].
>
> 3. **Why is this case addressed separately in Section 5?**
>
> - Firstly, Sec.4 is to analyze the empirical paradigm of privileged **policy** learning/**expert policy distillation**, while Sec.5 is to analyze the empirical paradigm of privileged **value** learning/**asymmetric actor-critic**. **We have shown that the empirical paradigms in Sec.4 applied to observable POMDPs suffer from sub-optimality (Prop 3.1).** Therefore, we cannot handle this case in Sec. 4. Meanwhile, this sub-optimality further motivates us to propose our Def 3.2 (that is neither stronger or weaker than $\gamma$-observability).
>
> - A more fundamental reason is that for those problems under Def 3.2, the paradigm of Sec. 4 can actually achieve poly sample + poly computation, while it is known that the quasi-poly computation complexity for $\gamma$-observable POMDP is **unimprovable** (Golowich et al., '23). Therefore, we analyzed **another** empirical paradigm in Sec.5 and show it can be used to handle the this $\gamma$-observable POMDP case we cannot handle before.
>
> 4. **Furthermore, what are the specific differences in the final guarantees (Theorem 4.5) between Examples E.3 and E.4?**
>
>     We thank the reviewer for bringing this question that can indeed help us **further justify the provable benefits** of privileged information.
>
> -  Firstly, there are **no** specific differences in the final guarantees between Examples E.3 and E.4. In other words, Theorem 4.5 holds as long as the POMDP satisfies Def 3.2.
>
> - Secondly, **the reason why there are no differences is that our guarantees do not suffer from the exponential dependency on decoding length anymore!** Note that references [A, B] that studied Example E.3 ($k$-decodable POMDP) **has to suffer from the exponential dependency on $k$ both statistically and computationally**, which explains why they have to assume $k$ to be a small constant. In contrast, under privileged information, we show that such exponential dependency on $k$ **can be removed** (with natural and **practical-relevant**  algorithms), which further explains why we can handle the new case Example E.4 that can not be handled by existing literature.

---

> ### Author Response · Authors · 2024-08-13
> **Response to further questions of Reviewer zFhw (Cont'd)**
>
> ---
>
> We are grateful to the reviewer for their dedicated efforts in reviewing our paper and for engaging in the discussion period, which has undoubtedly helped improve our work. We are looking forward to your further feedback. Thank you again.
>
> ---
>
> [A] Efroni, Yonathan, et al. "Provable reinforcement learning with a short-term memory." International Conference on Machine Learning. PMLR, 2022.
>
> [B] Guo, Jiacheng, et al. "Provably efficient representation learning with tractable planning in low-rank pomdp." International Conference on Machine Learning. PMLR, 2023.

---

> > ### Comment · Reviewer_zFhw · 2024-08-13
> > **Thanks!**
> >
> > I appreciate the detailed responses. Now the picture is more clear to me, and I am now slightly more on the positive side.
> >
> > However, I still think the submission in the current form needs improvement in terms of writing to clearly convey key challenges and intuition. In the revised version, it would be nicer if a bit more concise and crisp version of the responses to be included accordingly.

---

> > > ### Author Response · Authors · 2024-08-13
> > > **Thanks for your dedicated efforts!**
> > >
> > > We are excited to hear that our responses help address your concerns and will make sure the discussions with you (that are quite effective in our opinion) are included in the later revised version!

---

### Author Rebuttal · Authors · 2024-08-07

## Additional experimental results

In response to the reviewers, to make our experimental evaluation more sufficient, we **have added new results** by testing our algorithms on more POMDP problems of larger size than the original problems in the paper. Meanwhile, we also addressed the problem of overlapped legends in some figures. We hope those additional experimental evaluations and re-plotted figures will address the concerns regarding the empirical evaluation of our algorithms.

---

### Decision · Program_Chairs · 2024-09-25

**Decision:**

Accept (poster)

**Comment:**

This paper presents novel analysis of partially observable RL in the case when privileged information is available during training (ability to observe ground truth states during training). In particular, the two most common privileged settings are considered---(fully observable) expert distillation and asymmetric actor-critic. Theoretical analysis of standard methods in these frameworks is conducted and a more efficient novel algorithm is proposed for the later case. Extensions to the MARL case are considered and some experiments are provided.

While partially observable RL (RL in POMDPs) is popular, there is lack of theoretical analysis of many of the methods. Similarly, many algorithms focus on the privileged setting in an attempt to improve performance by using additional information from a simulator during training. A better understanding of these algorithms as well as improved algorithms would be beneficial to the community.

The paper makes an important contribution but it is very dense and the significance of the results is somewhat unclear. Because there are multiple different types of privileged information considered as well as the multiagent (POSG) case, there are many topics covered in the paper, making it hard to fit everything in clearly. In terms of significance, there some question about how practical the analysis and algorithms are. The analysis makes some strong assumptions, such as the deterministic filter condition, but these assumptions still fit with a large portion of POMDPs (including several of those used in papers). The proposed algorithms are unlikely to scale well in their current form but they have the potential to inform future algorithms in the field. The author response and discussion was helpful for clarifying some of the confusions so the paper should be updated to include versions of this text. The additional results were also helpful to better understand the benefits of the approach. The paper would be strengthened by including more discussion about how future work could build off of the theory and algorithms in this paper to develop scalable methods that still outperform current approaches.